# Multi-omic spatial effects on high-resolution AI-derived retinal thickness

V. E. Jackson [1,2,8], Y. Wu[3,8], R. Bonelli[1,2,4,8], J. P. Owen[3], L. W. Scott[1,2], S. Farashi[1,2], Y. Kihara[3], M. L. Gantner[4], C. Egan [5,6], K. M. Williams [5,6,7], B. R. E. Ansell [1,2], A. Tufail [5,6], A. Y. Lee [3,9] & M. Bahlo [1,2,9] ✉

Retinal thickness is a marker of retinal health and more broadly, is seen as a promising biomarker for many systemic diseases. Retinal thickness measurements are procured from optical coherence tomography (OCT) as part of routine clinical eyecare. We processed the UK Biobank OCT images using a convolutional neural network to produce fine-scale retinal thickness measurements across > 29,000 points in the macula, the part of the retina responsible for human central vision. The macula is disproportionately affected by high disease burden retinal disorders such as age-related macular degeneration and diabetic retinopathy, which both involve metabolic dysregulation. Analysis of common genomic variants, metabolomic, blood and immune biomarkers, disease PheCodes and genetic scores across a fine-scale macular thickness grid, reveals multiple novel genetic loci including four on the X chromosome; retinal thinning associated with many systemic disorders including multiple sclerosis; and multiple associations to correlated metabolites that cluster spatially in the retina. We highlight parafoveal thickness to be particularly susceptible to systemic insults. These results demonstrate the gains in discovery power and resolution achievable with AI-leveraged analysis. Results are accessible using a bespoke web interface that gives full control to pursue findings.

The retina comprises multiple anatomical layers of more than ten types of highly specialised cells that enable sight[1–5], and features clinically relevant zones such as the macula, fovea and optic nerve. Many hereditary and acquired human retinal conditions impinge predominantly on the macula, such as age-related macular degeneration (AMD), diabetic retinopathy and macular telangiectasia Type 2 (MacTel)[6–8].

Optical Coherence Tomography (OCT), is a widely used imaging method that extracts information about retinal morphology, including overall retinal thickness (RT)[9]. Retinal thickness is a composite measure across several sublayers and is influenced by multiple determinants at any particular location in the macula, such as the amount of vascularisation, atrophy, ischemic damage, and the presence or absence of particular cell types, such as rod and cone cells[10]. RT is highest in the perifoveal region, dipping to a minimum at the foveal pit, located at the center of the macula, an area enriched for cone cells, yet depleted for rod cells. Such substantial morphological variation over such a small area suggests the presence of tightly spatially regulated biological processes[4,5].

The retina is part of the central nervous system, empirically confirmed with transcriptomic clustering of retina with brain[11]. The field of

[1]Population Health and Immunity Division, The Walter and Eliza Hall Institute of Medical Research, Parkville, Victoria, Australia. [2]Department of Medical Biology, The University of Melbourne, Parkville, Victoria, Australia. [3]Department of Ophthalmology, Roger and Angie Karalis Johnson Retina Center, University of Washington, Seattle, WA, USA. [4]Lowy Medical Research Institute, La Jolla, CA, USA. [5]Moorfields Eye Hospital NHS Foundation Trust, London, UK. [6]Institute of Ophthalmology, University College London, London, UK. [7]Section of Ophthalmology, King's College London, London, UK. [8]These authors contributed equally: V. E. Jackson, Y. Wu, R. Bonelli. [9]These authors jointly supervised this work: A. Y. Lee, M. Bahlo. ✉e-mail: bahlo@wehi.edu.au

oculomics is growing rapidly, with AI-enabled retinal imaging-based disease prediction models being developed for neuropsychiatric disorders such as schizophrenia[12] and cognitive decline[13], and for non-neurological diseases such as cardiovascular disease[14], with promise for preventative health care[15]. RT also has diagnostic potential for many diseases, particularly neurodegenerative disorders such as dementia, Parkinson's disease[16] and multiple sclerosis, where it was first proposed as a potential biomarker in 2011[17].

Previous genome-wide association studies (GWAS) of retinal OCT data from UK Biobank (UKBB) made use of the imaging platform's associated algorithm (TOPCON/TABS), with overall RT[18] and retinal sublayers[15,19,20] summarized over either the widely used Early Treatment of Diabetic Retinopathy Study (ETDRS) or macula 6 grids. Here, we processed retinal OCT data from UKBB, with a deep learning-based image segmentation method to produce a high-resolution RT dataset, which we used to investigate the relationships between RT and genetic variation, in addition to metabolites, blood traits, immunological traits and disease (Fig. 1).

Here, we demonstrate that retinal imaging may act as a window to the brain, by revealing associations with neurological disorders such as multiple sclerosis, and highlight retinal thickness as a potential biomarker for vascular, and endocrine disorders. Furthermore we show for several diseases and metabolites, specific regions of the macula are driving these associations. Our genetic analyses identify 294 RT associated genetic loci, highlighting the complex spatial anatomy of the macula, providing context for novel candidate genes and their biological mechanisms. Our rich multi-omics results are easily accessible in their entirety through a bespoke interactive browser (https://retinomics.org/), and set a new benchmark for complex, two dimensional image analyses in population samples.

## Results

### Retinal thickness imaging data

Retinal thickness is defined as the distance between, as well as including, the inner limiting membrane (ILM) and retinal pigment epithelial (RPE) layers (Fig. 1, inset).

OCT imaging data for at least one eye was available from 85,793 UKBB participants. All individuals with evidence of retinal diseases were excluded from the analyses (Supplementary Table 1) but individuals with other non-retinal eye diseases or self-reported vision problems, in the absence of retinal disease, were retained.

OCT images were filtered for overall quality and then processed using a deep convolutional neural network (DCNN), based on a method used in Olvera-Barrios et al[21], to produce RT estimates over a grid of 128 (superior/inferior axis) by 256 (temporal/nasal axis) pixels, followed by further quality control (QC, Supplementary Figs. 1–7). Each pixel captured an area of the retina measuring approximately 46.88 by 11.72 $\mu m^2$, with the total area analyzed covering 6000 by 6000 $\mu m^2$. Scans were aligned with the fovea at the center, with images from the left side reflected around the foveal midpoint to be anatomically aligned to the right side for further analysis (Supplementary Fig. 8). RT measurements are expressed as pixels, where a pixel corresponds to approximately 3.5 $\mu m$.

Extensive QC of the aligned RT data removed individuals, scans, and pixels with poor data (Supplementary Fig. 9). Missing pixel values were imputed using a generalized additive model to produce a complete dataset for RT for all individuals, over 29,041 pixels. Individuals with data available from both eyes had their RT values averaged such that a single set of RT values was used for each individual for all subsequent analyses. The final dataset included 54,844 participants with OCT data, with 36,653 with images for both eyes and 18,191 with images for one eye only.

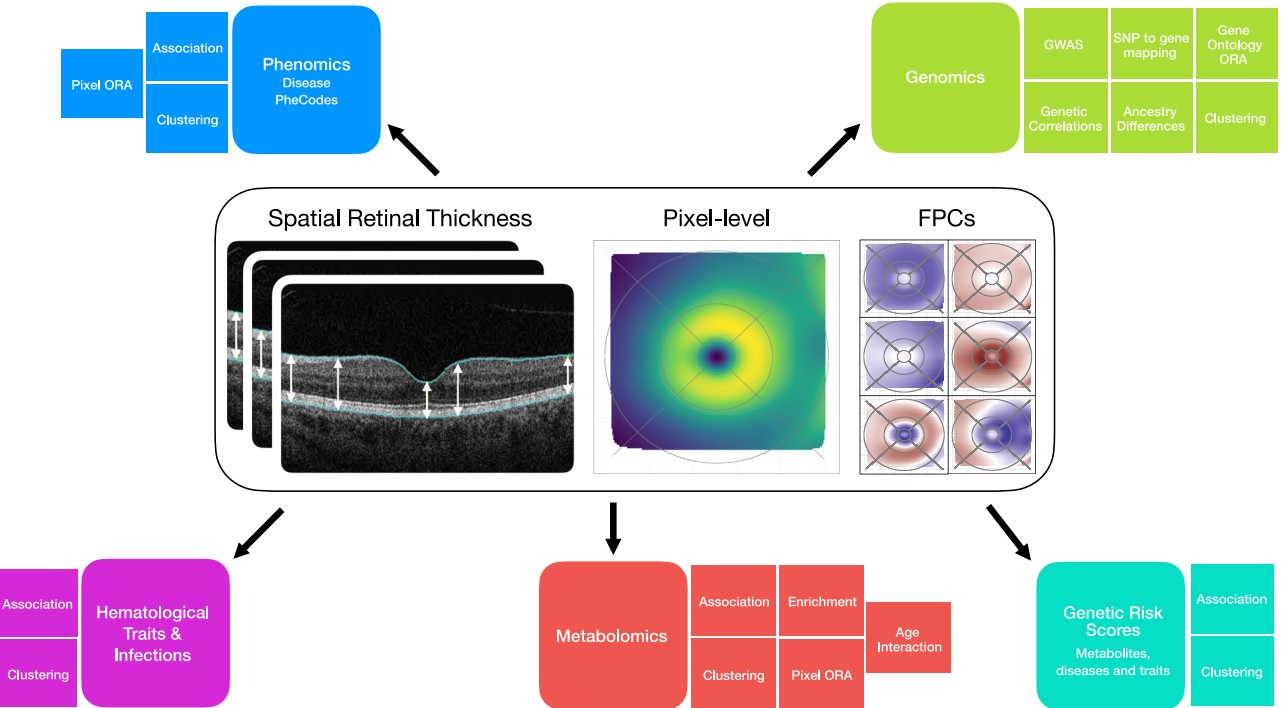

**Fig. 1 | Retinal thickness definition and overview of analyses.** Definition of retinal thickness used as the primary outcome measure, based on retinal thickness produced by a deep conovlutional neural network (DCNN). RT captures the thickness between the internal limiting membrane and the retinal pigment epithelium. RT association analyses included disease PheCodes, genomics, hematological traits & infections, metabolomics and genetic risk score associations. ORA over-representation analyses, GWAS genome-wide association studies, FPCs Functional Principal Components. OCT scan image reproduced by kind permission of UK Biobank.

## Functional principal component analysis captures broad spatial patterns

Functional principal component (FPC) analysis is a dimensionality reduction method, akin to principal component analysis, which estimates the primary modes of variation of functional data, such as curves or surfaces. We applied this approach to the reprocessed scans (RT measurements across all 29,041 pixels), using tensor product splines as base functions, using the MFPCA R package[22,23]. Scree plots revealed that six FPCs captured ~95% of the RT variation (Supplementary Fig. 10), each capturing striking spatial patterns. The variation explained by each FPC was assessed by visual examination of plots showing: i) the correlation between the FPC scores and each pixel, and ii) the mean pixel-level RT measures for the 100 individuals with the highest FPC scores, versus the 100 individuals with the lowest FPC scores (Fig. 2 and Supplementary Fig. 11). FPCs 1 and 2 capture variation in the perifovea, and outer retinal regions; FPC3 corresponds to thickening on the nasal side; FPC4 represents reduced parafoveal thickness; FPC5 captures the shape of the foveal pit; and FPC6 denotes thickening of the parafovea and nasal side. For each individual, FPC scores were extracted, representing the contribution of each FPC to that individual's scan.

## Overview of analysed cohort

For all analyses, the cohort was restricted to unrelated individuals, whose broad continental ancestries were European (EUR, $n = 43,148$ individuals), Central and South Asian (CSA, $n = 1179$) or African (AFR, $n = 1161$), based on genetic similarity to individuals from the 1000 Genomes Project and Human Genome Diversity Panel. The outcome RT datasets used for analyses consisted of 29,041 pixels (pixel-level analyses) and six FPC scores (FPC analyses), defined above. Basic characteristics of the analysed cohort are given in Supplementary Table 2, and summary RT measures, stratified by ancestry are shown in Supplementary Fig. 12.

We examined RT (FPCs and pixel-level) for relationships with characteristics previously shown to be associated with RT. On the pixel-level, we recapitulated previously reported effects for age, sex and spherical equivalent (Supplementary Fig. 13)[24]. Males had higher RT values than females, particularly in the fovea and parafovea. Age had a non-linear relationship with RT, with a maximum at 54 years, declining thereafter. Higher values of spherical equivalent (greater hyperopia) showed highly significant associations with increased RT values in the para- and perifovea, and more modest associations with lower RT values within the fovea. In addition, increased standing height, as a proxy for body size, was associated with higher measures of RT in the parafovea. FPCs were also associated with these factors, to differing degrees (Supplementary Table 3).

For the subsequent multi-omics and clinical variables analyses, all associations with RT were investigated with adjustment for age, age-squared, sex, standing height, spherical equivalent and ancestry (10 ancestry Principal Components), with device and eye (left, right, both) as technical covariates. The multi-omics study data analysis plan is summarized in Fig. 1, with sample sizes for each analysis in Supplementary Table 4.

## Genetic association analyses

GWAS were undertaken using genotypes imputed to the combined Haplotype Reference Consortium and UK10K panel[25]. The primary discovery analyses were conducted on the unrelated EUR ancestry individuals, with secondary analyses on the CSA and AFR individuals, to assess ancestral heterogeneity. Associations with 11,239,006 SNPs from the 22 autosomes and the X chromosome, were examined using PLINK 2 (version 20221024). For all GWAS, inflation was assessed using the LD-score regression intercept, as implemented by LDSC v.1.0.1[26]. LD-score regression intercepts close to 1 show that results were well calibrated (Supplementary Fig. 14). Consolidation of all GWAS results from across the 29,041 pixel-level and 6 FPCs was achieved using an ad hoc iterative procedure that incorporated genomic spatial correlation, via LD clumping, to allocate SNPs and RT pixels/FPCs to independent loci (Supplementary Fig. 15). We report loci meeting Bonferroni corrected genome-wide significance thresholds for each approach: $p < 5e-8 / 29,021 = 1.72e-12$ for the pixel-level analyses; $p < 5e-8 / 6 = 8.33e-9$ for the FPC analyses.

## Number and distribution of loci identified through both FPC and pixel-level analysis

We identify 224 RT-associated genetic loci that met the pixel-level Bonferroni corrected threshold (Supplementary Data 1). Seven loci harboured secondary signals, giving 231 independently associated signals in total. The number of loci that achieve significance for each pixel has a symmetric distribution with a mean of 22.8 loci (sd = 5.7, min = 4, max = 42) per pixel (Supplementary Fig. 16). The density of the number of loci forms concentric rings mainly concentrated in the foveal region, suggesting changing patterns of association. These rings may represent the echoes of retinal development.[27] A similar pattern was observed with the heritability estimates of pixel-level RT (Supplementary Fig. 17), with estimates ranging from 0.180 to 0.406 and varying across the macula, with lower heritability estimates in the foveal pit, and the highest heritability in the inner parafovea, and the nasal side of the outer perifovea. The FPC-based RT association analyses identified 127 independently associated SNPs at 120 loci (Supplementary Data 2), with 4, 11, 36, 45, 42, 23 loci meeting $p < 8.33e-9$ for FPCs 1 to 6, respectively (Fig. 2). Heritability estimates ranged from 0.027 (FPC1) to 0.339 (FPC5) (Supplementary Fig. 17).

A total of 294 RT genetic loci were identified collectively through either the pixel-level or FPC analyses, with 92 identified through both approaches. All loci were mapped to candidate genes via positional mapping, colocalization with eQTLs in retina (EyeGEx)[11], blood and brain (GTEx v7)[28], and chromatin interaction data from retina[29] and brain[30] (Supplementary Data 3–5). All genes implicated via these mappings, or annotations are given in Supplementary Data 6. Associations, and the top candidate genes with the most lines of evidence for each of the RT loci, are summarised in Fig. 3.

A majority of the loci identified in the Gao et al 2018[18] analyses reached the Bonferroni corrected significance threshold in the pixel-level and/or FPC analyses (89/140, 64%), with an additional 31 loci meeting $P < 5e-8$ (Supplementary Figs. 18 and 19). The concordance with the three previous analyses of retinal sublayers was lower with 25/47 (Currant et al.[20]), 52/111 (Currant et al.[19]) and 106/259 (Zekavat et al.[15]) loci identified through our approach.

The top two loci from our pixel-level analyses were amongst those identified previously. First, rs150488004 (pixel-wise minimum $p = 3.94e-152$, beta = 2.03; FPC2 $p = 6.54e-46$), is a long distance enhancer SNP affecting *LINC00461*, a long non-coding RNA now known to be integral for early Müller glial cell and astrocyte development in the human retina[2,31] and which has been previously identified in multiple macula phenotype GWAS[32,33]. Second, was rs3138142, coding for a synonymous SNP in *RDH5* (pixel-wise minimum $p = 3.27e-125$, beta = 0.82; FPC4 1.06e-58), which codes for the visual cycle enzyme 11-cis retinol dehydrogenase 5, essential for night vision, with mutations in this gene causing a retinal phenotype called fundus albipunctatus (OMIM #136880)[34]. Unsupervised hierarchical clustering analysis of the sentinel SNP effects from the pixel-level analysis, revealed 10 clusters impacting retinal thickness in a similar manner (Supplementary Fig. 20). Unsupervised hierarchical clustering of the pixels highlighted concentric regions of the macula similarly affected by genetic perturbations (Supplementary Fig. 21).

## Novel RT associations

There was a significant gain in the number of loci beyond previously published work (Supplementary Data 7, 8, Supplementary Fig. 18),

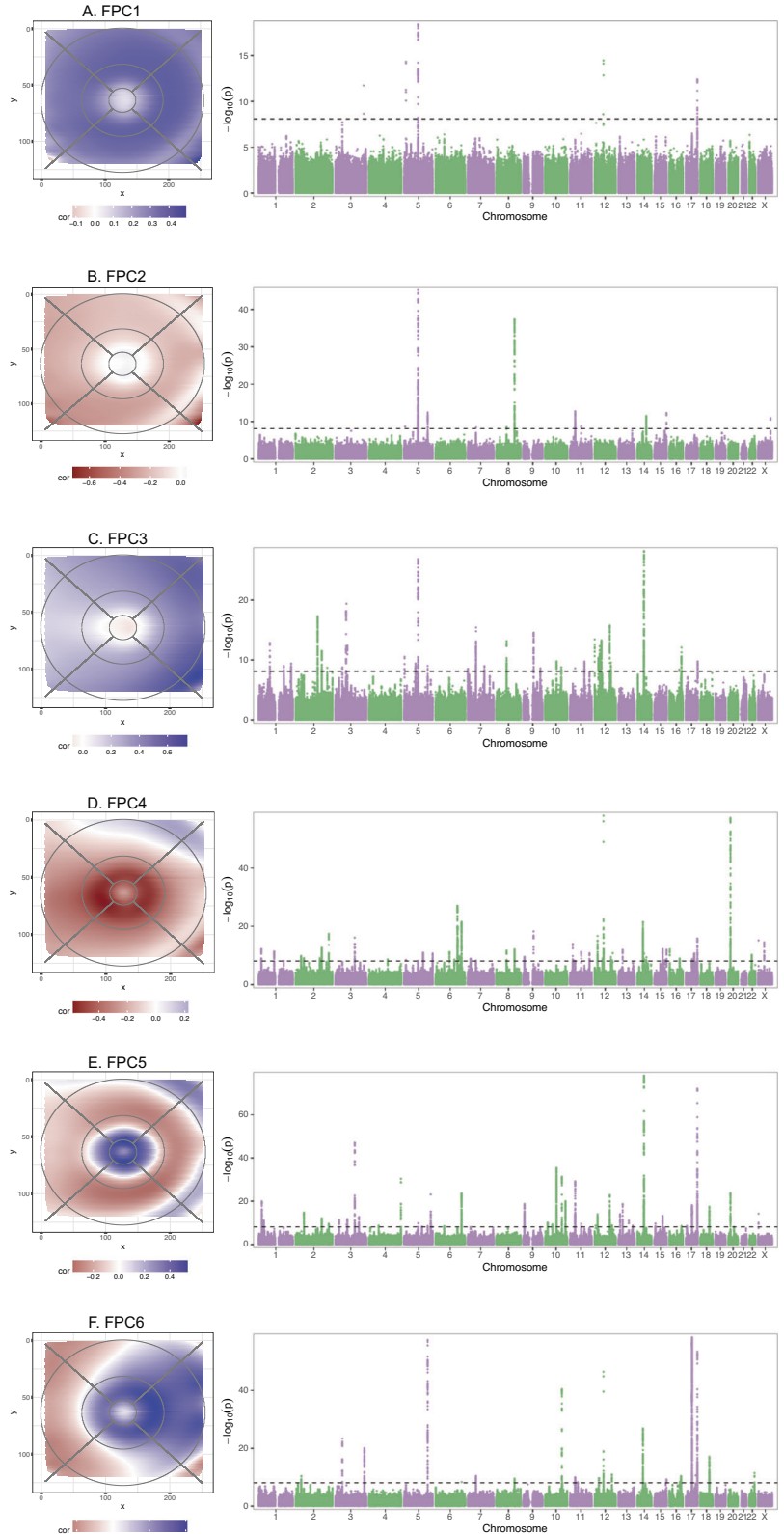

**Fig. 2 | Manhattan plots of GWAS for FPCs 1-6.** For each FPC 1-6 (**A**–**F**), Manhattan plots (right) summarising the GWAS results, with each point representing a SNP, ordered by chromosome and position (x-axis). The y-axis indicates the -log10 *p*-value (*p*-values based on a two-sided t-test, for the SNP beta in the linear regression). To the left, FPC representations are shown as the Pearson correlation coefficient between individual FPC scores and each pixel-wise RT value.

despite the substantial and strict Bonferroni testing correction. We identified 123 novel RT loci, 64 uniquely from pixel-level analysis, 26 uniquely from the FPC analysis, with 33 found in both. Many of these have been previously implicated in GWAS for ocular traits, such as the

*MACROD2* locus (pixel-level sentinel rs62202906, minimum *p* = 1.93e-54, FPC4 sentinel rs62202889, p = 5.78e-58), which has been previously identified as a suggestive finding for thyroid-associated orbitopathy, an immune-mediated eye disorder[35]. Our spatial analysis reveals that

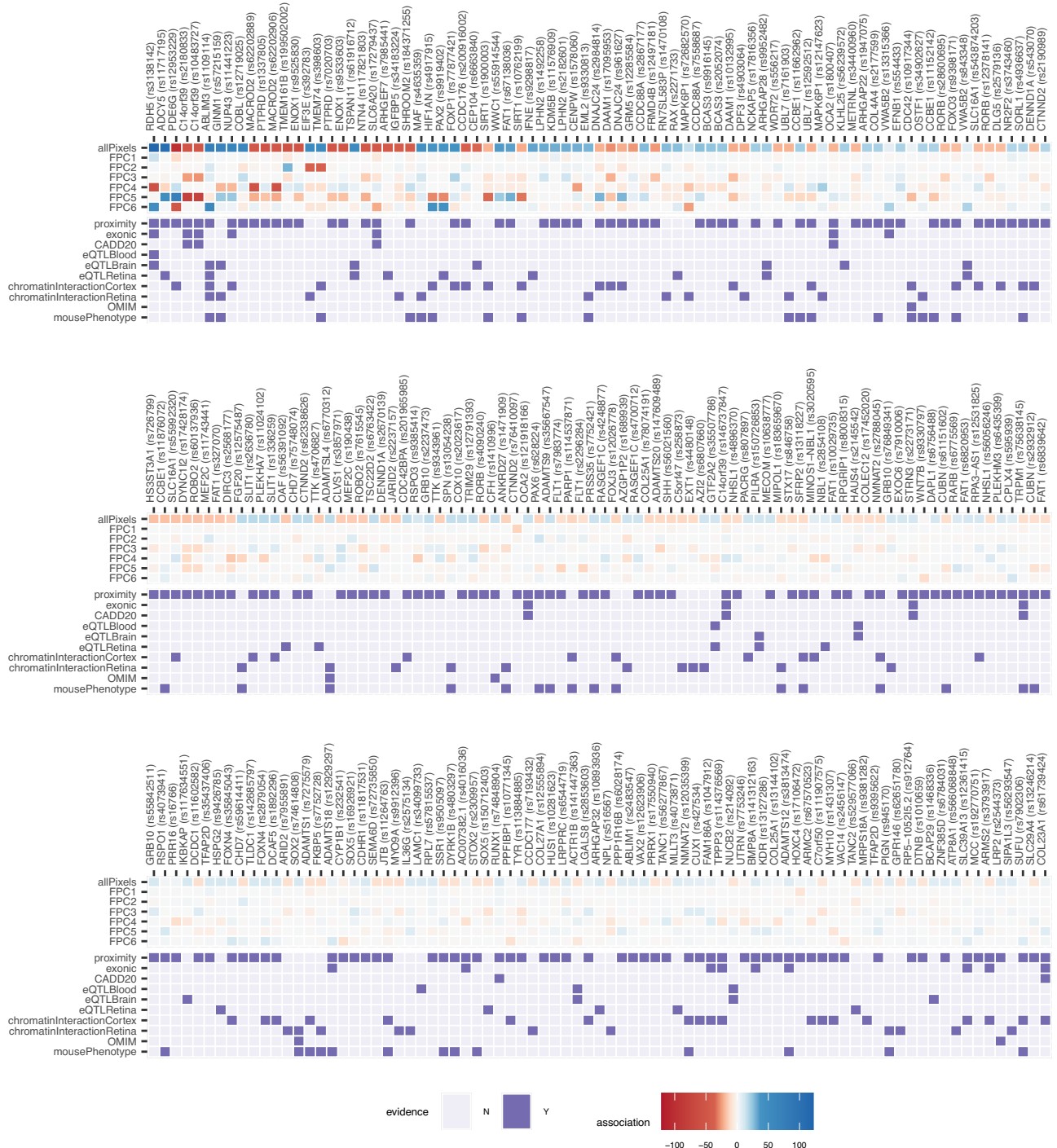

**Fig. 3 | Prioritized genes from GWAS.** Summary results for loci identified through the pixel-level and FPC GWAS. For each locus, the upper seven rows summarize the strength and direction of the genetic association; cells are coloured based on the direction of the effect estimate (blue, positive beta; red, negative beta) with the scale corresponding to the Bonferroni adjusted -log10 *p*-value for the sentinel SNP (*p*-values based on a two-sided t-test, for the SNP beta in the linear regression; Bonferroni adjusted *p*-values shown to allow comparison across pixel-level and FPC results). The bottom 10 rows summarize locus SNPs to gene mapping evidence, based on proximity to gene boundary ( < 5kB), presence of variants that are exonic; with CADD score > 20; colocalizaition with eQTLs; or overlap chromatin-interacting regions. Implicated genes with retina-associated phenotypes in OMIM and/or in mouse models, are also indicated. For each locus, the gene stated represents the top candidate gene, i.e., the gene with the most lines of evidence for that locus (highest "evidenceScore", see "methods"). All candidate genes are listed in Supplementary Data 6.

the signal mainly originates from the nasal perifoveal region, with thinning effects on the superior quadrant and thickening effects on the inferior quadrant (Fig. 4A). rs61916712 (pixel-level minimum *p* = 5.59e-56) showed a strikingly different association pattern, highly focused within the fovea (Fig. 4B); this locus overlaps a retinal eQTL for

*TSPAN11* (Posterior probability (PP) of shared GWAS and eQTL signal = 0.99, Supplementary Data 4), and is a GWAS hit for optic disc morphology[36,37]. Another novel signal rs74614808 (pixel-wise min *P* = 5.60e-15) was detected in a region with a significant chromatin interaction with *SOX2* in retina, adult and fetal cortex (Supplementary

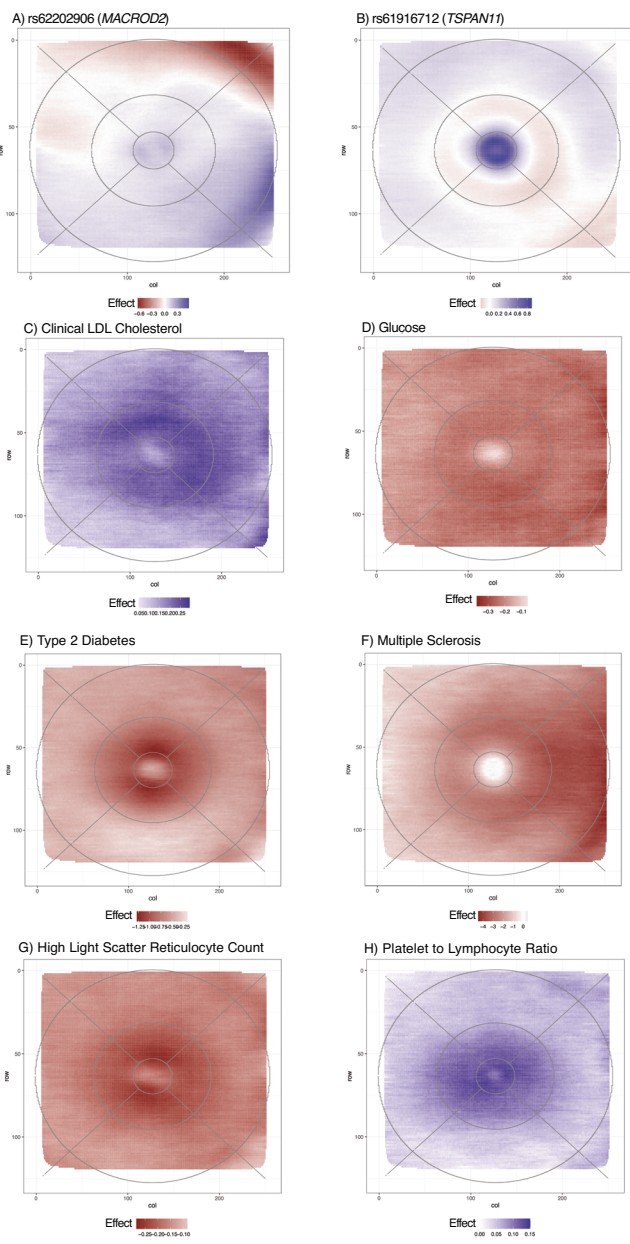

**Fig. 4 | RT spatial associations for selected 'omics signals. A** rs62202906 (*MACROD2*). **B** rs61916712 (*TSPAN11*), **C** Clinical LDL Cholesterol, **D** Glucose, **E** Type 2 Diabetes w/o complications, **F** Multiple Sclerosis, **G** High light scatter reticulocyte count, **H** Platelet to lymphocyte ratio. Blue indicates a positive effect estimate (beta) of the marker on retinal thickness, while red indicates a negative effect. In panels **A**, **B** effect size is change for each copy of the effect allele. For Panels **E**−**F** effect size is for presence of disease vs absence. In all other panels effect size relates to a 1 SD unit increase.

Data 5). Variants in *SOX2* cause microphthalmia, a birth defect in which one or both eyes fail to fully develop, with optic nerve hypoplasia and abnormalities of the central nervous system[38]. *Sox2* KO mice have abnormal optic disc and retina blood vessel morphology[39].

Amongst the novel findings were four X chromosome loci found to be associated with RT. Two of these have clear ocular roles: rs554433 (pixel-level *p* = 1.82e-24), an intronic variant in *SHROOM2*, a gene involved in pigmentation of the retinal pigment epithelium, and rs626840 in *EFNB1* (FPC4 *p* = 3.28e-15). Mutations in *EFNB1* cause craniofrontonasal syndrome, an X-linked developmental malformation, which is reported to cause a number of ophthalmologic abnormalities, including strabismus, nystagmus and hypermetropia[40] and has been

linked to glaucomatous optic neuropathy[41]. The spatial signal indicates the strongest effect in the perifoveal superior/nasal quadrant, the closest region to the optic nerve, with the risk allele leading to retinal thickening.

### Cross-ancestry comparisons, and biological insight

We assessed the effects of all loci identified in our EUR discovery analyses in the AFR and CSA ancestry individuals (Supplementary Fig. 22, Supplementary Data 9). In the pixel-level, and FPC analyses, the EUR and CSA effect estimates for the top pixel, or top FPC showed strong correlations (pixel-wise r = 0.618, *p* = 1.141e-24; FPC r = 0.630, *p* = 2.00e-14), with weaker correlations for the EUR and AFR effects (pixel-wise r = 0.305, *p* = 4.54e-06; FPC r = 0.249, *p* = 6.8e-03). For each sentinel identified in the pixel-level analysis, we also examined whether the effect estimates for all pixels across the scan were correlated between ancestries. The median correlation, for EUR versus CSA individuals was *r* = 0.306, with 28.0% of sentinel SNPs having r > = 0.5. Again, correlations were lower in EUR vs AFR (median correlation: r = 0.195; 11.5% sentinels with *r* > = 0.5).

Of the loci identified through our discovery pixel-level analysis, four were significantly associated (p < 0.05 / 224) with RT in CSA individuals: rs3138142 (*RDH5*); rs10164933 (*LINC01248*); rs1109114 (*ABLIM3*); rs183659670 (*NNAT*), with the last of these SNPs being novel. Of note, rs183659670 is associated with *NNAT* expression in the retina (PP = 0.95 of shared GWAS and eQTL signal, Supplementary Data 4); this gene encodes a proteolipid involved with brain and nervous system development. In the FPC analysis, two SNPs met the Bonferroni corrected threshold (*P* < 0.05 / 120) for CSA individuals: rs7983774 in *FLT1*, and rs1900003 near *PBLD* and *ATOH7*, both most significantly associated with FPC5. *ATOH7* encodes a member of the basic helix-loop-helix family of transcription factors, and is thought to play a role in retinal ganglion cell and optic nerve formation, with mutations in this gene causing nonsyndromic congenital retinal nonattachment[42].

We undertook a phenome-wide association study (PheWAS) for all SNPs (sentinel, and secondary SNPs) identified in the pixel-level and FPC analyses. 1,666 traits were identified with at least one SNP association (*p* < 1e-05) (Supplementary Data 10 and Supplementary Fig. 23). The top associated trait was height, which showed association with 46 pixel-wise SNPs and 25 FPC-wise SNPs. Two of the previous GWAS of retinal thickness sub-layers[20], retinal nerve fibre layer (RNFL) thickness and ganglion cell inner plexiform layer (GCIPL) thickness, were included in the PheWAS database. Both measurements were amongst the top associated phenotypes for pixel-wise (RNFL: 3rd, GCIPL: 9th) and FPC-wise (RNFL: 3rd, GCIPL: 24th) SNPs.

We additionally examined genome-wide genetic correlations between our FPC results, and a range of traits and diseases (Supplementary Fig. 24, Supplementary Data 11). The most statistically significant correlations were between FPC5 and urate levels, heart failure, reticulocytes, venous thromboembolism (VTE) and diabetes, and between FPC4 and receipt of disability allowance, a proxy for disability. The traits showing the largest magnitude correlations included ranitidine use (used to treat gastro-oesophageal reflux, FPCs 1 and 5), alcohol-related mental health problems (FPC6), cataract (FPC2), and having received psychiatric care (FPC4). Further investigation is required to understand the causality of these associations.

Gene set over-representation analysis was performed on mapped candidate genes, implicated via the pixel-level and FPC analyses, to identify enrichment of gene-ontology terms. Both analyses were enriched (Benjamini-Hochberg corrected *p* < 0.05) for genes involved in eye and visual system development and morphogenesis, pattern specification processes, and cell differentiation (Supplementary Data 12, Supplementary Fig. 25).

Given the previously demonstrated role of smoking on retinal degeneration[43] and retinal sub-layer thickness[15], we undertook sensitivity analyses for our reported sentinel SNPs, to determine the effect

of adjusting for smoking status (current/former/never). This additional adjustment made limited difference to the associations (Supplementary Fig. 26), suggesting the identified genetic associations were not acting via smoking.

## Metabolomic association analyses

For each association we describe below, the direction of effect is captured by the average beta (av. beta) parameter. A positive beta indicates that the metabolite/trait is associated with increased retinal thickness, while a negative beta represents an association with retinal thinning. After extraction and cleaning (Methods), 325 metabolic measures, from 10,668 participants were included in the association analyses. We examined associations at the single metabolite level and performed hierarchical clustering on both metabolites and pixels to investigate potential spatial effects (Methods). After correction for False Discovery Rate (FDR) using Benjamini-Hochberg multiple testing correction (Methods), all metabolites were significantly associated with RT in at least one pixel (Figs. 5A, Supplementary Data 13, Supplementary Fig. 27). The inclusion of BMI as a covariate did not change the global results (Methods, Supplementary Fig. 28). Metabolites calculated as derived lipoprotein ratios were the most significantly associated class of metabolites. The ratio of linoleic acid to total fatty acids (Linoleic Acid to Total Fatty Acids percentage) had the strongest positive effect on retinal thickness (number of sig. pixels: 29,041, average beta: 0.30, median log10(p): 11.15), while Phospholipids to Total Lipids in HDL percentage had the strongest global negative effect (number of significant pixels: 29,040, average beta: -0.25, median log10(p): 8.41). We identified 10 metabolic clusters, each associated with pixels co-located in specific retinal regions (Figs. 5B, Supplementary Data 13). Clusters 1 and 2 have the highest association with retinal thickness, and contain highly related metabolites, with enrichment for cholesterols, particularly in LDLs (Fig. 4C) and VLDLs, apoB, as well as omega 6 fatty acids. Clusters 5, 6, and 8 showed a negative association with RT and include triglycerides, glucose (Fig. 4D), branched-chain amino acids (BCAA), alanine and chylomicrons. Multiple metabolites associated with RT are linked to retinal disorders with alterations in systemic metabolites (AMD, MacTel, and diabetes). Metabolite changes that are associated with increased AMD risk[44,45], were found to be associated with retinal thinning: lower levels of cholesterol (sig. pixels: 25,606, av. beta: 0.17, median log10(p): 3.98), LDL (sig. Pixels: 26,606, av. beta: 0.16, median log10(p): 3.60), VLDL (sig. Pixels: 7,538, av. beta: 0.12, median log10(p): 1.11) and apoB (sig. Pixels: 25,957, av. beta: 0.15, median log10(p): 3.00); higher levels of triglycerides (sig. pixels: 10,825, av. beta: -0.12, median log10(p): 1.34), BCAA (sig. pixels: 17,823, av. beta: -0.12, median log10(p): 1.77) and alanine (sig. pixels: 6360, av. beta: -0.14, median log10(p): 0.85). Similarly, changes in glycine (sig. pixels: 19,586, av. beta: 0.13, median log10(p): 2.01), sphingomyelin (sig. pixels: 21,055, av. beta: 0.16, median log10(p): 2.68) and alanine that are associated with increased risk of the retinal degenerative disorder, MacTel[33,46], were also found to be associated with retinal thinning. Many of the metabolites negatively associated with retinal thickness, triglycerides, glucose (sig. pixels: 28,878, av. beta: -0.22, median log10(p): 6.98), BCAA and alanine, are elevated in metabolic syndromes and diabetes[47], which also correlates with retinal thinning through the often combined effects of retinal neuropathy, changes in vasculature and occurrence of edema. Unexpectedly, omega-3 fatty acids (sig. pixels: 3229, av. beta: -0.11, median log10(p): 0.84), which are important for retinal function[45], are negatively associated with retinal thickness. Specific retinal regions were identified as being consistently affected by metabolic disturbances. These regions corresponded, to some extent, with recognised retinal anatomical landmarks, as described with the ETDRS grid (Supplementary Fig. 8), but also revealed clusters spanning multiple sectors of the grid. Pixel over-representation analysis (ORA) revealed that the parafovea, particularly its lower temporal side, was susceptible to

metabolic influences (Fig. 5C). This was confirmed by the many metabolites significantly affecting FPC4 (Supplementary Fig. 29), which captured parafoveal thickness variation. The temporal perifoveal region was also specifically impacted by metabolic dysregulation. Age−metabolite interaction analyses, revealed a generalized increase in metabolic effect on retinal thickness with age (Supplementary Data 14). ORA revealed that pixels located in the parafoveal area were again the strongest contributors to the interaction effects on RT (Supplementary Fig. 30). Metabolic polygenic risk scores (PRSs) have recently been endorsed as a tool for trait associations[48]. Associations with metabolite genetic risk scores (GRSs) and all pixels were performed in the same fashion as for the measured metabolites (Methods, Supplementary Data 15). We found phosphatidylcholines and their lysophosphatidylcholines to be associated with retinal thickness, as well as certain amino acids such as lysine (sig. pixels: 4,373, av. beta: -0.08, median log10(p): 2.46), and threonine (sig. pixels: 17, av. beta: 0.07, median log10(p): 1.03) (Supplementary Data 15, Supplementary Fig. 31); these amino acids mainly affected the parafoveal areas.

## Disease PheCodes

We examined associations between pixel-level RT and 863 diseases as defined by PheCodes (combining ICD-9 and ICD-10 codes as defined by the Phecode Map 1.2 https://phewascatalog.org/phecodes). Data were available for 35,900 participants. After FDR correction, 386 diseases (44%) were significantly associated with RT (Supplementary Data 16, Fig. 6A and B), with the majority (82%) presenting as a thinning effect, as well as effects on multiple FPCs (Fig. 6C). More generally, even among those not statistically significant, there was a negative relationship between disease status and retinal thickness. The inclusion of smoking as an additional covariate did not change the results (Supplementary Fig. 32). Multiple sclerosis (N = 133 cases reported in UKBB, sig. pixels: 26,350, av. beta: -2.27, median log10(p): 7.64) had the largest negative global effect, with retinal thinning observed in MS patients compared to controls, and the strongest effects observed in the nasal perifoveal region closest to the optic disc (Fig. 4E). MS results in oligodendrocyte demyelination, with impacts on the optic nerve as previously reported[49]. Metabolic and cardiovascular disorders were among the diseases most strongly associated with thickness. Essential primary hypertension was associated with retinal thinning, and displayed the highest amount of global significance, likely in part due to the high prevalence of this disorder (N = 10,767, sig. pixels: 29,041, av. beta: -0.37, median log10(p): 11.17). Endocrine/metabolic class disorders were also highlighted among the top associated traits with retinal thinning, including type 2 Diabetes (T2D) (N = 2210 sig. pixels: 27,357, av. beta: -0.58, median log10(p): 8.32, Fig. 4F), hypercholesterolemia (N = 5227, sig. pixels: 17,572, av. beta: -0.25, median log10(p): 3.03), other chronic nonalcoholic liver disease (N = 397, sig. pixels: 22,419, av. beta: -0.85, median log10(p): 3.73) and gout (N = 742, sig. pixels: 17,804, av. beta: -0.44, median log10(p): 3.31). T2D and hypertension were also identified through our genetic correlation analyses, and are recognised comorbidities for multiple retinal disorders including AMD[50] and MacTel[51]. T2D has a recognised retinal complication in diabetic retinopathy, affecting about one-third of T2D patients, representing a major public health burden[52]. These results complement our metabolic RT association results, where many of the metabolites most highly associated with RT have also been found to be associated with these disorders. Investigation of retinal disorders was limited, given the study design of depletion of individuals with overt retinal disease. Nevertheless, we found retinal thinning to be associated with cataracts (N = 1153, sig. pixels: 25,865, av. beta: -0.59, median log10(p): 5.47), and senile cataracts,(N = 1023, sig. pixels: 19,164, av. beta: -0.47, median log10(p): 3.36) as well as astigmatism (N = 402, sig. pixels: 14,167, av. beta: -0.67, median log10(p): 2.67). ORA revealed a pronounced susceptibility of the parafoveal ring to different diagnoses and diseases, confirming once again the sensitivity of this

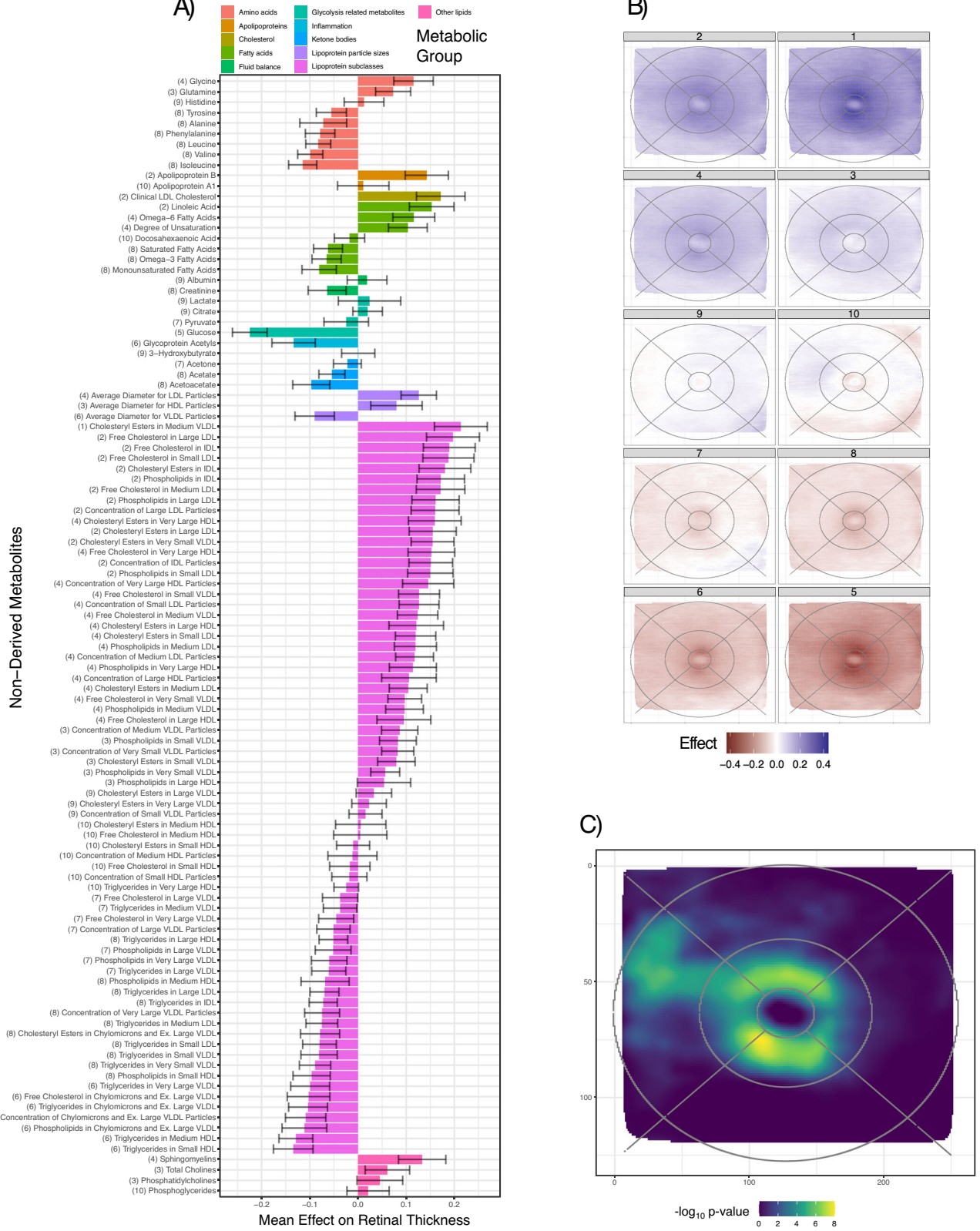

**Fig. 5 | Overview of metabolomic results. A** Mean (bar length) and standard deviation (error bars) of effect estimates (betas) for non-derived metabolite variation with RT across all $n$ = 29,041 pixels. Color represents the metabolic group of each metabolite. Cluster numbers are shown in brackets before each metabolite name. **B** Pixel-wise average effect on RT across all metabolites within each cluster from unsupervised hierarchical clustering. **C** Pixel-wise metabolic over-representation analysis results. Shown are Benjamini-Hochberg adjusted $p$-values for over-representation; $p$-values calculated using a hypergeometric distribution.

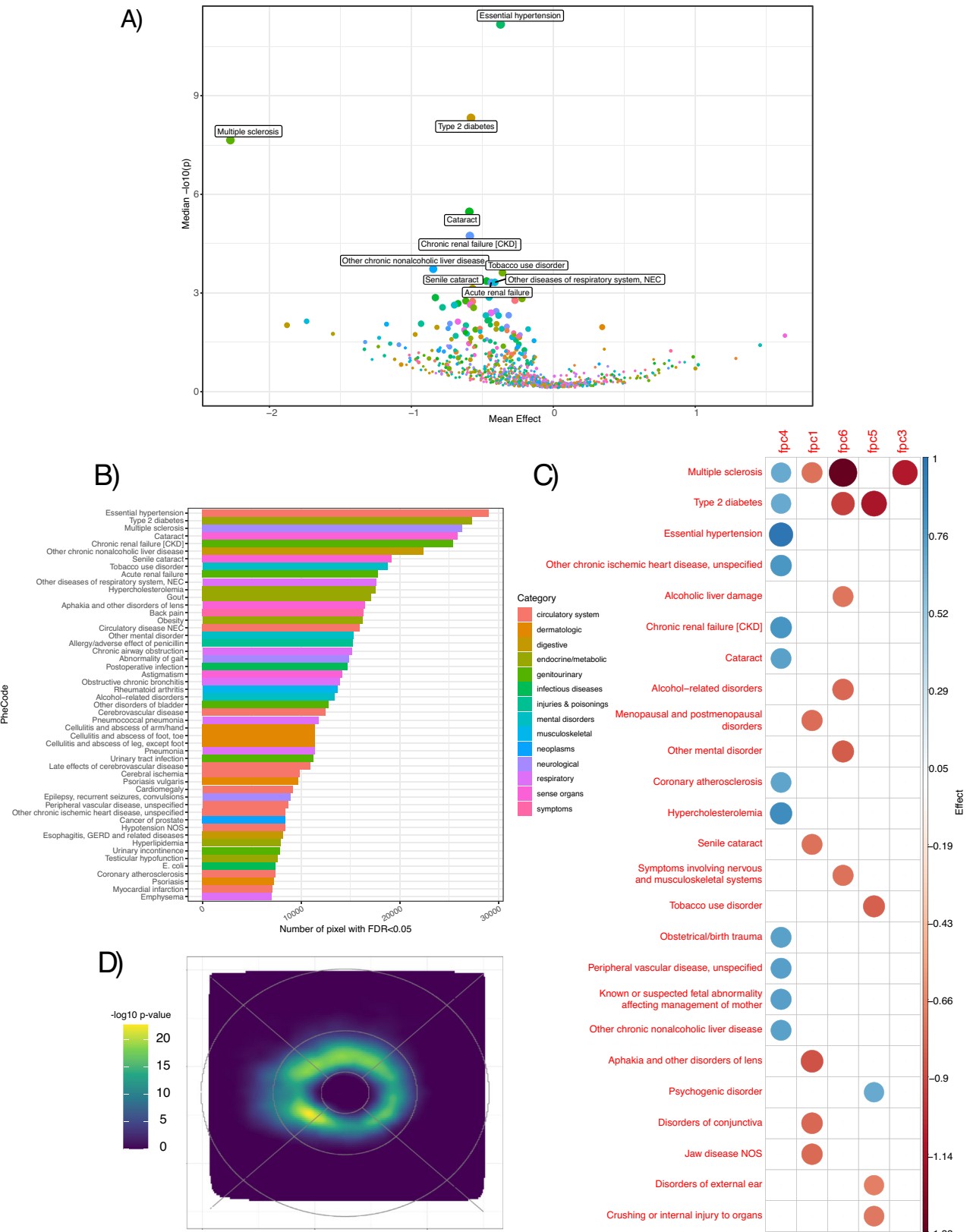

**Fig. 6 | PheCode associations with RT. A** Volcano plot showing mean effect size (beta) across across all $n = 29,041$ pixels and median -log10($p$-value) for each Phe-Code ($P$-values based on a two-sided t-test, for the PheCode beta in the linear regression model). **B** Barplot showing number of RT pixels showing significant associations with top PheCodes. **C** Heatmap showing significant association between PheCodes and RT FPCs. Color represents effect estimate direction and magnitude (red = negative, blue = positive). Size represents the magnitude of effect. **D** 2D smoothed results from Over-Representation analysis (ORA) on main PheCodes results. Shown are Benjamini-Hochberg adjusted $p$-values for over-representation; $p$-values calculated using a hypergeometric distribution.

macular zone to disease (Fig. 6D). Specifically, within this area of the retina, the effect is higher in the superior and temporal regions. We also examined associations at the pixel-level between RT, and PRS or GRS, for a number of diseases (collectively referred as genetic scores below, see "Methods"). This allowed for investigation of associations between RT and genetic susceptibility to diseases, including a number of retinal diseases, and related phenotypes. We examined 58 trait genetic scores from 43,147 participants, and after FDR correction, 48 scores were found to significantly affect the RT of at least one pixel (Supplementary Data 17, Supplementary Fig. 33). The averaged macular thickness[18] GRS was the most significantly associated (sig. pixels: 29,041, av. beta: 1.08, median log10(p): > 323), followed by retinal vascular caliber[53] (sig. pixels: 29,040, av. beta: 0.17, median log10(p): 14.83), AMD[54] (sig. pixels: 16,601, av. beta: −0.05, median log10(p): 2.80), and MacTel[33] (sig. pixels: 15,966, av. beta: 0.06, median log10(p): 2.68). The more recent Han et al[54] AMD GRS resulted in a more powerful association signal than the UKBB PRS release, which is based on Fritsche et al[55]. A higher burden of genomic risk of AMD, MacTel, T2D and celiac disease (sig. pixels: 10,198, av. beta: 0.04, median log10(p): 1.56) were all associated with parafoveal thinning. We found a significant negative association between RT and PRS for neuropsychiatric disorders such as schizophrenia (sig. pixels: 1862, av. beta: -0.03, median log10(p): 0.76), Alzheimer's (sig. pixels: 1,216, av. beta: 0.02, median log10(p): 0.47) and Parkinson's disease (sig. pixels: 1,135, av. beta: 0.01, median log10(p): 0.63) disease although patterns of association differed across diseases. PRS related to metabolic traits (e.g., HDL cholesterol, celiac disease, HbA1c, apolipoprotein A1) as well as cardiovascular disorders (e.g., cardiovascular disease, resting heart rate, ischemic stroke, and atrial fibrillation) also showed significant association with thickness in particular areas of the retina. There were 25 diseases captured through both the PheCodes and genetic score analyses in this study. For each disease, we examined the correlation of the effect estimates from these two analyses, and we found the highest association agreement for type 2 and type 1 diabetes, glaucoma, psoriasis, hypertension, asthma and Parkinson's disease (Supplementary Data 18).

### Associations with blood cell traits and markers of inflammation

After data pre-processing as detailed in Methods, data on 33 blood and inflammation biomarkers were able to be matched to RT data for 39,611 participants (Supplementary Data 19, Supplementary Fig. 34). Various reticulocyte traits were significantly associated after FDR correction with reduced retinal thickness, especially in the parafoveal area (Fig. 4G), a finding consistent with our genetic correlation analyses. Interestingly, higher counts of immune system cell types such as leukocytes (sig. pixels: 28,451, av. beta: -0.15, median log10(p): 10.45) and neutrophils (sig. pixels: 28,241, av. beta: -0.13, median log10(p): 7.40) were associated with parafoveal and temporal perifoveal thinning. Associations with inflammatory markers such as the ratio of platelet count to lymphocyte count ratio (Fig. 4H) confirmed the high sensitivity of the parafovea to inflammation, an observation previously reported for diseases such as Covid-19[56]. We did not identify any association of RT with the 70 tested infectious disease antigens (Supplementary Data 20). This was likely due to the small sample size available for this analysis ($n = 764$). All association results at the pixel level are accessible through a user-friendly bespoke web browser (https://retinomics.org/).

## Discussion

Using one of the largest, systematically-collected OCT datasets in the world (UKBB), and through the application of AI, we generated the highest-resolution spatial dataset of RT ever produced, and demonstrated that the retina provides a window to human health. Our analyses reveal RT to be intricately related to a plethora of factors, spanning the genome, metabolites, blood traits, and diseases, with the

parafoveal area most enriched for associations. All relationships identified through these analyses represent observed correlations, with further work required to disentangle the complex relationships between these facets. Overall, we found reduced RT, or retinal thinning, to be associated with poorer health, and increased burden of disease.

We employed two approaches to conduct genetic analyses using this data, firstly using dimensionality reduction, in the form of functional principal components, and secondly through our pixel-level approach, which required bespoke methods to summarise the data from over 29,000 GWAS. We note that a similar AI approach, applied to the UKBB whole body imaging data identifying genetic loci influencing skeletal proportions, some of which are risk factors for diseases of the skeleton, such as osteoarthritis[57]. These complementary approaches, applied to AI-reprocessed data identified a number of novel genetic associations, not identified in previous GWAS conducted with RT derived directly from the TOPCON scanner, despite the heavy multiple testing burden. Moreover, these results give micron-level spatial resolution, highlighting distinct spatial patterns for several loci, whereas previous analyses were averaged across regions on the ETDRS, or Macula 6 grids. Many of our novel loci have previously been implicated in Mendelian retinal diseases, or ocular traits, and our GWAS findings overall were enriched for genes involved in eye development and cell differentiation.

In some of our most important findings, reduced RT is highly associated with MS. This result provides strong, independent confirmation of multiple reports of the utility of OCT as the source of biomarkers for MS and MS progression[58], summarized in Britze et al[59]. Additionally, it indicates that future studies utilizing RT as a biomarker should focus on the nasal perifoveal region of the macula as this region contains the greatest signal.

We additionally showed that a range of neurodegenerative and cardio-metabolic disorders were highly associated with retinal thinning. We illustrated that the retina has unique metabolic sensitivities, with RT associated with multiple systemic metabolic diseases, and metabolites previously implicated in several retinal diseases[33,44,60,61]. For several of these metabolites and diseases, we supported findings, through associations both with directly measured phenotypes, and their genetically-predicted proxies. We also identify that disease genetic scores sometimes reveal association signals not shown by their clinical record counterparts (PheCodes). For many of the diseases linked to RT, through observed associations, or via genetics, further work is needed to determine whether these diseases may be causally affecting RT, or simply correlated due to a shared genetic basis, or other confounding factors.

Further understanding of our genetic results will require single cell and spatial transcriptomics of the retina to investigate specific cell types being affected, as well as determine biological mechanisms. Some of the genetic loci we describe here, and previously linked to disease, are already being investigated with these approaches[31,62]. Our work provides compelling evidence and extensive additional localising information to allow better targeting of the retina with these expensive technologies. Existing GWAS have examined thickness of retinal sublayers, averaged over the entire macula area[15,19,20], and have shown some associations to be layer specific; future work will extend our spatial approach to the retinal sub-layers, allowing us to further tease apart many of the associations and explore cell type specific effects.

Our investigations focused on RT in retinal-normal individuals. In excluding individuals with overt retinal diseases, who may have major and unusual variation in RT, our analyses were undertaken on a more homogeneous cohort, thereby maximising our power for discovery. While our analyses primarily intended to capture variation in healthy eyes, it is likely our exclusion criteria will not have captured all individuals with retinal disease, and a small minority of diseased participants will have been included. Investigation of RT in the context of

overt disease will also be of interest but will require a nuanced analysis approach which is beyond the scope of the current work.

This work has some shortcomings. The complexity of the high resolution phenotypes used here, required us to make a number of pragmatic decisions regarding our approach. Among these was the decision to impute missing data points, and in our GWAS to exclude loci with fewer than five SNPs showing association with p < 5e-5. The imputation assumed data were missing at random, however missingness could have been more likely to occur for pixels with more extreme values of RT, while the GWAS filtering may have resulted in some true associations being excluded. The UKBB is very European-centric with > 90% of participants being of European ancestries. Our cross-ancestry genetic association analyses suggested significant genetic heterogeneity in RT; however the number of individuals with CSA and AFR ancestries included in these analyses were relatively modest, and we were therefore underpowered to comprehensively explore differences across ancestries. Further investigation in individuals with greater ancestral diversity is required and emerging efforts to increase diversity in population cohorts will hopefully allow this. The UKBB is a uniquely massive resource and as such, replication of our multi-omics findings was not feasible. However, we strove to validate many of the main findings through complementary approaches, such as utilising genetic scores alongside directly measured phenotypes and employing co-localisation approaches.

This paper developed a hypothesis-generating resource and describes a series of exploratory analyses. While we have attempted to account for major confounders (sex, age, height, spherical equivalent, and genetic ancestry) in all our analyses, our models will inevitably not have accounted for all confounding factors, and some of our association results will have been impacted by bias. As in traditional, observational epidemiology, genetic associations may be subject to confounding, in particular by socioeconomic factors[63,64]. Bias may have also been introduced through our phenotyping approach, for example if the accuracy of foveal alignment was impacted by some non-random factor, like eye size. Future studies, which more comprehensively account for confounders and biases through tailored models, specific to the disease/trait being examined, are needed to fully characterize all reported associations. We also note that whilst the UKBB was intended to be representative of the ageing UK population, it is subject to a range of selection biases[65]. All described associations thus must be interpreted in this context.

In summary, we conducted multi-dimensional GWAS, Omics and clinical data studies, supported by innovative statistical methods for imaging and GWAS at scale. Our findings, which can only be fully appreciated through the bespoke web-browser that accompanies this work, aim to direct future biomarker studies and biological experiments. This work validated and refined previous associations and will encourage others to reprocess high-dimensional data with AI.

## Methods

### Data and data access sharing
Data was sourced from the UK Biobank using project applications 28541. Research was approved by the Walter and Eliza Hall Institute of Medical Research Human Ethics Committee (HREC 17/09LR).

### OCT scans in UK biobank
A subset of 85,726 UK Biobank individuals had retinal OCT data available. Scans were performed with the TOPCON 3D OCT 1000 Mk2. An OCT scan consists of a volume of 128 B-scans, each composed of 512 A-scans. Scans were obtained on up to two occasions: "instance 0" (baseline); or "instance 1" (first follow-up visit). The UK Biobank protocol states that one scan per eye should be recorded at each instance; however, a subset of individuals had two scans for one or more eyes at a single instance, while some had only one eye measured at an instance. Initial assessment visits were conducted during 2006-2010,

with a total of 68,530 participants. First repeat assessment visits were performed during 2012-2013, with a total of 19,579 participants. 2316 individuals had measures at both instances (Data-Fields 6070 and 6072 in UK Biobank). At these visits, other relevant information collected included refractive error, intraocular pressure, visual acuity and eye checks with questions regarding any surgery that had been performed. In addition imaging device number was also recorded, which permitted technical error correction.

### Generation of macular retinal thickness data using machine learning
First, we developed an ML pipeline to extract the area between the internal limiting membrane (ILM) and retinal pigment epithelium (RPE) from OCT B-scans and to detect the layer boundaries without need for human annotations. The Topcon 1000 produces a 128 raster OCT volume over a 6 mm*6 mm*2.275 mm area around the macula. The pixel resolution of each OCT B-scan is 512*650, with the 650 pixels corresponding to the anterior to posterior direction. We measured the ILM-RPE thickness at every other pixel, i.e., 256 locations. The ML pipeline consisted of two steps. In the first step, we used the A star (A*) algorithm[66] to obtain layer boundaries by following the bright bands of the ILM and the RPE on OCT B-scans. These bright bands were then turned into segmentation masks for the ILM and RPE. A* is widely used in pathfinding and can be used to track retina layer boundaries in this use case. However, A* is not robust to poor contrast or abnormal brightness and random noise, and quality control on A* predicted segmentation was performed. Specifically, A* was applied to 1 random B-scan from 8500 FDA data. Then only masks whose ILM-RPE thickness were between 30 and 80 pixels, which translates to 180 to 480 μm at an axial resolution of 6 μm/pixel) and SD < =10 pixels (60 μm) were accepted. This resulted in 6409 after the thickness restriction. Next, another 1250 FDA files were sampled and 1 random B-scan from them predicted using A*; these masks were filtered for minimal ILM-RPE thickness as above. Then we computed the sum of entropy for each segmentation mask where each pixel had a probability value of being foreground. We regarded examples with high entropy value as hard examples, which were sorted in descending order, and the top 200 images by entropy were added to the 6409 B-scans for training the deep learning model. Examples of A-star segmentations that were accepted and rejected are provided in Supplementary Fig 1, which shows three pairs of images with the raw B-scan on the left and the A-star segmentation on the right. The second and third segmentations in this figure were rejected as they were not fully correct. These reviewed B-scans and A-star segmentation masks were used as training data for the second part of the pipeline, the deep learning model called PSPNet. PSPNet was a state-of-the-art segmentation model that incorporated a pyramidal structure to allow segmentation at varying image and resolution scales. Our PSPNet achieved an intersection over union (IOU) of 0.97 on the test set. IOU is computed as $TP/(TP + FP + FN)$, where TP is True Positive pixels where the ground truth and PSPNet prediction agree, FP is the False Positive pixels and FN is the false negative pixels where ground truth and PSPNet disagree. As IOU incorporates true positives and false discoveries, it is a very conservative score, even compared to the Dice coefficient to which it has direct functional equivalence, and 0.97 an extremely high IOU. An ablation study was performed on PSPNet hyperparameters, but there was little difference in model IOU performance of 0.97. In addition, the popular UNet model was also used, but underperformed compared to the PSPNet.

Next, we performed multiple rounds of quality control on the ML pipeline's predicted segmentation masks. Quality control was performed at the location level with the location thickness rejected, and removed, if it failed any of the following six criteria. First, the thickness measurements from B-scans that were too faint to contain any ILM-RPE layers were removed (Supplementary Fig. 2a). Second, thickness

measurements for OCT scan regions that were too faint were removed (Supplementary Fig. 2b). Third, locations where the location thickness was too thin (less than 30 pixels or 105μm) or too thick (more than 165 pixels or 577.5μm) were removed (Supplementary Figs. 3a and 3b); for reference the typical thickness range is 154-232μm for males and 173-252μm for females. Fourth, regions where the predicted ILM or RPE boundaries were directly adjacent to regions that are too faint were removed as this suggests the OCT scan was not centred properly and was cut off (Supplementary Fig. 4). Fifth, locations where the location thickness was discontinuous, i.e., where either the predicted ILM or the predicted RPE coordinates were discontinuous (Supplementary Fig. 5). Sixth, the location thickness was rejected if the standard deviation of a rolling window of the 10 locations enclosing the location was very high (20 pixels or 70μm), i.e., highly unusual disturbances in the ILM-RPE layer (Supplementary Fig. 6).

After the quality control, each OCT volume, containing 128*256 locations, was aligned to the fovea point. The fovea point was determined to be the centre of the area with the thinnest retina for each OCT volume scan and its location given as a tuple of slice number (between 1-128) and B-scan x-coordinate (between 1-256) giving a total of 32,768 measure locations for RT per eye. Moreover, the ILM-RPE thickness measurements for OCT scans from the left eye was reflected across the fovea point, so that it was aligned anatomically to the thickness measurements from OCT scans from right eyes.

Hereafter, locations are referred to as pixels.

## Secondary quality control filtering of aligned data
Additional filtering was conducted at the individual, OCT scan and pixel level for quality control. The overall filtering strategy is summarised in Supplementary Fig. 9.

The focus of these analyses were on healthy retina so individuals with overt retinal disease, or likely retinal disease were discarded (see Supplementary Table 1). Individuals with self-reported or mild, or non-retinal vision problems were retained, striking a balance between loss of power due to the removal of many participants and exclusion of some potential eye disease participants still being retained. Individuals with mild or non-retinal vision problems that were retained included those with: cataracts; reported as wearing glasses; myopia; astigmatism; presbyopia; hypermetropia. Additionally, participants were not excluded based on field 2227: "Do you have any other problems with your eyes or eyesight?".

OCT scans were also removed based on Machine Learning model learnings. Quality control experiments as part of the machine learning modelling of the RT (described above) revealed diseased retinas (see examples in Supplementary Fig. 7) for several individuals with no self-reported retina pathologies.

Scans were then trimmed to remove pixels with high levels of missingness or potentially spurious measurements. Firstly, pixels that were missing in > 10% scans were removed. Then, entire rows or columns were removed if > 50% of pixels had been removed based on the 10% missing threshold. This resulted in the removal of 3727 pixels (11%) and resulted in a trimmed grid of 29,041 pixels.

## Selection of scans
Using the trimmed grid, the best scan per eye, per instance, was selected for each individual. Firstly, scans were excluded if > 10% of pixels in the trimmed scan were missing, or if refractive error for the corresponding eye and instance was unavailable. If there remained more than one scan available for an eye at an instance, the scan with the lowest missingness was retained, or in the case of a tie, one scan was randomly selected.

If an individual had scan data for both left and right eyes from the same instance, both eyes were selected. For the subset of individuals who had both eyes available at both instances, the instance with fewest missing data points (across both eyes) was identified, and the scans

taken from that instance. For individuals who did not have a scan for both eyes measured at the same instance, the single eye with fewest missing data points was used (either eye/instance). In the case of ties, one eye was randomly selected.

## Imputation of missing values per scan
Imputation of scans was then carried out. For each scan, generalized additive model (GAM) was fitted using the "bam" function of mgcv R package (version 1.8-40) with the following model:

$$Z_i = \beta_0 + f(x_i, y_i) + \epsilon_i$$

where $Z_i$ is the depth measurement at position $(x_i, y_i)$ and $f(x_i, y_i)$ is a tensor product smooth function (k = 12, 12). Where a pixel had missing data in an individual scan, the value was imputed using the predicted value from the GAM, assuming that missingness was at random.

## Generation of final phenotypes
Finally, the pixel-level retinal thickness phenotypes were generated using the imputed scans. If an individual had both left and right eyes available, the mean value for each pixel was taken, otherwise, the single available eye was used. A final filtering step removed individuals where their final scan phenotype had any pixels where thickness measurements were < 30 or > 150, or where the scan-wise standard deviation was > 15.

Using the final pixel-level retinal thickness phenotypes, functional principal component analysis (FPCA) was undertaken using the MFPCA R package (version 1.3-10)[23], and using penalised 2D tensor product splines, with k = (12,12) as the maximum degrees of freedom. 100 FPCs were estimated, with the top six FPCs explaining > 95% of the variance selected for follow-up analyses. For each individual, FPC scores were extracted, representing the contribution of each FPC to that individual's scan phenotype.

## Quality control based on genetic data
The genotyping, quality control and imputation of the UK Biobank cohort was undertaken by UK Biobank[25]. Imputed genotypes to the combined Haplotype Reference Consortium and UK10K panel (version 3; Category 100319) and genotype quality control metric files (Category 100313) were downloaded for the full UK Biobank cohort. Individuals were excluded if: they had been excluded by UK Biobank before imputation, due to high heterozygosity or missingness ( > 5%); there was a mismatch between their genetically predicted and recorded sex, or they had a sex chromosome aneuploidy; they had an apparent excess number of relatives in the UK Biobank cohort ( ≥ 10 relatives); or they had withdrawn consent. Broad continental ancestries for all individuals were obtained from UK Biobank returned dataset 2442, and were based on genetic similarity to individuals from the 1000 Genomes Project and Human Genome Diversity Panel, as generated through the Pan-UK Biobank project (https://pan.ukbb.broadinstitute.org/). Individuals of European (EUR), Central and South Asian (CSA) and African (AFR) ancestries were identified for genetic analyses. The remaining individuals were excluded as their continental ancestry groups had < 1000 individuals in each.

The full set of EUR, CSA and AFR individuals were then restricted to a maximally unrelated set. Pairwise relationships, up to 3rd degree were identified, based on the kinship coefficient estimates from KING[67], as made available through the UK Biobank genetic data release (3rd degree relatives defined as kinship > 0.088). Any individual with multiple ( > 1) relatives in the set was identified and removed. Remaining relative pairs were identified (each of whom were related to one individual in the set only). One of each pair was then randomly selected for exclusion.

Variants with minor allele frequency (MAF) < 0.05%, or imputation INFO scores < 0.8 were excluded. QCTool v2 was used for sample and

variant based filtering, then bgen files were converted to PLINK binary format, using PLINK 2 (version 20221024).

## Genome-wide association analyses

Genome-wide associations were undertaken for the pixel-level retinal thickness phenotypes, and the FPC score phenotypes, using the glm command in PLINK 2 (version 20221024). Covariates included in the model were sex, age, age-squared, standing height, device ID, eye (left, right, or mean), spherical equivalent (for the corresponding eye, or the mean of left and right, if the mean of both eyes was used for the phenotype) and the first ten ancestry principal components as provided by the UK Biobank (field number 22009). Quantitative covariates were standardised to have mean 0 and variance 1, and the variance inflation factor upper bound flag was set to 500, to allow for the multicollinearity of age and age squared. Chromosome X was analysed by coding hemizygous males as homozygous.

For all GWAS, inflation was assessed using the LD-score (LDSC) regression intercept, as implemented by LDSC v.1.0.1[26]. Heritability estimates were also calculated using stratified LDSC[68], with from SNPs in 5 MAF and 5 linkage disequilibrium (LD) score bins, generated for EUR ancestries individuals as part of the Pan-UK Biobank project (https://pan-ukb-us-east-1.s3.amazonaws.com/ld_release/UKBB.ALL.ldscore.tar.gz).

## Identification of independent significant loci

Identification of independent significant loci was carried out separately for the FPC analyses (6 GWAS) and the pixel-level analyses (29,041 GWAS).

Linkage disequilibrium clumping was undertaken for each set of GWAS results, using PLINK v1.90 to identify statistically independent signals, using strict criteria: index SNPs with $p$-value < 5e-8 identified, with clump SNPs having $p$-value < 5e-5, and $r^2 > 0.001$ and < 5000 kb distance to the index SNP. SNPs were allowed to belong to more than one clump. LD clumps were collated to give a list comprising all genome-wide significant ($p < 5e-8$) index SNPs, and the corresponding clump SNPs. The results for all N SNPs in this list were then extracted from each GWAS ($N_{pixel-levelSNPs}$ x 29,041 associations for the pixel-level analyses; $N_{FPCSNPs}$ x 6 associations for the FPC analyses).

Identification of loci was then carried out, using the extracted associations, and the clumping results, here, described in the context of the pixel-level results:

While at least one pixel-SNP association has $p$-value < 5e-8:
1. Order all extracted associations by $p$-value.
2. Take the pixel-SNP combination with smallest association $p$-value.
3. If this SNP is an index SNP in the clumping results for this pixel, then:
   a. Define a new locus with the index SNP as the sentinel SNP and the pixel as the top pixel for the locus.
   b. Find all corresponding clump SNPs ($p < 5e-5$) for that locus, from the clumping results; these SNPs are allocated to the locus as supporting SNPs.
   c. Find all pixels with $p < 5e-5$ for either the sentinel, or any of the supporting SNPs; these pixels are allocated to the locus.
   d. Remove all allocated SNP-pixel combinations from the list of extracted associations.
4. Else, if this SNP is not an index SNP in the clumping results for this pixel, then
   a. Find the clump where the SNP is included as a clump SNP.
   b. Extract the index SNP for this clump, and confirm this index SNP has already been assigned to a locus.
   c. Remove any remaining clump SNPs from the list of extracted associations.
5. Repeat steps 1-4 until no pixel-SNP associations with $p$-value < 5e-8 remain.

Loci were identified from the 6 FPC GWAS, using an equivalent approach.

The above process resulted in two lists of loci: one for the pixel-level analyses, containing loci with at least one pixel-SNP association meeting $p < 5e-8$; one for the FPC analyses, containing loci with at least one FPC-SNP association meeting $p < 5e-8$. Finally, these lists were filtered, to remove loci with fewer than 5 SNPs assigned (ie fewer than 5 SNPs in the region with $p < 5e-5$), as these were deemed to be likely false positives.

The loci identification process utilised the conventional genome-wide significance level (P < 5e-8), however, this does not account for the heavy multiple testing burden of our approaches. The final reporting, and subsequent follow-up of loci were therefore restricted to SNP associations, meeting conservative Bonferroni corrected thresholds: $p < 5e-8$ / 29,041 = 1.72e-12 for the pixel-level analyses; $p < 5e-8$ / 6 = 8.33e-9 for the FPC analyses.

## Identification of secondary genetic association signals

For all loci meeting the Bonferroni-corrected significance level outlined above, we sought to identify whether there were independent, secondary SNP association signals in the region. For the top pixel/FPC for each locus, we first ran cojo-select, in GCTA v.1.94.1[69], using $p$-value thresholds of $p < 1.72e-12$ for the pixel-level analyses and $p < 8.33e-9$ for the FPC analyses. If any SNPs additional to the sentinel were identified through the cojo analyses, we ran conditional analyses using PLINK2 for that region with the sentinel SNP as a covariate, to determine true independent signals. We report any SNP meeting $p < 1.72e-12$ (pixel-level) or $p < 8.33e-9$ for (FPCs) in the conditional analyses as a secondary SNP; these are listed in the "SNP_Secondary" columns in Supplementary Data 1 and 2.

## Annotation and follow-up of GWAS loci

Sentinel, and secondary SNPs were annotated using the variant effect predictor (VEP v.109[70]). Loci were also annotated to indicate whether they were identified through previous GWAS of retinal thickness[15,18–20]. For these annotations, the sentinel SNP along with all other SNPs allocated to each locus were compared to previously reported SNPs. If no SNP allocated to a locus was included amongst previously reported SNPs, that locus was deemed novel.

Follow-up was undertaken for loci meeting the strict Bonferroni corrected thresholds described above. Initial mapping of loci to candidate genes via positional mapping, eQTL data, and chromatin interaction data, was undertaken based on annotations obtained via FUMA (v1.5.3)[71]. For each set of loci (pixel-level associated loci; FPC associated loci), the list of sentinel SNPs was uploaded as a predefined list of lead-SNPs, with mapping of SNPs to genes carried out using the uploaded sentinels, and all SNPs in LD with r2 > 0.5 with the sentinels (collectively referred to herein as candidate SNPs).

Positional mapping identified genes overlapping with, or < = 5 kb from, candidate SNPs using ANNOVAR (2017-07-17). eQTL mapping identified genes (cis, up to 1 Mb) whose expression are associated (FDR < 0.05) with candidate SNPs, in retina (EyeGEx)[11], blood (GTEx v7 tissues: EBV-transformed lymphocytes; whole blood) or brain (GTEx v7 tissues: amygdala; anterior cingulate cortex BA24; caudate basal ganglia; cerebellar hemisphere; cerebellum; cortex; frontal cortex; hippocampus; brain hypothalamus; nucleus accumbens basal ganglia; putamen basal ganglia; Spinal cord cervical c-1; brain substantia nigra)[28]. For candidate eQTL-mapped SNPs identified by FUMA, Bayesian colocalisation analysis was performed using coloc (version 5.2.3)[72] with SuSiE (version 0.12.4)[73], allowing for multiple causal variants. Summary statistics for each SNP's top pixel/FPC were colocalised with GTEX v7 (https://gtexportal.org/home/downloads/adult-gtex/qtl) and EyeGEx eQTL data using default parameters. Results were restricted to those with mapped eQTL P < 5e-06 and a posterior probability of a shared causal variant (H4) > 0.8.

Chromatin interaction mapping by FUMA used HiC data from fetal and adult human brain[30], to link candidate SNPs to genes based on overlapping enhancer-promoter and promoter-promoter interaction regions (restricted to significant interactions with P < 2.31e-11). Chromatin interaction mapping was also performed with pre-processed HiC data for retina (https://www.ncbi.nlm.nih.gov/geo/query/acc.cgi?acc = GSE202471). Both directions were considered and restricted to interactions with FDR < 0.01.

The candidate genes identified through these mappings, were additionally annotated as to whether they were associated with eye or retina-related phenotypes in mice or humans. For mouse phenotypes, we utilised a set of genes from the International Mouse Phenotyping consortium (https://www.mousephenotype.org accessed May 2nd 2023), causing "abnormal eye morphology" (MP:0002092), and with human orthologs[39]. To identify relevant genes in humans, we queried the Online Mendelian Inheritance in Man (OMIM) database (https://omim.org/downloads/ accessed May 2nd 2023) for genes with "retina" or "retinal" included in the entry, and clinical synopsis including "Head & neck"[74].

For completeness, we report all genes implicated via the above mappings, or annotations (Supplementary Data 6). For certain follow-up analyses, we further prioritised candidate genes, based on the number of lines of evidence for that gene (prioritisations specified with the relevant analyses). To this end, we created a score, which summed up whether the gene was: i) implicated via positional mapping; ii) implicated via mapping with an exonic candidate SNP; iii) implicated via mapping with a candidate SNP with a CADD score > = 20; iv) implicated via retina eQTL data; v) implicated via blood eQTL data; vi) implicated via brain eQTL data; vii) implicated via HiC data from brain; viii) implicated via HiC data from retina; ix) associated with eye abnormalities in mice; x) associated with a retina OMIM phenotype. This score is reported in the "evidenceScore" column of Supplementary Data 6.

## Clustering of sentinel SNP effects
We performed unsupervised hierarchical clustering analysis on both pixels and SNPs, based on the effect sizes for the sentinel SNPs, across all 29,041 pixels. The number of clusters was determined from visual inspection of the clustering dendrogram.

## Associations in non-European ancestries
We undertook association analyses for all identified loci, in individuals from UK Biobank AFR and CSA ancestries. For each locus, we examined associations with the sentinel SNP at each identified locus with all 29,041 pixels (pixel-level analyses), or for the FPC(s) for which the SNP was significant. Association analyses were undertaken using PLINK 2 (version 20221024), similarly to the analyses of European individuals. For each locus we report the results for the pixel, or FPC, with the most significant association observed in European individuals. We examine concordance of direction of effect across ancestries. We deemed associations in AFR and CSA individuals to be statistically significant, if they met the Bonferroni corrected p-value threshold for the number of loci tested. We additionally compared whether the pattern of association across the entire scan was similar across ancestries. For each SNP identified in the pixel-level analysis, we calculated the the correlation of effects across all pixels, using Spearman's Rank Correlation.

## PheWAS and Genetic correlation analyses
Phenome wide association studies (PheWAS) were performed on sentinel and secondary SNPs with curated publicly available data through the openGWAS platform (https://gwas.mrcieu.ac.uk/), via the R-package ieugwasr (version 1.0.0). SNPs were queried against batches: EBI database of complete GWAS summary data (ebi-a), GWAS summary datasets generated by many different consortia initially developed for MR-Base rounds 1 and 2 (ieu-a, ieu-b), pan-ancestry genetic

analysis of the UK Biobank (ukb-e) and an expanded set of genome-wide association studies of brain imaging phenotypes in UK Biobank (ubm-b), filtered for significance threshold P < 1e-05.

Genome-wide genetic correlation analyses were carried out for FPCs 1-6, using LD-score regression[75], implemented via the Complex Traits Genetics Virtual Lab[76]. We examined correlations with 1309 traits in total (Supplementary Data 11).

## Gene set over-representation analysis
Gene set over-representation analysis (ORA) for Gene Ontology Terms was performed using the R-package *ClusterProfiler (v. 4.8.3)*[77]. ORA was carried out for genes implicated via the pixel-level and FPC results separately, using two sets of genes in each case: (i) all candidate genes (ii) the gene(s) with the most lines of evidence (i.e., highest "evidenceScore") for each SNP. Correction for multiple testing was performed using Benjamini-Hochberg ad-hoc correction, and we report all GO terms with corrected p-value < 0.05.

## Sensitivity analyses
We undertook sensitivity analyses on the subset of 42,970 individuals, for whom smoking status could be derived (UK Biobank Field 20116: 3666 current smokers; 15,228 previous smokers; 24,076 never smokers. Genetic associations for the reported sentinel SNPs were repeated for the top pixel and/or FPC, with smoking status included as an additional covariate. The effect estimates and p-values were then compared for analyses with and without adjustment for smoking.

Given the distribution of a subset of the pixel-level RT phenotypes showed some deviation from normality (Supplementary Fig. 35 and 36), we also undertook sensitivity analyses by repeating top SNP-pixel associations using rank inverse normal transformed phenotypes. Comparison of effect sizes and p-values based on associations of raw RT phenotypes versus transformed phenotypes showed high consistency (Supplementary Fig. 37), thus we only report the untransformed results.

## Other 'omics and clinical data analyses
We next assessed the associations with metabolomics, blood and immune trait markers, and disease codes (PheCodes) on RT. For metabolite and disease codes, we examined associations with both direct measures, and using genetic risk scores. These analyses were restricted to EUR individuals, due to limited sample sizes for other ancestries. In the following sections, we outline the cleaning of each data type, then the analyses undertaken, where the same protocol was utilised for each data-type.

## Genetic scores preparation
Polygenic Risk Scores (PRS) and Genetic Risk Scores (GRS) were used to investigate the relationship between genetic susceptibility to a number of diseases and traits and RT. It is worth noting that prediction power of PRS and GRS may vary substantially as the variant weights of the PRS were constructed to optimise prediction, while the variant weights of the GRS were derived from standard GWAS summary statistics.

We used three types of genetic scores in this study. Type 1 scores comprise *trait PRS* and *GRS* composed mainly of scores describing disease susceptibility. The *trait PRS* are further subdivided into three subclasses. Two of these, the *Standard* and *Enhanced PRSs*, were downloaded from the UK Biobank having been generated and deposited in the UKBiobank as described by Thompson and colleagues[78] (Categories 301 and 302). These PRS had already undergone extensive QC and data processing and thus were used, as is, in the association analyses.

Type 2 scores comprise *metabolic GRS* measuring genetic predisposition to blood abundance of a set of metabolites, measured with the Biocrates platform[48]. The third type of scores are defined as

*Internal GRS (*Type 3 GRS). These were GRS of particular relevance to the RT phenotype and included two macular disorders and two measures related to the RT phenotype. This included GRS for: (i) Macular Telangiectasia type 2[33], (ii) Age-related macular degeneration[55], (iii) Retinal thickness[18], and (iv) retinal venular and arteriolar calibre[32,53]. To construct the type 2 and type 3 GRS, we extracted the top SNPs at each genome-wide significant locus associated with the trait as reported by the authors. For type 2 *metabolic GRS*, we discarded all metabolites that had fewer than three top SNPs available to construct the *GRS*. We then used the R-package *bigSNPR (version 1.12.2)*[79] to extract the selected SNPs from UK Biobank and create the scores.

### Blood traits data preparation
Blood trait data (Category 100081) was downloaded from UK Biobank. Blood trait data was collected using Beckman Coulter LH750 instruments for all 500,000 participants of the UK Biobank at the baseline visit. Cleaning of the blood traits was undertaken, based on the methods employed in ref. 80. In brief, all blood traits were first log, or logit (in the case of ratios or proportions) transformed. Using a restricted set of central measurements (measures < 3.5 median absolute deviations from the median), a generalised additive model (GAM, using the mgcv version 1.8-40 R package) was fitted, to model the effects of several technical covariates (time of measurement, instrument, acquisition route, day of the week). In the full dataset, model residuals were calculated and used to generate traits adjusted for technical effects. Finally, outliers ( > 6 median absolute deviations from the median) were excluded. The log/logit transformed, adjusted traits were used for downstream analyses.

Following the work from Nøst and colleagues[81] we used this data to calculate systemic inflammation markers. Specifically, we used peripheral lymphocyte, neutrophil, monocyte and platelet counts to calculate neutrophil-to-lymphocyte ratio (NLR), platelet- to-lymphocyte ratio (PLR), lymphocyte-to-monocyte ratio (LMR) and the systemic immune-inflammation index (SII). These were analysed in the same manner as the directly measured blood traits.

### Infectious disease antigens preparation
Infectious disease antigen data (Category 1307) was downloaded from UK Biobank. Antigens were measured using the Luminex platform. Only samples whose antigens were measured were kept in the dataset. Antigen measurements were log-transformed prior to use.

### Metabolomics data preparation
Measurements of metabolic data from non-fasting venous blood of around -120,000 UK Biobank participants[82] were downloaded (Category 220). Metabolic quantifications included a total of 249 metabolic biomarkers. Of these, 168 were directly measured and quantified by NMR spectroscopy using the Nightingale platform, while 81 biomarkers were ratios or derivative measurements. The metabolomics data was processed with the R package *ukbnmr (version 1.5)*[83]. This procedure removed technical variation as well as normalising the metabolic values. Non-derived metabolites were then square-root transformed to achieve symmetry of the distribution. Assuming missingness at random, metabolic values were imputed using the Multiple Imputation Chain Equation by the R-package *MICE (version 3.15.0)*[84]. Five different imputations of the entire data were generated and the average of these five imputation set values was retained as the final set of imputation values. Composite metabolites and ratios were then re-computed using *ukbnmr*. A total number of 325 metabolites were included in this study.

### Phecodes data preparation
To extract ICD10 and ICD9 diagnoses for each individual we used the R-package *ukbtool (version 0.11.3)*[85]. This data was then transformed into binary format where each patient was represented either by 0 or 1

for each ICD10 or ICD9 code depending on whether they were unaffected or affected. From the PheWAS catalogue (https://phewascatalog.org/) we downloaded the PheCode Map v 1.2 for both ICD9 and ICD10 and matched each code to the respective PheCodes. PheCodes categories were also provided by the PheWAS catalogue (https://phewascatalog.org/files/phecode_definitions1.2.csv.zip).

### Omics and clinical data analyses
Genetic scores, metabolomics, PheCodes, blood traits and infection antigens ("factors of interest") were tested against retinal thickness using the same protocol. Firstly, all data from participants for whom no OCT imaging data were available were discarded. Secondly, to allow for comparison of effect sizes across biomarkers/phenomes, each of them was standardised to have mean equal to zero and standard deviation equal to one. Thirdly, association testing was performed using the R package *limma (version 3.50.3)*[86] as follows: The "assay matrix" containing the retinal thickness measurements, was constructed as a row for each pixel analysed and a column for each subject. The model matrix was composed of columns containing the factor of interest, the intercept, and a column for each of the covariates: (i) genetically derived sex at birth, (ii) age, (iii) imaging device number, (iv) standing height, (v) mean refractive error measured by spherical equivalent, (vi) eye and (vii) the first ten genetic principal components. Participant data presenting with missing information for any of these covariates was discarded. Regression analysis was performed by building separate independent least-squares regression models for each pixel thickness, using the *lmFit* command, with the assay and model matrices as input. The function *eBayes* was then used to borrow information between pixels about the variability of the data and apply the empirical Bayes statistics to the results from the linear models, hence increasing power. This process was repeated for each factor in each data axis analysed. Fourthly, as there is no widely approved method for multiple testing correction when dealing with such a multidimensional omics study, correction for the 'Omics and clinical data was performed using three different methods using the ad-hoc R-function *p.adjust*. All *p*-values within each omic/clinical data type were corrected (first) jointly using a Bonferroni correction, (secondly) using Benjamini-Hochberg correction for false discovery rate and (thirdly) using the Benjamini-Hochberg correction for false discovery rate across all pixels but only within each factor of interest. The results in the manuscript are provided using the second approach. For comparison, supplementary data tables present significant pixels associated to each factor of interest using all three methods.

The same strategy as described above was used to test each association of each data-type with the first six FPCs. Given that the FPCs capture specific grid-wide patterns, a more lenient FDR threshold of 1% was used to define significance of this analysis.

The effect of BMI on the association between retinal thickness and metabolite levels was tested via the inclusion of BMI in the regression model. The correlation between the effect size of each metabolite for each pixel across the corrected and non-corrected model was estimated and visualised for each metabolic class. The same approach was used to test for the effect of smoking status on the association between retinal thickness and PheCodes.

Interaction effects between metabolic levels and age on retinal thickness were evaluated following the same approach as that described above, but by adding an interaction term for metabolic level and age, in addition to the main effects, in the limma linear regression model. To test whether the effect of the metabolites on thickness varied with age, we counted the proportion of significant interaction terms where the sign is equal to the sign of the main effect of the metabolite on retinal thickness.

Correlations between the estimated effects from the PheCode-RT pixel-level associations, and the corresponding effect estimates from

the genetic score-RT associations, were examined by calculating the Pearson's correlation coefficient across all pixels. For each pair, three analyses were performed, one using every pixel, one using only those pixels where the genetic score had a significant effect, and one using only those pixels where both the genetic score and PheCode had significant effects.

For plotting and interpretation purposes, we summarised each factor of interest's effect on global retinal thickness using a suite of measures. These were: (i) number of pixels significantly associated with each factor, (ii) the average beta of the factor across significantly associated pixels (iii) the average beta of the factor across all pixels, (iv) the median -log10(p-value) of the fatcor association across all pixels. Values for all of these measures are available as columns in each of the associated result tables, which are specific for the eight types of data being tested (Supplementary Data 13–20). Each results table also has data type-specific descriptors in the columns preceding these measures.

We performed hierarchical clustering analysis on both pixels and factors of interest for the metabolomics and blood traits analyses. These omics were selected for this analysis due to their large number of variables and prior knowledge of the existence of structured correlation patterns. Clustering of metabolites and blood traits was performed to capture shared (spatial) effect patterns on pixel-level RT. Clustering was performed using unsupervised hierarchical clustering. The number of clusters was determined visually based on the hierarchical clustering tree produced by the analysis.

Pixel-level over-representation analysis (ORA) was performed by using the R-package ClusterProfiler (*version 4.2.2*)[77]. Each pixel $i$ was defined as a set and each test pixel $i$-marker was defined as an entry of the set. Over-representation of significant pixel-factor association between pixels was then evaluated. ORA was only performed for the metabolomics biomarker main effect results, the metabolite-age interaction results and the PheCodes. For PheCodes ORA was performed on a random grid of pixels due to the computational burden. Plotting of ORA results was performed by smoothing log10(p-values) over the 2D grid using the loess function.

### Display of association results with RT (pixel level)

All association results are displayed using the same orientation for results as per Supplementary Fig. 8.

Top = superior, towards top of head

Bottom = inferior, towards bottom of head

Left hand side = temporal, towards the ears

Right hand side = nasal, towards the nose

Results are displayed either as p-values (adjusted or unadjusted), which may be transformed with a -log10 transform, or as betas (effect estimates).

The Early Treatment Diabetic Retinopathy (ETDRS) grid[87], which segments the macula into nine sectors and is widely used, is overlaid on most images to provide further spatial context for the interpretation. This grid was added by drawing the grid coordinates onto the each plot and thus represents an approximate mapping. (Supplementary Fig. 8)

### Web browser

An interactive website, https://retinomics.org, was developed to help visualize the GWAS and metabolomics results in the paper. The website has interactive pages for each type of factor of interest, such as GWAS, metabolites, hematology, infections, genetic scores and PheCodes. For each type of factor, there are two views. First, there are heatmap plots to show the effect and p-value of selected factor of interest. Second, there are macula location plots that shows the link between all significant factors for that macula location. The location plots are manhattan/region plots for the GWAS and volcano and bar

plots for the other types of factors. Users can interact with the plots by clicking or selecting a biomarker of interest to trigger the loading of its heatmaps or click a macula location in the heatmaps to load its corresponding location plots. The website was implemented in HTML, native Javascript, and d3js.

All results are displayed with the same orientation on the pixel grid as described in Supplementary Fig. 8.

### Reporting summary

Further information on research design is available in the Nature Portfolio Reporting Summary linked to this article.

## Data availability

The GWAS summary statistics for the six FPCs, and pixel-level results for all 'omics and disease code analyses, have been made available to download from https://doi.org/10.17605/OSF.IO/KZUGV. Summary statistics for the six FPC GWAS have also been uploaded to the GWAS catalogue, under accession numbers GCST90455721- GCST90455726. Due to the size of the full pixel-level GWAS results, these have not been uploaded to a public repository; however, these data will be made available to researchers on request, by contacting the corresponding author. Look-ups for pixel-level results, including SNPs, and all 'omics data, can be made using our online portal (https://retinomics.org/).

All data used in this study can be accessed from the UK Biobank upon request, via an application process (https://www.ukbiobank.ac. uk/). RT pixel-level and the six 2D FPC data will be returned to the UK Biobank.

A number of publicly available datasets, or databases were additionally used by this work: LD score bins, generated for EUR ancestries individuals as part of the Pan-UK Biobank project (https://pan-ukb-us-east-1.s3.amazonaws.com/ld_release/UKBB.ALL.ldscore.tar.gz); eQTL data from GTEX v7 (https://gtexportal.org/home/downloads/adult-gtex/qtl); HiC data for retina (https://www.ncbi.nlm.nih.gov/geo/query/acc.cgi?acc=GSE202471); the International Mouse Phenotyping consortium (https://www.mousephenotype.org); the Online Mendelian Inheritance in Man (OMIM) database (https://omim.org/downloads/).

## Code availability

We used publicly available open-source software for these analyses. Scripts for the genetic analyses can be found at https://doi.org/10. 5281/zenodo.14202924. Code and data for the retinomics.org can be found at https://doi.org/10.5281/zenodo.14199389.

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

## Acknowledgements

This work was supported by funding from the Australian Government National Health and Medical Research Council (NHMRC) from APP1195236 (M.B.) and APP1181010 (B.R.E.A., S.F.). This work was also made possible through the Victorian State Government Operational Infrastructure Support and NHMRC Independent Research Institute Infrastructure Support Scheme (IRIISS). We also thank the Lowy Medical Research Institute (LMRI) for laboratory support to A.Y.L. and M.B. and institutional support to C.Y.E.. C.Y.E. and A.T.l received a proportion of their financial support from the UK Department of Health through an award made by the National Institute for Health Research to Moorfields Eye Hospital NHS Foundation Trust and UCL Institute of Ophthalmology for a Biomedical Research Centre for Ophthalmology. A.T. and C.Y.E. acknowledge financial support from the Said Foundation.

## Author contributions

M.B. and A.Y.L. conceptualised the study and co-led the project. C.E., A.T., and K.W. shared clinical expertise to remove retinal disease individuals and other confounders from the UKBB cohort used in the primary analysis. A.Y.L. led the development of the CNN for the OCT data. Y.W., supported by J.O. and Y.K., developed the CNN. Y.W. developed the webportal with feedback from R.B. and overall direction from A.Y. and M.B.. V.E.J. supported by L.W.S. conducted GWAS analyses and follow-up. R.B. conducted multi-omics association analyses. V.E.J, R.B. and L.W.S. generated figures and results. B.R.E.A. provided feedback on analyses and figures. S.F. developed early analyses for the genomics data. M.L.G. contributed interpretation of the metabolomics results. M.B.

drafted the manuscript and led the 'omics analyses. All authors read and provided feedback on the manuscript.

## Competing interests

A.Y.L. reports grants from Santen, personal fees from Genentech, personal fees from US FDA, personal fees from Johnson and Johnson, personal fees from Boehringer Ingelheim, non-financial support from iCareWorld, grants from Topcon, grants from Carl Zeiss Meditec, personal fees from Gyroscope, non-financial support from Optomed, non-financial support from Heidelberg, non-financial support from Microsoft, grants from Regeneron, grants from Amazon, grants from Meta, outside the submitted work; This article does not reflect the views of the US FDA. A.T. reports no direct funding conflicts with the content of the paper. Other disclosures not directly related to the content of the paper: Allergan; Annexon; Apellis; Bayer; 4DMT; Genetech; Heidelberg Engineering; Iveric Bio; Novartis; Oxurion; Roche; VisionAI. C.Y.E. reports receiving financial support from the UK Department of Health through an award made by the National Institute for Health Research to Moorfields Eye Hospital NHS Foundation Trust and UCL Institute of Ophthalmology for a Biomedical Research Centre for Ophthalmology. C.Y.E. receives consultant fees from Heidelberg Engineering and Inozyme Pharma unrelated to the scope of the current work. No competing interests reported for M.B., J.O., M.L.G., R.B., B.R.E.A., K.W., V.E.J., L.W.S., S.F., Y.K., Y.W.
