## [Transparent Peer Review file · Nature Communications]

Multi-omic spatial effects on high-resolution AI-derived retinal thickness

Corresponding Author: Professor Melanie Bahlo

Version 0:

Reviewer comments:

Reviewer #1

(Remarks to the Author)

This paper explores the relationship between retinal thickness and various factors including genetic variants, metabolomic profiles, and systemic diseases. Utilizing data from the UK Biobank and advanced AI methodologies, the study processed optical coherence tomography (OCT) images to achieve fine-scale measurements of retinal thickness. Key findings include the identification of novel genetic loci associated with retinal thickness, the association of certain metabolites with retinal health, and the impact of systemic diseases like multiple sclerosis on retinal thinning. This comprehensive analysis demonstrates the potential of AI-enhanced imaging in identifying biomarkers for systemic diseases through retinal examination. The cloud-based website is valuable for the research community. I have following comments or questions.

1. My primary concern is the relevance and impact of the findings. The analysis concentrated on measuring retinal thickness in a healthy retina, which is informative, but the more pressing question for most people is the role that retinal thickness plays in the progression of retinal diseases, particularly at advanced stages. A further concern is the absence of replication across diverse datasets. Can the proposed methods maintain robustness when applied to datasets exhibiting varied image quality levels? Another concern is lack of comparison with traditional methods of measuring retinal thickness, which might highlight the advantages or potential limitations of the AI-driven approach.
2. Regarding your two-step machine learning pipeline, it appears that authors utilized segmentation outcomes from the A star algorithm as pixel-wise labels for your training model, which is based on the Pyramid Parsing Network with a ResNet-18 backbone. Could author elaborate on the accuracy of the A star algorithm's segmentation results in comparison with labels that have been manually graded?
3. Considering the extensive body of research on OCT image segmentation, an ablation study of the Pyramid Parsing Network model's parameters, as well as a comparison of its results with those from leading deep learning approaches, should be added.
4. It is essential to address the absence of segmentation results and details regarding the labeling process for the deep learning model. The inclusion of such information would substantially improve the clarity, reproducibility, and credibility of the methodology proposed.
5. It is mentioned each OCT B-scan has a pixel resolution of 512 by 650. Does the value of 650 represent the axial depth measured from the probe?
6. In the ML pipeline, the dataset comprises a total of 6,409 OCT-segmentation pairs. Could you specify the criteria used for their selection?
7. Are the red lines in the supplementary figures manually emphasized to draw attention?
8. It is specified that pixels with a variance greater than 55 were excluded. Could you elaborate on the rationale behind setting this particular threshold? Given that a larger variance often signifies more information, is it justifiable to discard such informative pixels during this process?

9. In the imputation process for missing pixels, author assumed that the data is missing at random. However, could there be a higher likelihood of missingness for larger or smaller retinal thickness (RT) measurements due to the idiosyncrasies of the OCT machine's performance?

10. The methodology indicates that when both left and right eye data were available, the average value for each pixel was calculated. Was there any consideration given to using either the minimum or maximum pixel values as a representation of individual-level pixel values instead?

11. The paper states that the first two functional principal components (fPCs) capture variation in the perifoveal and outer retinal regions. Could you provide a detailed description of the methodology used to determine the interactions between each fPC and the specific retinal regions? Is the determination influenced by larger weights assigned during the calculation of fPCs? If so, how did you set thresholds for "large weights".

Reviewer #2

(Remarks to the Author)

This paper introduces an approach to automatically identify the retinal thickness (RT) from B-scans. Then using the estimations of many B-scans (about 128 of these scans) taken from the same patient, they reconstruct the spatial thickness around the foveal. This output (Figure 1) is represented as a grid; the color indicates the thickness, and the x-y coordinate indicates the spatial location (e.g. the center is the foveal). This grid contains 29,041 pixels, and each pixel would have a value indicating the thickness of the retina. Next, the authors conducted association study for these pixels against a wide variety of traits; for example, measuring if a specific pixel value is associated with a specific SNP (essentially 29,041 GWAS experiments), along with metabolites, age, and comorbidities.

The authors also applied fPCA (functional principal component analysis) to extract the fPC scores of each grid (e.g., the grid image of 29,041 pixels) based on 6 main fPCs (e.g., basis functions). Then, using the fPC scores as features of each image, they conducted the same types of association studies mentioned above. Several questions and comments are raised in reviewing this manuscript:

1. Method section "Generation of macular retinal thickness data using machine learning" Line 28: Would it possible to see an example of the output after applying A-star algorithm?

Line 30: Using the A-star algorithm, the authors isolated the segmentations of the ILM and RPE layer (e.g, the bright bands representing ILM and RPE) from the rest of the image for a total of 6409 B-scans.

The ILM and RPE segmentations of these 6409 scans and their corresponding B-scans were used to train a neural network segmentation model. By knowing the locations of these bright bands (via the A-star algorithm), can we already measure the retina thickness? Suppose the answer is "yes" for this question, then my next questions are as follows.

Why was the A-star algorithm not applied to the rest of the OCT scans; is it because the algorithm needs manual human labeling for the start and end points? Why do we need to train the neural network if we know how to precisely create the ground-truth segmentation with A-star algorithm?

Line 33: Would it be possible to see an example of the segmentation output of the CNN? There are 2 cited papers here, but I don't see how one would use these 2 papers to construct the CNN. Is the segmentation CNN model in this paper an implementation of this original source <https://github.com/hszhao/PSPNet> (which is one of the cited papers)? For example, was the segmentation CNN model in this paper a finetuned version of <https://github.com/hszhao/PSPNet> ? Having more information would be helpful.

2. The study highlights the interaction between genetics, metabolism, age, and various comorbidities in retinal thickness. How many of the positive genetic loci and flanking regions have been previously associated with diseases such as diabetes or neurodegenerative disorders, or with metabolic pathways that may influence metabolite levels? Are we simply seeing an indirect effect between genetic loci and systemic disease risk, rather than direct biological effects such as those described for gene function. Multiple testing and corrections do raise the confidence that these effects are real. However, it would be valuable to add more information to the discussion to postulate the weight of these indirect effects on RT versus direct genetic effects. Causality and direction of effect is of course not possible to predict from GWAS analyses without functional studies, and the appropriate level of caution of interpretation is laid out by the authors.

3. Supplementary Figure 9 indicates the fPCA is a "two dimensional functional principal component analysis". What is the purpose for using fPC scores when the pixel-level grid already captures much finer detail?

For example, Figure 5A shows the spatial effects of the pixel-level vs. the SNP rs62202906. Thus, we know which locations

around the foveal having positive/negative associations with this SNP. Then, why would we need to also run an GWAS using the fPC scores? Figure 5A looks similar to the fPC4 in the Extended Data 1 which represents “reduced parafoveal thickness”. Hence, the main text reports: “Similarly, the MACROD2 locus (pixel-level sentinel rs62202906, minimum $p = 1.93e-54$, fPC4 sentinel rs62202889, $p=5.78e-58$)”. It seems like the fPC4 does not add much new info than using the pixel-level grid.

For Figure 5B, the SNP rs61916712 is not associated with any of the fPCs, and is associated only when using the pixel-level grid. So, how do we handle cases when fPCs and the pixel-level have different results? In what situations should we trust fPCs instead of pixel-level and vice-versa? For example, Page 5 Line 7: “We identified 140 novel RT loci, 70 uniquely from pixel-level analysis, 28 uniquely from the fPC analysis, with 42 found in both”. What may be causing these differences, is it purely by chance that sometimes fPC scores detect signals when pixel-level fails to? Can this difference occur because of the multiple-testing correction (possibly setting a very conservative p-value cutoff) in pixel-level which we may not have to worry about with just 6 fPC scores? Intuitively, it feels like both fPCs and pixel-level should produce very similar results (because fPC scores are good low-dimension representation of the pixel-level). If anything, perhaps the pixel-level grid feels more informative than the fPC scores, because fPCs are low-dimensional representation of the pixel-level grid.

If possible, can you cluster the OCT scans using the fPC scores (this would be similar to how one may use Principal Components Analysis for clustering)? Would there be any clear and obvious clusters, and would there be any interesting characteristics for these clusters (e.g., a few specific traits are more enriched for some specific clusters)?

Minor comments:

1. “Overrepresentation” and “over-representation” are being used interchangeably; maybe just stick with one version?
2. Sometimes “fPC” is used, and other times “FPC” (for example, “FPC” is used in Method section “Associations in non-European ancestries”).

Reviewer #3

(Remarks to the Author)

The term and concept of “oculomics”, using data to characterise ocular and systemic health have earned a well-deserved legitimacy. Powered by the availability of data from the usual UK Biobank, the manuscript by Jackson et al. is another attempt to use retinal scans as mirrors of systemic health. This work describes how several thousands of OCT scans are compared, pixel by pixel, starting from the fovea, and features obtained are correlated with genotypes, metabolites, and ICD10 codes. The authors find many new genetic loci and associations, and the retinal features are correlated in some ways with a variety of conditions, cardio- and cerebrovascular, chronic and acute inflammatory diseases, liver and respiratory diseases, gastrointestinal neuro-psychiatric, metabolic and cancer.

There are, however, several question marks and potential problems with this paper. When something is associated with almost everything, as in this case, readers may be forgiven for thinking of potential problems with the data structure, analyses, or bias. This manuscript does not contain enough information to dispel such doubts. There is no real replication or validation (similar data are difficult to find outside the UK Biobank), and the authors have not provided any explanation, however tentative, for the underlying biological reasons for some of these associations. Some appear credible (such as multiple sclerosis), but others more far-fetched (like gastric reflux and toothaches). Often, it is unclear what the implications and conclusions from the results are (e.g., “alcohol taken with meals” – is that good or bad for retinal thickness?). There is a lingering doubt that results sometimes derived from a handful of cases (ICD10 codes) may have been overinterpreted and perhaps affected by bias.

Another related question concerns the alignment of OCT retinal features across different images and whether the pixels at a certain distance and location in each image are truly comparable with others. Different factors affect the magnification of retinal images differently, and retinas of larger eyes are physically more stretched than those in smaller eyes. Therefore, two pixels with the same ETDRS grid coordinates may have different characteristics and are not necessarily comparable. If true, the phenotype used for analyses would have flaws that a simple linear correction (age, sex, and body height) can't correct. For example, spherical equivalent is the product of corneal power and eye size, each of which would deform images in different ways. And astigmatism would present a different kind of challenge altogether.

The authors do not provide any metrics of reliability for their genetic results. No measure of inflation is provided. In addition to potential issues with the phenotype, the authors also ran their analyses using simple linear regression. It is well known that one-third of the samples are related to one another, which in theory could unduly inflate the significance of the results. It is possible that with decreasing sample sizes, related samples would gradually become fewer and less problematic, but the gold standard is to run linear mixed regressions, which account for sample non-independence. The inflation factor parameters and LD regression output are routine diagnostic tools that are reported to reassure readers of the quality of results obtained from genetic analyses.

The authors rightly use Bonferroni correction for multiple testing. If properly applied, this correction would have been conservative, since pixels are not independent of other adjacent pixels. However, the use of Bonferroni is inconsistent. While it is generally used to control for the total number of multiple tests, here it is applied to families of analyses, with a different correction factor for the pixel analyses and another for fPCs. And in all tables and supplements, everything above the nominal threshold of $5e-8$ is reported. It is unclear if or what multiple testing correction was applied to metabolites, their ratios and ICD code regressions. The manuscript does not seem to follow a single and consistent strategy on multiple testing correction.

The authors' correlation of genetic effect sizes across different ethnic groups lacks sufficient detail. These correlations would

be primarily more a function of linkage disequilibrium differences between two populations than informative replication of associations. A good comparison should take LD into account (such as the Popcorn method). Details on how the metabolites were analyzed are also very sparse. These analyses are anything but straightforward, and major potential confounders need to be controlled for. BMI and sex are two examples, both of which were already associated with the outcome and therefore potential sources of systematic bias. The authors have tested for associations between polygenic risk scores (PRS) of retinal features with diseases and other complex traits. It is not clear what the justification was for only including a small subset (approximately 10%) of the entire Biobank sample. This may have been the subset for which they had optical coherence tomography (OCT) data, but it is unclear why the authors preferred to run analyses on a much smaller sample when they could have calculated PRS for all half a million participants. The UK Biobank has information on just over 400 cases with multiple sclerosis, how many cases were available for each of the analyses reported? The authors defined diseases based on ICD10 codes, which only partially overlap with the ICD9 information. It is quite common to have a disease reported only in ICD9 and not under ICD10 in the UK Biobank and there is a possibility that some genuine cases could have been counted as controls. Ambiguous or loose terminology is frequently seen in the manuscript. It is not clear what "clustering" analyses were conducted, as they are not described in the Methods section. The authors mention "Concordance was lower in EUR vs AFR" (perhaps meaning "correlation"?), and there is an interchangeable use of Spearman's "rho" and "r" (the former, often used in the text, applies to entire populations, the latter to representative population samples, which would be appropriate in this manuscript). The term "median scan-wise correlation" is difficult to understand, and "log-adjusted p-value values" is almost certainly incorrect. As with wording, the authors could, and should, have been more careful with formatting. Tables and figures were not following the same order in which they were called for in the text. The manuscript starts with a Supplementary Figure 7, and Supplementary Figure 1 is only seen on Page 32. Often, there is very little legend to explain the content (for example, "Supplementary Figure 14: Clustering of pixels, based on SNP effects for loci identified through the pixel-level analyses"). In many instances, unsupported statements are made, e.g. "rs183659670 is associated with NNAT expression in the retina" or "another novel signal ... in a region with a significant chromatin interaction with SOX2 in adult and fetal cortex." (no citation, no such analyses reported in the Methods).

Reviewer #4

(Remarks to the Author)

The manuscript "Multi-omic spatial effects on high-resolution AI-derived retinal thickness" by Jackson et al. presents a comprehensive analysis of genetic, omic, blood trait and disease associations with spatially defined retinal thickness in the human macula, the region in the retina in charge of central vision. This is the first study to apply a deep learning approach, convolutional neural network to over 90k optical coherence tomography (OCT) images from ~55k participants in the UK biobank, to measure retinal thickness in >29,000 pixels in the macula, generating a high resolution spatial retina thickness map. By further applying functional principal component analysis (fPCA) to the 29k pixel-level retinal thickness phenotypes, the authors were able to identified 6 fPCs that captured the majority of variance explained across all participants, which can more easily be used for downstream functional analyses. Genome-wide association study analysis was applied to the retinal thickness across >29k pixels and to the top 6 fPCs, identifying 224 unique loci across all pixels and fPCs tested, with a high concentration of associations in the foveal region. Over half these loci were not found to be associated with retinal thickness in previous GWAS the authors inspected, though there is a more recent study they didn't compare their results to - Zekavat, et al. *Sci Transl Med* 2024 (PMID: 38266105; Zekavat et al., medRxiv May 2023, <https://doi.org/10.1101/2023.05.16.23290063>). Many of the new loci were found to be associated with ocular traits, such as intraocular pressure, vertical cup-disc ratio, and age-related macular degeneration. While the sample size for non-European participants is 40 fold smaller than that of European participants, the authors evaluated the genetic correlation with spatial retinal thickness between the EUR, African and central and south Asian sample sets, and found stronger correlation between the EUR and CSA genetic associations compared to EUR and AFR. The authors integrated various functional genomic data and annotations to prioritize candidate genes for the retinal thickness loci. In addition, the authors test for associations of circulating metabolites, blood and immune biomarkers, ICD10 codes and genetic risk scores of different ocular and systemic disease with macular thickness across the fine-scale spatial grid. They found that specific metabolite associations clustered spatially in the retina, that retinal thinning was associated with multiple systemic disorders such as multiple sclerosis in the nasal perifoveal region, and that parafoveal thickness was particularly susceptible to systemic insults. While some of these retinal thickness associations with systemic diseases were previously reported (Zekavat et al 2024) this is the first time that correlation with spatial location was examined, which may have important clinical implications. Notably, the authors created an interactive website to visualize the OCT features, GWAS, metabolite associations, and other results reported in this paper, which is incredible valuable for investigating the results and generating or testing new hypothesis for follow up studies.

This study was overall done with statistical rigor, and could have great value to the ocular disease community as well as those studying other complex traits. However, there are several important points on the methodologies used or factors considered in the analyses that I think need to be addressed for the results and conclusions to be sound.

Major comments:

1. The authors removed individuals from the analysis in with ocular diseases or taking medications for eye diseases listed in Supplementary table 1. I would recommend also removing individuals that have had laser or surgery treatments to the eye used to treat glaucoma or high eye pressure. The UK biobank codes for these are: 5327 - "Ever had laser treatment for glaucoma or high eye pressure", 5326 "Ever had surgery for glaucoma or high eye pressure"

2. The authors performed GWAS of pixel-level retinal thickness phenotypes and the top 6 fPC score phenotypes using the glm (generalized linear regression) command in PLINK 2.0. glm fits a linear regression model for continuous variables, such as the retinal thickness phenotypes. A linear regression model assume that the residuals are normally distributed. Have the authors checked the distributions of these phenotypes? Are they normally distributed? If there is skewness, rank inverse normal transformation of the response variable might be appropriate to prevent potential outlier associations.

3. Smoking is a known environmental factor that affects retina degeneration and retinal layer thickness as shown in Zekavat SM, et al. *Sci Transl Med.* 2024 (PMID: 38266105). Tobacco use was also amongst the top ICD10 traits associated with retinal thickness phenotypes in this paper in Figure 4d. I think it would be important to check how the GWAS results change when correcting for smoking status in the GWAS model, as this may be a confounding factor. I would recommend adding smoking as a covariate also in the association models of the PRS, metabolomics, ICD10, CCI scores, blood traits and infection antigens with retinal thickness phenotypes.

4. In the Methods section under "Identification of independent significant loci" the authors describe the LD clumping algorithm they employed across all associated variants and retinal thickness pixel-level phenotypes to identify the independent signals for the genome-wide significant loci. It might be informative to use COJO (<https://yanglab.westlake.edu.cn/software/gcta/>) or another similar tool applied to summary statistics to identify secondary, tertiary, etc independent signals in an associated region.

5. The authors removed loci with less than 5 LD-independent SNPs in the region with $P < 5e-5$, as these were assumed to be false positives. Did the authors test for anti-correlation with minor allele frequency, MAF? I am concerned that true associations may be discarded in this approach, as low frequency variants may have few or no LD proxy variants.

6. What is the lambda of the GWAS? This is a standard measure used to evaluate the calibration of the GWAS and identify GWAS with significant inflation or deflation. If lambda values are found to be significantly higher than 1 (e.g., >1.1 , >1.2), computing the intercept in the LD score regression model can help determine if the large lambda is due to uncorrected bias (e.g., population stratification) or to true polygenicity. This step is particularly important since there is no replication of the genetic associations in an independent study. If the lambda is high, I would recommend reporting the genome-wide significant loci using genomic control adjusted p-values.

7. The authors test whether any of the 224 loci they found to be associated with retinal thickness phenotypes have been previously reported, inspecting three publications, but not the latest one which performed GWAS on the different retinal layer thicknesses measured in macula OCT images from the UK biobank - Zekavat SM et al., 2024 (PMID: 38266105). Some of the 140 novel loci reported in this manuscript might have been found in Zekavat et al.

8. To prioritize the causal genes underlying the GWAS loci of retina thickness and fPC phenotypes, the authors integrated a variety of functional genomics data with the genetic loci. I have a few suggestions to make their analysis more specific and rigorous. They mapped eQTLs in retina, blood and brain to GWAS loci using linkage disequilibrium between the variants. It has been shown that not all GWAS variants which are also significant eQTLs at $FDR < 0.05$ or that are in LD with an eQTL, their causal mechanism is via the eQTL (GTEx consortium Science 2020; Barbeira et al *Genome Biology* 2021, PMID: 33499903). A more rigorous approach is to use Bayesian colocalization methods applied to co-occurring GWAS and QTL signals to assess the posterior probability that the GWAS and eQTL loci are tagging the same causal variant, such as eCAVIAR (PMID: 27866706) or SharePro (<https://www.biorxiv.org/content/10.1101/2023.07.24.550431v1>), both of which assume allelic heterogeneity in the loci, which is often the case, and take local LD into account.

The authors use chromatin interaction maps (Hi-C) from fetal and adult human brain to link GWAS SNPs to genes based on overlapping enhancer-promoter and promoter-promoter interaction regions. While I assume that many chromatin interactions are shared between brain and retina, there are chromatin interactions that are likely retina-specific. Thus, I would recommend also testing the overlap of retina Hi-C interactions (Marchal et al., *Nat Comm* 2022; PMID: 36207300) with the GWAS loci to prioritize potential implicated genes.

9. The authors test whether the 224 retinal thickness loci identified are associated with other ocular diseases and traits, stating that "Many of these have been previously implicated in GWAS for ocular traits". They highlight/describe a few results on page 7 referencing Figure 5. Can the authors be more specific about how many of the 224 retinal thickness loci (known and new) were associated with other ocular traits with published GWAS, and can the authors summarize these results in Supplementary tables or add relevant columns to Supplementary Tables 5 and 6. It would also be interesting to perform a PheWAS of the 224 loci with other traits to test the pleiotropic effect of these variants on non-ocular traits, or look these results up in the UKB or other biobank PheWAS.

10. The authors note in the Methods section on page 28 that they used polygenic risk scores (PRS) to investigate the relationships of traits with retina thickness. The authors should distinguish between two types of PRS here. The PRS they define as type 1 (trait PRS) and type 2 (metabolic PRS, blood abundance of metabolites) that the authors downloaded from the UK biobank, are indeed PRS as these scores considered the effect sizes of variants genome-wide irrespective of whether they passed a given significance cutoff. However, the type 3 PRS or internal PRS computed for macular disorder and retinal thickness related measures, and the PRS computed for metabolites measured with the Biocrates platform, based on the definition on page 28, lines 21-24, are not PRS but rather a genetic risk score (GRS), as the score only considers the effects of a set of variants that are genome-wide significant in the respective GWAS and not all variants genome-wide. Please clarify this point in the paper and use the correct terminology in the Results section. PRS have been shown to be more powerful than GRS, and it would be better to use a PRS if the summary statistics (beta and std) are available for the given GWAS and not only the genome-wide significant loci.

11. For the correlation between PRS and ICD10 pixel-level effects, I would suggest correcting for age, sex, top 10 genotype PCs.

12. The authors refer to PRS, metabolomics, ICD10 scores, Charlson Comorbidity Index (CCI) computed based on disease categories, blood traits and infection antigens all as omics data in the Results and Methods (-Omics Data Analyses). I think the authors should make the distinction in the text between the omics data, such as PRS and metabolomics, and phenotype/disease or lab based traits, such as ICD10 scores, CCI and blood traits and infection antigens.

Minor comments:

1. I would change the header "Main" to "Introduction".
2. In the Introduction, the authors can add a few more references to the first sentence, such as = Liang et al., Cell Genomics 2023, Yan et al Cell Reports 2020; and add Monavarfeshi et al., PNAS 2023 as a reference for single cell data in the optic nerve head and optic nerve.
3. The authors should add a few references to the second sentence in the Introduction on page 1, lines 4-5 and lines 8-15.
4. There are a few additional references that would be appropriate to add on page 1, lines 18-22, such as the association of OCT-measured retinal layer thickness with cognitive function and decline (Sekimitsu et al., British Journal of Ophthalmology 2023; PMID: 36990674) or with neuropsychiatric and cardiometabolic disorders (Zekavat et al. , 2024; PMID: 38266105).
5. On page 1, line 28, the authors should add the reference Zekavat et al. , 2024; PMID: 38266105 to the GWAS of retinal sublayers performed.
6. In the final paragraph of the Introduction, where the authors summarize the main results of the paper, the authors can add a few more sentences with a bit more detail, be a bit more specific on the main findings.
7. In the description of the quality control of the genotype data in the Methods section, the authors note that pairwise relationships, up to 3rd degree were removed. Can they add more details on the method and cutoff they used to identify related individuals.
8. In the Methods section under "Genome-wide Association Analyses" the authors note that the covariates were standardized with mean 0 and covariance 1. Did the authors mean 'variance' of 1, not covariance?
9. In the Results section, it would be helpful to clarify that the functional PCA is applied to the pixel level retinal thickness phenotypes from the CNN analysis of OCTs. This is described in the Methods section, but would be good to mention this in the Results section, so it will be easier for the reader to follow what was done.
10. Figure 4 legend, a legend for the bubble size and color is missing. Please define ORA in figure legend.
11. On page 5, lines 18-19 I think the reference is to Suppl Table 13, not Suppl Table 12. I might recommend adding the FDR cutoff used for the gene set enrichment results in the Results section.
12. The authors performed unsupervised clustering of the SNPs based on their association results with ~29k pixel level phenotypes and identified 10 clusters (show in Suppl Figure 13). Can the authors list the SNPs found for each of the 10 clusters in a Supplementary table?
13. On page 2, lines 38-45, the authors describe some of the patterns observed from performing association testing of the retinal thickness phenotypes (pixel level phenotypes and 6 fPCs) with different characteristics including sex, age, height and spherical equivalence, however they don't show these results in supplementary tables or figures. Supplementary figure 10 shows the spatial distribution of the association significance (-logP) with the pixel level phenotypes, but not the effect size and direction of effect. This would be informative to show and record. Can the authors include these results in Suppl Tables and figures.
14. The authors created an "evidenceScore" to prioritize candidate genes in a given GWAS locus based on the number of lines of evidence for that gene. They point to a column in Supplementary Table 10, which I assume is the column named "sum_cols". It would be good to clarify this in the Methods section.
15. In the 'Metabolomic association analyses' section in the Results section, it would be informative to add some statistics on the top metabolite associations highlighted, e.g., beta, adjusted p-value, number of pixels they are associated with.
16. It might be worth noting that most of the metabolite associations with retinal thickness phenotypes were with pixels in the foveal and nasal regions.
17. On page 6, lines 10-11, the authors point to Supplementary Table 15, for the age by metabolite interaction results, but I do not see those results there. Also, on page 6, lines 14-18, the authors discuss the association results of the metabolite PRS with retinal thickness and point to Supplementary Table 16 and Extended Data 8D. There is no panel D in Extended Data 8.
18. When you first mention testing for association of PRS with retinal thickness in the Results section in regards to metabolic PRS on page 6, line 14-18, I think it would be helpful to add a sentence on how the PRS correlation for metabolic PRS as with the other traits, was applied to all pixel retinal thicknesses and that the average beta and median p-value across all pixels was used to evaluate correlation (as in Supplementary Table 16 and other Suppl tables), and point the reader to the Methods for more details. This was not clear to me until I read the Methods section.
19. When the ICD10 disease associations with retinal thickness results are described on page 6, it would be informative to provide information on the direction of effect, i.e. is that disease associated with thinning or thickness of the retina.
20. In the Discussion on page, lines 21-23, the authors note that "These complementary approaches, applied to AI-reprocessed data identified a greater number of loci, than previous GWAS conducted with RT derived directly from the TOPCON scanner, despite the heavy multiple testing burden." This is not precise, as in a recent GWAS and PheWAS of retinal thickness measures based on the TOPCON scanner (Zekavat et al., 2024), 259 unique loci were found that is comparable to the 224 loci found in this manuscript. You can emphasize the added value of your work with the spatial information.
21. In the Discussion on the bottom of page 8 to page 9, the authors note "Future work will extend this approach to retinal sub-layers which will further tease apart many of the associations since sub-layers are composed of particular cell types." This has been done in Zekavat et al., 2024, where they measured the thicknesses of different retinal layers in macula OCTs, such as the inner retinal fiber layer or ganglion cell complex layer. Please edit accordingly.

Version 1:

Reviewer comments:

Reviewer #1

(Remarks to the Author)

The authors have provided detailed responses to address most of my comments. The replies have provided a clearer

understanding of the approach and its limitations. To further strengthen the paper, I have several additional questions:

1. Regarding Question 6 and your response to Question 2, after performing A* segmentation, did you manually accept/reject the generated masks individually, ultimately retaining 6,409 out of 8,500 pairs? You mentioned that there are 1,250 hard pairs of B-scans with manually reviewed segmentation. Did you evaluate the performance of the A* algorithm on these 1,250 hard examples?
2. In Supplementary Figure 8, it is noted that the cleaned dataset includes 54,844 subjects with OCT data from at least one eye and corresponding phenotypes. I assume that most image quality control was performed by this step. Could you provide more details on how the dataset was further reduced to 6,409 + 1,250 pairs of B-scans and their corresponding masks, given that each eye has 128 B-scans and there are over 54,000 subjects?
3. One limitation of the paper is its lack of independent validation, which is not available at the current stage. If new data are available in the coming years, authors should make the detailed results (e.g., summary statistics) and code available for other groups to replicate some of the findings.
4. The fonts in most figures are too small to read.

Reviewer #2

(Remarks to the Author)

Authors addressed reviews in their entirety.

Reviewer #3

(Remarks to the Author)

The second version of the manuscript is a definite improvement over the previous one. The authors have diligently gone through the reviewers' comments and responded to the critiques.

While there are many improvements for which the authors may be commended, this second review will mainly focus on what could be better, or what remains unclear in the manuscript.

Before getting to more detail, it seems that sometimes the authors in their rebuttal tend to simply engage in a discussion with the reviewers - this is great but some of the reviewers' comments were about a lack of clarity and reviewers were simply acting as a sample of the journal's readers. Some of the questions were legitimate and the reader would benefit from the inclusion in the manuscript or supplementary information of some clarifications that the authors are currently sharing with the reviewers. This is a suggestion that the authors may want to consider moving forward.

In their rebuttal letter, the authors retort that 'We deliberately steered clear of defining "good" or "bad" for retinal thickness... We summarised patterns of retinal thickness, with our novel FPC approach, prior to any disease association for example... These will aid biological understanding of retinal function.' While this is irreproachable, there was a broader question that went beyond the use of epithets for conditions and phenotypes. There are questions about the presence of the confounding or bias that the authors seem to be evading. It is unclear what certain associations and correlations are teaching us. Some of these associations are exceedingly broad, ranging from dental symptoms, esophageal conditions, to habits or lifestyles. The doubt is that these associations (that were removed from the current version) may not be aiding us much but are arising from some biasing factor (very hard to imagine any real biological relationship between GERD or toothaches and retinal images).

A previous and comment was about pixels with the same ETDRS grid coordinates having potentially different characteristics, maybe due to biasing effects of refraction, eye size, etc. The authors' response is that "previous research papers... have aligned OCT volumes by registering the fovea and measuring anatomical regions relative to the fovea... these effects do not invalidate our work nor much of the previous research in retinal analysis." And then "if anatomical alignment to the fovea were invalid or fortuitous, one would expect the peaks of SNP significance to be randomly uniformly distributed and isolated and create a bias towards the null hypothesis".

It may be difficult to accept the validity of this argument. While there may be some precedent set by similar publications, there are questions about how a potential bias may have affected the results of this paper. The assumption the authors are making is that anatomical alignment to the fovea were fortuitous, or random and a source of type II error. The real concern here is the possibility that an anatomical misalignment happened due to the presence of biasing factors. For instance, a non-random factor (refraction, eye size) may affect the size of the retinal images, the concentric patterns described on pages 4, line 30 and page 5 line 17 of the revised version of the manuscript may be explained by non-genetic factors (socio-economic, nutrition, education). Bias related to socio-economic factors and nutrition has in the past been difficult to control by simply using linear adjustment and is frequently encountered in genetic association and often generates correlations similar to those observed by the authors. This point is not meant to be destructive or invalidate this work and manuscript. But when the most significant correlations for the first factor on table S15 are still reported for "milk type used" and "bread type: brown", this will take some acknowledging and possibly explaining.

The authors respond to the criticism about inconsistent use of multiple testing correction methodologies, and they clarify their position that despite the choice to adopt Bonferroni (an over-conservative approach), to use multiple approaches for multiple testing corrections in our manuscript that align with those traditionally in literature used for each of the respective data axes. The reviewer has sympathy for this argument, but this is also why this approach that seeks to have the best of several worlds may unfortunately be called inconsistent. Previous publications will follow have ways to correct for multiple testing, but rarely

they would use all of them in one place, and much less frequently ignore them in the supplementary tables (as tables S5 and S6) where the correction is made for the genome-wide level but not number of phenotypes. This reviewer is not recommending going to extremes, but some effort may need to be made in that respect.

In the previous iteration, the reviewers found that descriptions of the metabolome statistical analyses were not exhaustive and clear. The authors responded by writing that "all our models were corrected for genetically derived sex at birth, age, imaging device number, standing height, mean refractive error measured by spherical equivalent, eye and the first ten genetic principal components".

This reviewer finds this description unclear; 'limma' is just a package and linear models can be very different. What goes into the design matrix, how was the relationship between the different dependent variables modeled? For some people familiar with 'limma', Bayesian statistics are applied to independent variables (originally RNA probes, but this can be easily extended to other omics, such as metabolites). The authors mention that an empirical Bayes framework to model relationships between outcome variables (here RT pixels), which is all great, but model specifications are not well explained or documented.

A very minor point: the authors write in their letter that "had we included related individuals, we would, as the reviewer notes, needed to have used a more sophisticated model, and this would have been prohibitively computationally expensive." The authors' approach (linear regressions and, not linear mixed models) is very reasonable. Having said that, these techniques are not such a big deal, are widely used and computationally efficient since they take only hours to complete. However, this is a minor point, and the reviewer is happy to accept the authors' approach to GWAS and the drop in power is negligible, just as the authors state in their reply.

Reviewer #4

(Remarks to the Author)

The authors have done a lot of work to address my comments most of which are satisfactory. There are just a few mostly minor points/comments that I think need to be addressed or to be corrected before the manuscript can be accepted for publication. The pages and line numbers I refer to below were taken from the revised manuscript with the edits tracked in red (478786_1_related_ms_9158970_sspfgs.pdf).

1. The authors performed colocalization analysis between the GTEx and Retina eQTLs and the GWAS loci of retina thickness and fPC phenotypes, and used Hi-C from retina and brain to propose target genes. Can the authors add to a table the list of loci and genes proposed to be the underlying causal gene based on colocalization and Hi-C data. It would also be biologically meaningful to test in what pathways the significant genes are enriched from the colocalization and Hi-C analyses using gene set enrichment analysis.

2. In reply to question 9, it was not clear to me why the authors chose not to include the bar plots showing the top PheWAS results for ocular traits in the manuscript. Also, why are the number of variants associated with some of the ocular traits different between the bar plots shown in the rebuttal below the plots included Suppl fig. 18 different than the bar plots included in Suppl fig. 18.

3. I would suggest the following edit in bold to this sentence on that was added on page 33, lines 1-20: "It is worth noting that prediction power of PRS and GRS may vary substantially as the variant weights of the PRS were constructed to optimize prediction, while the variant weights of the GRS were derived from standard GWAS summary statistics."

4. The authors have clarified that they use both polygenic risk scores and genetic risk scores in the manuscript on page 9, lines 28-30, by adding the following sentence (and a clear explanation in the Methods section): "We also examined associations at the pixel-level between RT, and PRS or genetic risk scores, for a number of diseases (for simplicity collectively referred as PRS below, see Methods)." However, I would suggest not to change the word GRS to PRS in the Results section since they refer to different scores and have different levels of power. I would suggest keeping GRS and editing the sentence on page 9, lines 28-30, as follows:

"We also examined associations at the pixel-level between RT, and PRS or genetic risk scores (GRS), for a number of diseases (see Methods)."

5. On page 4, lines 27-28, the authors note: "We identify 224 unique RT-associated genetic loci that met the pixel-level Bonferroni corrected 28 threshold (Supplementary Table 5)." But I only see 221 variants with "clusterAssignment" values in Supplementary Table 5. Also, there are 959 variants in Supplementary Table 5. What are the other variants? Are they variants that pass genome-wide significance that are in LD with the lead 224 significant variants? Please clarify this somewhere in the text, or if you have legends for the Supplementary tables you can add the description there.

6. In regards to my question #19 "When the ICD10 disease associations with retinal thickness results are described on page 6, it would be informative to provide information on the direction of effect, i.e. is that disease associated with thinning or thickness of the retina." the authors did not seem to address it. The authors added number of cases, number of significant pixels, average beta, and median -log10 p-value for all Phecode associations, but I was asking about adding information on the relative direction of effect of the ICD10 associations on retinal thickness. For example do the phenotypes increase or decrease retinal thickness? Can the authors also add this to the relevant Supplementary table (22?). I think the authors tried to address this is on page 8. There seems to be several typos on page 8 lines 28-29 in this sentence (under the subheader "Disease PheCodes"): "More generally, even among those non-significant, there was a negative relationship between disease sta We assessed the effects tus and retinal thickness."

Version 2:

Reviewer comments:

Reviewer #1

(Remarks to the Author)

The authors have addressed my additional comments. It will be valuable work in the field.

Reviewer #3

(Remarks to the Author)

The authors have responded to criticism by acknowledging some limitations of their work. No further comments or suggestions.

Reviewer #4

(Remarks to the Author)

The authors have addressed all of my comments. I just have a two small corrections:

1. In regards to the colocalization analysis performed between the pixel-level retina thickness and fPC phenotypes and GTEx and Retina eQTLs the authors used FUMA that uses an older release of GTEx (v7), which while is contained within GTEx release v8, it is not directly described in the cited publication: GTEx Consortium. The GTEx Consortium atlas of genetic regulatory effects across human tissues. *Science* 369, 1318–1330 (2020). To avoid confusion in the Results section, I would suggest adding the release version of GTEx as follows: "(GTEx v7)" in the Results section on page 4, line 43.
2. On page 4, line 43, there is a typo - 'eyeGX' should be 'EyeGEx'.
3. In response to my previous comment #2, the authors have assessed the normality and skewness of the pixel-level retinal thickness phenotypes and the effect of performing rank inverse normal transformation of the pixel-level retinal thickness phenotypes and 6 fPC score phenotypes on the GWAS results. They show that they are not perfectly normally distributed and have shown that the beta's and p-values highly correlate between the RINT normalized and raw values. The authors added in the Methods section the sentence: "All phenotypes were confirmed to follow approximately normal distributions. " Based on their plots there is some skewness to their phenotype distributions, but it does not seem to have a significant effect on the GWAS results. I would suggest including the phenotype distributions, Pearson's moment coefficient of skewness heatmap and the scatter plot between the GWAS summary statistics with and without the transformation in Supplementary Figures and referring to them in the Methods section. I think this would be useful for the readers to see.

Notes

- We have used the following formatting in responses:
 - **General responses**
 - *Additions to the manuscript*
- We have provided both a “clean” version of the revised manuscript, as well as a version with changes tracked. All line numbers relate to the clean version.

REVIEWER COMMENTS

Reviewer #1 (Remarks to the Author):

This paper explores the relationship between retinal thickness and various factors including genetic variants, metabolomic profiles, and systemic diseases. Utilizing data from the UK Biobank and advanced AI methodologies, the study processed optical coherence tomography (OCT) images to achieve fine-scale measurements of retinal thickness. Key findings include the identification of novel genetic loci associated with retinal thickness, the association of certain metabolites with retinal health, and the impact of systemic diseases like multiple sclerosis on retinal thinning. This comprehensive analysis demonstrates the potential of AI-enhanced imaging in identifying biomarkers for systemic diseases through retinal examination. The cloud-based website is valuable for the research community. I have following comments or questions.

1. My primary concern is the relevance and impact of the findings. The analysis concentrated on measuring retinal thickness in a healthy retina, which is informative, but the more pressing question for most people is the role that retinal thickness plays in the progression of retinal diseases, particularly at advanced stages. A further concern is the absence of replication across diverse datasets. Can the proposed methods maintain robustness when applied to datasets exhibiting varied image quality levels? Another concern is lack of comparison with traditional methods of measuring retinal thickness, which might highlight the advantages or potential limitations of the AI-driven approach.

Our work focused on understanding the drivers of retinal architecture in retinal-normal individuals. The UK Biobank represents a “healthy” ageing population, and thus is a dataset suited for examining “normal variation” in retinal thickness in the general population. Our approach aimed to exclude individuals with suspected retinal diseases, who may have major and unusual variation in RT, thus creating a more homogeneous cohort, thereby maximising our power for discovery. While we did identify a number of individuals in UK Biobank with retinal disease, phenotyping primarily based on linked health records, is limited in detail and may be subject to misclassification. Investigation of RT in the context of overt disease is of great interest, but may be most informative in disease cohorts of individuals with deeper retinal phenotyping.

As we note in the discussion, the UK Biobank is a uniquely massive resource. At present, there are no other datasets that have retinal OCT imaging available, in addition to the wealth of genetic and phenotypic information recorded,

that would allow comparable studies to comprehensively replicate our findings. Our genetic discovery analyses were restricted to individuals of European ancestries; we sought to replicate identified SNP associations in individuals in UK Biobank with other ancestries, and found effects to be positively correlated. Our discussion now notes (page 11, lines 24-25):

Further investigation in individuals with greater ancestral diversity is required; and emerging efforts to increase diversity in population cohorts will hopefully allow this.

We also aimed to give further weight to many of the main findings through complementary approaches, such as utilising PRS alongside directly measured phenotypes.

We did not have access to the TopCon 1000 segmented ILM-RPE thickness that was made available to Patel et al (<https://pubmed.ncbi.nlm.nih.gov/26746598/>), at the time of commencement of the experiments. While the ILM-RPE thickness used in Patel et al have recently been made available, they only provide average ILM-RPE thickness by ETDRS grid locations. In this work, we sought to study the spatial relationships between ILM-RPE thickness across the macula given the richness of the UKBiobank dataset. This is why we developed our own pipeline to segment the ILM-RPE ourselves to achieve this objective.

2. Regarding your two-step machine learning pipeline, it appears that authors utilized segmentation outcomes from the A star algorithm as pixel-wise labels for your training model, which is based on the Pyramid Parsing Network with a ResNet-18 backbone. Could author elaborate on the accuracy of the A star algorithm's segmentation results in comparison with labels that have been manually graded?

The segmentation pipeline has 3 parts, and the first two parts are described in Muller et al, <https://jov.arvojournals.org/article.aspx?articleid=2783598>. We will describe these first two parts in more detail in the methods, as we had originally focused on the additional third part for this paper. First, in our pipeline, we used A* to generate initial segmentation masks for the deep learning part of our pipeline. A* is widely used in pathfinding and can be used to track retina layer boundaries in this use case. However, A* is not robust to poor contrast or abnormal brightness and random noise. Therefore, only ILM-RPE thickness between 30 and 80 pixels, corresponding to 180 to 480 micrometers, were retained. A* was applied to 1 random B-scan from 8500 FDA data, resulting in 6409 after the thickness restriction. Next, the 6409 B-scans and their segmentation masks, along with another 1250 hard examples, B-scans and their manually reviewed segmentations were retained; no corrections were made to A-star segmentations; they were only rejected or accepted as correctly segmented. We provide some examples of A-star segmentations that were accepted and rejected below. Each pair of images show the raw B-scan on the left and the A-star segmentation on the right. We rejected the second and third

segmentations as they were not fully correct. These below figures have been added as Supplementary Figure 1.

Supplementary Figure 1: Example A segmentations with the raw B-scan on the left and the A* segmentations on the right. A* does well in some cases such as the top pair (a), but can fail locally in the middle pair or catastrophically in the last pair (b and c).*

These reviewed B-scans and A-star segmentation masks were used as training data for the second part of the pipeline, the deep learning model called PSPNet. PSPNet was a state-of-the-art segmentation model that incorporated a pyramidal structure to allow segmentation at varying image and resolution scales. Our PSPNet achieved an intersection over union (IOU) of 0.97 on the test set. IOU is computed as $TP/(TP + FP + FN)$, where TP is True Positive pixels where the ground truth and PSPNet prediction agree, FP is the False Positive pixels and FN is the false negative pixels where ground truth and PSPNet disagree. As IOU incorporates true positives and false discoveries, it is a very conservative score, even compared to the Dice coefficient to which it has direct functional equivalence. Therefore, an IOU of 0.97 is an extremely high score.

Furthermore, for this paper, we realized that we needed an additional quality control step in the pipeline. This quality control step filters out erroneous segmentations from the PSPNet, which was often due to the poor quality of the B-scans in the UKBiobank TopCon 1000 OCT volumes. While the previous two parts in the pipeline had controlled for OCT brightness and contrast, it had not controlled for OCT capture quality, as many UKBiobank OCTs are captured off-center, with parts of the retina missing. In addition, our new addition to the pipeline detects discontinuities in the layer segmentation, and ignores those areas of discontinuity, while preserving the remaining parts of the layer segmentation.

This process is now described in detail in the methods (p23 line 31 to p24 line 17):

A is widely used in pathfinding and can be used to track retina layer boundaries in this use case. However, A* is not robust to poor contrast or abnormal brightness and random noise. Therefore, only ILM-RPE thickness between 30 and 80 pixels, corresponding to 180 to 480 micrometers, were retained. A* was applied to 1 random B-scan from 8500 FDA data, resulting in 6409 after the thickness restriction. Next, the 6409 B-scans and their segmentation masks, along with another 1250 hard examples, B-scans and their manually reviewed segmentations were retained; no corrections were made to A-star segmentations; they were only rejected or accepted as correctly segmented. Examples of A-star segmentations that were accepted and rejected are provided in Supplementary Figure 1, which shows three pairs of images with the raw B-scan on the left and the A-star segmentation on the right. The second and third segmentations in this figure were rejected as they were not fully correct. These reviewed B-scans and A-star segmentation masks were used as training data for the second part of the pipeline, the deep learning model called PSPNet. PSPNet was a state-of-the-art segmentation model that incorporated a pyramidal structure to allow segmentation at varying image and resolution scales. Our PSPNet achieved an intersection over union (IOU) of 0.97 on the test set. IOU is computed as $TP/(TP + FP + FN)$, where TP is True Positive pixels where the ground truth and PSPNet prediction agree, FP is the False Positive pixels and FN is the false negative pixels where ground truth and PSPNet disagree. As IOU incorporates true positives and false discoveries, it is a very conservative score, even compared to the Dice coefficient to which it has direct functional equivalence, and 0.97 an extremely high IOU*

3. Considering the extensive body of research on OCT image segmentation, an ablation study of the Pyramid Parsing Network model's parameters, as well as a comparison of its results with those from leading deep learning approaches, should be added.

We performed an ablation study when designing the PSPNet part of the pipeline for Muller et al, <https://jov.arvojournals.org/article.aspx?articleid=2783598>. There was little difference to the model test IOU of 0.97 with model parameter tweaks. In addition, other baseline models, such as UNet did not beat PSPNet's IOU of 0.97.

The following text has been added to the methods (p24. line17-19):

An ablation study was performed on PSPNet hyperparameters, but there was little difference in model IOU performance of 0.97. In addition, the popular UNet model was also used, but underperformed compared to the PSPNet.

4. It is essential to address the absence of segmentation results and details regarding the labeling process for the deep learning model. The inclusion of such information would substantially improve the clarity, reproducibility, and credibility of the methodology proposed.

We added more details to our segmentation methodology in the methods. See response to comment 2 for more details.

5. It is mentioned each OCT B-scan has a pixel resolution of 512 by 650. Does the value of 650 represent the axial depth measured from the probe?

The Topcon 1000 captures a 6mm*6mm*2.275mm area around the macula, and this maps to 128 B-scan slices * 512 pixels * 650 *pixels. The 650 pixels correspond to the anterior to posterior direction.

This is now clarified in the methods (p.23 line 27):

with the 650 pixels corresponding to the anterior to posterior direction.

6. In the ML pipeline, the dataset comprises a total of 6,409 OCT-segmentation pairs. Could you specify the criteria used for their selection?

The training data started with A* predictions. However, A* is not robust to poor contrast or abnormal brightness and random noise. Therefore, only ILM-RPE thickness between 30 and 80 pixels, corresponding to 180 to 480 micrometers, were retained. A* was applied to 1 random B-scan from 8500 FDA data, resulting in 6409 after the thickness restriction. This removed the poorly segmented B-scans, many of which were not captured properly, either being off-centre or not containing the entire retina. See response to comment 2 for more details.

7. Are the red lines in the supplementary figures manually emphasized to draw attention?

Yes, we added the red lines to emphasise the segmentation achieved by PSPNet.

8. It is specified that pixels with a variance greater than 55 were excluded. Could you elaborate on the rationale behind setting this particular threshold? Given that a larger variance often signifies more information, is it justifiable to discard such informative pixels during this process?

While we agree that pixels with higher variance may be providing more information in terms of natural variation, in this case we believe the higher variances were a result of measurement area. All pixels with variance >55 were located at the edge of the macula (at the end of B-scans, Below figure, left). All of these pixels were in fact additionally flagged for exclusion, due to high levels of missingness (Figure, right); as such the variance filter was somewhat redundant, and we have now removed this.

Plot showing pixels flagged for exclusion to high variance (left), and due to high missingness (right).

9. In the imputation process for missing pixels, author assumed that the data is missing at random. However, could there be a higher likelihood of missingness for larger or smaller retinal thickness (RT) measurements due to the idiosyncrasies of the OCT machine's performance?

It is possible that missingness could be more likely to occur for scans with particularly high or low measurements of retinal thickness, thereby resulting in missingness not at random (MNAR). In practice, determining whether missing data are random or not is extremely challenging. Our interpolation strategy to impute values, used measurements from nearby, non-missing pixels, which we considered to be the best available predictors for the missing measurements. We tried to limit spurious values from imputation, by only using scans with at least 90% complete data.

We have added an additional note to the discussion stating the limitations of our imputation approach.

Discussion (page 11, lines 14-19):

The complexity of the high resolution phenotypes used here, required us to make a number of pragmatic decisions regarding our approach. Among these was the decision to impute missing data points, and in our GWAS to exclude loci with fewer than five SNPs showing association with $p < 5E-5$. The imputation assumed data were missing at random, however missingness could have been more likely to occur for pixels with more extreme values of RT, while the GWAS filtering may have resulted in some true associations being excluded

10. The methodology indicates that when both left and right eye data were available, the average value for each pixel was calculated. Was there any consideration given to using either the minimum or maximum pixel values as a representation of individual-level pixel values instead?

We had not considered taking the maximum or minimum values, and thank the reviewer for this suggestion. We considered several strategies for selecting measurements, including randomly selecting one eye, where both were available. Ultimately we chose to take the mean of the left and right eye, as this was an approach used by others previously,¹⁻³ and we felt it was in keeping with our aim of investigating factors affecting normal variation on retinal thickness.

11. The paper states that the first two functional principal components (fPCs) capture variation in the perifoveal and outer retinal regions. Could you provide a detailed description of the methodology used to determine the interactions between each fPC and the specific retinal regions? Is the determination influenced by larger weights assigned during the calculation of fPCs? If so, how did you set thresholds for “large weights”.

Interpretation of the fPCs was qualitative, and based on visual inspection of the plots presented in Extended Data 1. We have made this clear in the main text, by adding the following text to the results (page 3, line 27-30)

The variation explained by each FPC was assessed by visual examination of plots showing: i) the correlation between the FPC scores and each pixel, and ii) the mean pixel-level RT measures for the 100 individuals with the highest FPC scores, versus the 100 individuals with the lowest FPC scores (Extended Data 1).

Reviewer #2 (Remarks to the Author):

This paper introduces an approach to automatically identify the retinal thickness (RT) from B-scans. Then using the estimations of many B-scans (about 128 of these scans) taken from the same patient, they reconstruct the spatial thickness around the foveal. This output (Figure 1) is represented as a grid; the color indicates the thickness, and the x-y coordinate indicates the spatial location (e.g. the center is the foveal). This grid contains 29,041 pixels, and each pixel would have a value indicating the thickness of the retina. Next, the authors conducted association study for these pixels against a wide variety of traits; for example, measuring if a specific pixel value is associated with a specific SNP (essentially 29,041 GWAS experiments), along with metabolites, age, and comorbidities.

The authors also applied fPCA (functional principal component analysis) to extract the fPC scores of each grid (e.g., the grid image of 29,041 pixels) based on 6 main fPCs (e.g., basis functions). Then, using the fPC scores as features of each image, they conducted the same types of association studies mentioned above. Several questions and comments are raised in reviewing this manuscript:

1. Method section “Generation of macular retinal thickness data using machine learning” Line 28: Would it possible to see an example of the output after applying A-star algorithm?

Below are three example pairs of the A-star algorithm and the original B-scan. In the first pair, A-star does a good job. In the second pair, A-star makes a mistake to the left of the B-scan. In the last pair, A-star fails for half the B-scan, when the ILM is still visible. This shows that the A-star algorithm by itself is not robust enough to segment the OCTs in the UKBiobank.

Line 30: Using the A-star algorithm, the authors isolated the segmentations of the ILM and RPE layer (e.g, the bright bands representing ILM and RPE) from the rest of the image for a total of 6409 B-scans.

The ILM and RPE segmentations of these 6409 scans and their corresponding B-scans were used to train a neural network segmentation model. By knowing the locations of these bright bands (via the A-star algorithm), can we already measure the retina thickness? Suppose the answer is “yes” for this question, then my next questions are as follows.

The A-star is not sufficient for segmenting the B-scans as it often fails in areas of lower contrast as demonstrated in the examples provided above. That is why the A-star predicted segmentations were reviewed and only ones that were 100% correct were selected as training examples for the PSPNet deep learning model. These details are now described in detail in the methods (p23 line 31 to p24 line 17):

A is widely used in pathfinding and can be used to track retina layer boundaries in this use case. However, A* is not robust to poor contrast or abnormal brightness and random noise. Therefore, only ILM-RPE thickness between 30 and 80 pixels, corresponding to 180 to 480 micrometers, were retained. A* was applied to 1 random B-scan from 8500 FDA data, resulting in 6409 after the thickness restriction. Next, the 6409 B-scans and their segmentation masks, along with another 1250 hard examples, B-scans and their manually reviewed segmentations were retained; no corrections were made to A-star segmentations; they were only rejected or accepted as correctly segmented. Examples of A-star segmentations that were accepted and rejected are provided in Supplementary Figure 1, which shows three pairs of images with the raw B-scan on the left and the A-star segmentation on the right. The second and third segmentations in this figure were rejected as they were not fully correct. These reviewed B-scans and A-star segmentation masks were used as training data for the second part of the pipeline, the deep learning model called PSPNet. PSPNet was a state-of-the-art segmentation model that incorporated a pyramidal structure to allow segmentation at varying image and resolution scales. Our PSPNet achieved an intersection over union (IOU) of 0.97 on the test set. IOU is computed as $TP/(TP + FP + FN)$, where TP is True Positive pixels where the ground truth and PSPNet prediction agree, FP is the False Positive pixels and FN is the false negative pixels where ground truth and PSPNet disagree. As IOU incorporates true positives and false discoveries, it is a very conservative score, even compared to the Dice coefficient to which it has direct functional equivalence, and 0.97 an extremely high IOU*

Why was the A-star algorithm not applied to the rest of the OCT scans; is it because the algorithm needs manual human labeling for the start and end points? Why do we need to train the neural network if we know how to precisely create the ground-truth segmentation with A-star algorithm?

See response above.

Line 33: Would it be possible to see an example of the segmentation output of the CNN? There are 2 cited papers here, but I don't see how one would use these 2 papers to construct the CNN. Is the segmentation CNN model in this paper an implementation of this original source <https://github.com/hszhao/PSPNet> (which is one of the cited papers)? For example, was the segmentation CNN model in this

paper a finetuned version of <https://github.com/hszhao/PSPNet> ? Having more information would be helpful.

We trained a PSPNet per the repo cited in the original paper on the manually reviewed A-star predicted segmentations. This PSPNet achieved an intersection over union (IOU) of 0.97.

2. The study highlights the interaction between genetics, metabolism, age, and various comorbidities in retinal thickness. How many of the positive genetic loci and flanking regions have been previously associated with diseases such as diabetes or neurodegenerative disorders, or with metabolic pathways that may influence metabolite levels? Are we simply seeing an indirect effect between genetic loci and systemic disease risk, rather than direct biological effects such as those described for gene function. Multiple testing and corrections do raise the confidence that these effects are real. However, it would be valuable to add more information to the discussion to postulate the weight of these indirect effects on RT versus direct genetic effects. Causality and direction of effect is of course not possible to predict from GWAS analyses without functional studies, and the appropriate level of caution of interpretation is laid out by the authors.

We agree that more work is required to fully disentangle the complex relationship between genetics, metabolites, disease and retinal thickness. Our PheWAS and genetic correlation analyses suggest there is at least a partial shared genetic basis between RT and several of the disease/traits, shown to be associated with RT. For example T2D and hypertension showed association with measured retinal thickness, and were also found to have shared genetic determinants with retinal thickness, in our genetic correlation analyses. Determining whether these diseases' association with retinal thickness is due to causality or simple shared genetic effects is beyond the scope of the present paper, but may represent an important avenue for future work.

We have amended some sections of the discussion to reflect these important points.

Page10 lines 6-11:

Our analyses reveal RT to be intricately related to a plethora of factors, spanning the genome, metabolites, blood traits, and diseases, with the parafoveal area most enriched for associations. All relationships identified through these analyses represent observed correlations, with further work required to disentangle the complex relationships between these facets. Overall, we found reduced RT, or retinal thinning, to be associated with poorer health, and increased burden of disease.

page10 lines 38-41):

For many of the diseases linked to RT, through observed associations, or via genetics, further work is needed to determine whether these diseases may be causally affecting RT, or simply correlated due to a shared genetic basis, or other confounding factors.

3. Supplementary Figure 9 indicates the fPCA is a “two dimensional functional principal component analysis”. What is the purpose for using fPC scores when the pixel-level grid already captures much finer detail?

The pixel-based and FPC approaches, we consider to be complementary. While there is large overlap on the associations detected by the two approaches, we did find there to be a number of associations picked up via one approach and not the other. As the reviewer notes, the pixel-level approach allows the examination of thickness in very fine detail; however we think the FPCs are not only measuring retinal thickness but are also capturing additional nuances of retinal morphology. We provide some examples below to illustrate this.

For example, Figure 5A shows the spatial effects of the pixel-level vs. the SNP rs62202906. Thus, we know which locations around the foveal having positive/negative associations with this SNP. Then, why would we need to also run an GWAS using the fPC scores? Figure 5A looks similar to the fPC4 in the Extended Data 1 which represents “reduced parafoveal thickness”. Hence, the main text reports: “Similarly, the MACROD2 locus (pixel-level sentinel rs62202906, minimum $p = 1.93e-54$, fPC4 sentinel rs62202889, $p=5.78e-58$)”. It seems like the fPC4 does not add much new info than using the pixel-level grid.

In the example given by the reviewer, the SNP association with rs62202906 was identified in both the pixel-level and FPC4 analyses. There were some SNP associations picked up by FPC4, which were not identified through the pixel level analyses. FPC4 is capturing thinning around the parafovea, and inferior nasal quadrant, in combination with thickening in the superior nasal quadrant (or vice-versa). Similarly to rs62202906, these features are seen for rs773313456, and rs40160036, shown below. In both these examples, the features captured by FPC are observable, but no pixel shows association with SNP at $p < 1.72E-12$, or even the less conservative genome-wide significant level $p < 5E-8$.

For Figure 5B, the SNP rs61916712 is not associated with any of the fPCs, and is associated only when using the pixel-level grid. So, how do we handle cases when fPCs and the pixel-level have different results? In what situations should we trust fPCs instead of pixel-level and vice-versa? For example, Page 5 Line 7: “We identified 140 novel RT loci, 70 uniquely from pixel-level analysis, 28 uniquely from the fPC analysis, with 42 found in both”. What may be causing these differences, is it purely by chance that sometimes fPC scores detect signals when pixel-level fails to? Can this difference occur because of the multiple-testing correction (possibly setting a very conservative p-value cutoff) in pixel-level which we may not have to worry about with just 6 fPC scores? Intuitively, it feels like both fPCs and pixel-level should produce very similar results (because fPC scores are good low-dimension representation of the pixel-level). If anything, perhaps the pixel-level grid feels more informative than the fPC scores, because fPCs are low-dimensional representation of the pixel-level grid.

Some of these differences do indeed occur due to the heavy multiple testing burden of the pixel-level approach, and the more lenient threshold used for the fPC

analyses. For example, rs56277867 meets the FPC threshold of $p < 9.33E-9$, for its association with FPC4 ($p=9.61E-10$). The smallest pixel p-value is $2.64E-11$, falling just short of the Bonferroni corrected threshold of $p < 1.72E-12$.

As mentioned in our preceding response, there were some SNP associations picked up through the FPC analyses, which were not identified through the pixel level analyses, even at the standard genome-wide significant threshold ($p < 5E-8$). Some further examples of this are rs10771345 and rs9330797, both identified through the GWAS of FPC6; this FPC captures nasal/temporal differences, and this pattern is apparent in the associations with these two SNPs; yet no single pixel is significant at the $p < 5E-8$ level.

Conversely, there were a number of SNPs that were identified through the pixel-level analyses, but not the FPC analyses. Two examples of this are below. The signals for rs6663840 and rs726799 are highly focused, and these patterns are evidently not being strongly captured by the FPCs. The minimum p-values from the FPC analyses for these SNPs were for FPC1 ($p=5.44E-3$) and FPC3 ($p=7.93E-8$), respectively.

If possible, can you cluster the OCT scans using the fPC scores (this would be similar to how one may use Principal Components Analysis for clustering)? Would there be any clear and obvious clusters, and would there be any interesting characteristics for these clusters (e.g., a few specific traits are more enriched for some specific clusters)?

We thank the reviewer for this interesting idea. Scans could indeed be clustered using the fPC scores, and linking clusters to disease outcomes could be of particular clinical interest. We consider this analysis to be beyond the scope of the present paper, but agree this could be a promising avenue for future work.

Minor comments:

1. “Overrepresentation” and “over-representation” are being used interchangeably; maybe just stick with one version?

We thank the reviewer for pointing out these inconsistencies. We have updated so “over-representation” is consistently used.

2. Sometimes “fPC” is used, and other times “FPC” (for example, “FPC” is used in Method section “Associations in non-European ancestries”).

We have updated all instances to “FPC”.

Reviewer #3 (Remarks to the Author):

The term and concept of “oculomics”, using data to characterise ocular and systemic health have earned a well-deserved legitimacy. Powered by the availability of data from the usual UK Biobank, the manuscript by Jackson et al. is another attempt to use retinal scans as mirrors of systemic health. This work describes how several thousands of OCT scans are compared, pixel by pixel, starting from the fovea, and features obtained are correlated with genotypes, metabolites, and ICD10 codes. The authors find many new genetic loci and associations, and the retinal features are correlated in some ways with a variety of conditions, cardio- and cerebrovascular, chronic and acute inflammatory diseases, liver and respiratory diseases, gastrointestinal neuro-psychiatric, metabolic and cancer.

There are, however, several question marks and potential problems with this paper. When something is associated with almost everything, as in this case, readers may be forgiven for thinking of potential problems with the data structure, analyses, or bias. This manuscript does not contain enough information to dispel such doubts. There is no real replication or validation (similar data are difficult to find outside the UK Biobank), and the authors have not provided any explanation, however tentative, for the underlying biological reasons for some of these associations. Some appear credible (such as multiple sclerosis), but others more far-fetched (like gastric reflux and toothaches). Often, it is unclear what the implications and conclusions from the results are (e.g., “alcohol taken with meals” – is that good or bad for retinal thickness?). There is a lingering doubt that results sometimes derived from a handful of cases (ICD10 codes) may have been overinterpreted and perhaps affected by bias.

Our work aims to paint a picture of the landscape of retinal thickness across the macula, well beyond the use of retinal thickness as a biomarker for systemic health. We summarised patterns of retinal thickness, with our novel FPC approach, prior to any disease association for example. We also note that our association approaches are performed agnostically: we summarise both thickening and thinning.

We deliberately steered clear of defining “good” or “bad” for retinal thickness. Retinal degeneration is defined by the thinning of the layers, yet many retinal diseases tend to increase retinal thickness. On average, we observe a thinning of retinal layers associated with disease presence but we intentionally avoid assuming that either direction of effect has to be considered detrimental. Given these complex patterns of association and the fact that we have extended the interpretation to a comprehensive spatial evaluation across the macula, our attempts to describe the retinal thickness landscape are intended as an omnibus of findings to explore. Deeper investigation of underlying drivers are not the focus of this landscape paper. We note however that in this revision we have added several layers of additional

analyses which go some way towards interpretation of some of these associations, including pheWAS, heritability analyses and improved SNP to gene mappings. These will aid biological understanding of retinal function.

Another related question concerns the alignment of OCT retinal features across different images and whether the pixels at a certain distance and location in each image are truly comparable with others. Different factors affect the magnification of retinal images differently, and retinas of larger eyes are physically more stretched than those in smaller eyes. Therefore, two pixels with the same ETDRS grid coordinates may have different characteristics and are not necessarily comparable. If true, the phenotype used for analyses would have flaws that a simple linear correction (age, sex, and body height) can't correct. For example, spherical equivalent is the product of corneal power and eye size, each of which would deform images in different ways. And astigmatism would present a different kind of challenge altogether.

Many previous research papers, Knight et al. ⁴, Chaglasian et al. ⁴, and Mahmoudinezhad et al. ⁵, have aligned OCT volumes by registering the fovea and measuring anatomical regions relative to the fovea. While we agree with the reviewer that there may be magnification effects or distortion effects due to astigmatism or other factors, these effects also apply to the ETDRS grid. However, these effects do not invalidate our work nor much of the previous research in retinal analysis.

Furthermore, if anatomical alignment to the fovea were invalid or fortuitous, one would expect the peaks of SNP significance to be randomly uniformly distributed and isolated and create a bias towards the null hypothesis. However, our analysis showed the GWAS coefficients and p-values for the 200+ significant SNP loci demonstrate distinct patterns spatial clustering and diffusion from a peak, which strongly implies spatial patterns. Given the reviewer's concerns, if they are true, our associations are likely an underestimation of the true strength and thus we feel strongly that our results are a conservative estimate of the true association. We give examples below for some of the top SNPs:

In the top row of images captured from our informational website, <http://retinomics.org>, we show the coefficients and p-values of rs1504080 across the macula. rs1504080 exhibits positive coefficients to retinal thickness around the fovea in the center of the macula, but negative coefficients in the para-and peri-fovea regions. In addition, the p-values in the middle image of the top row show a strong temporal effect for rs1504080. Next, the second row shows the effects and p-values of rs3138142, which has strong positive effects in the inner macular ring, except at the fovea, with corresponding extreme p-values in that region.

The authors do not provide any metrics of reliability for their genetic results. No measure of inflation is provided.

We agree this is an important omission. For all GWAS, we have assessed inflation of the test statistics, using the LD score regression (LDSC) intercept. Overall, there was no evidence of inflation. For the pixel-level GWASs, the mean LDSC intercept was 1.022 (range 0.980 - 1.058). For fPCs 1-6, the LDSC intercepts were: 1.034; 0.993; 1.005; 1.010; 1.007; 1.025.

The LDSC analyses are now described in the methods (p. 28 lines 1-2):
*For all GWAS, inflation was assessed using the LD-score regression intercept, as implemented by LDSC v.1.0.1*⁶.

The LDSC intercepts for both the pixel-level and FPC GWASs are referred to in the main text, and summarised in Supplementary Figure 12.

Text added to results, page 4, lines 18-20:

For all GWAS, inflation was assessed using the LD-score regression intercept, as implemented by LDSC v.1.0.1²⁵. LD-score regression intercepts close to 1 show that results were well calibrated (Supplementary Figure 12).

Supplementary Figure 12: A) LD-score intercepts for each pixel, B) LD-score intercepts, with standard errors, for FPCs 1 to 6.

In addition to potential issues with the phenotype, the authors also ran their analyses using simple linear regression. It is well known that one-third of the samples are related to one another, which in theory could unduly inflate the significance of the results. It is possible that with decreasing sample sizes, related samples would gradually become fewer and less problematic, but the gold standard is to run linear mixed regressions, which account for sample non-independence. The inflation factor parameters and LD regression output are routine diagnostic tools that are reported to reassure readers of the quality of results obtained from genetic analyses.

Our analyses were restricted to unrelated individuals only, thus our GWAS only required a simple linear model, with adjustment for ancestry principal components to account for population stratification, we believe to be appropriate. Indeed, we found no evidence for inflation of the GWAS test statistics (as per previous response).

We opted to restrict analyses to unrelated individuals, since our analyses only included the 10% of UK Biobank with OCT images, and so excluding related individuals had less of an impact on sample size. This strategy has been adopted by others previously^{1,2,7}. Indeed, out of 44,786 European ancestry individuals passing all OCT scan/phenotype quality control (QC), and genotype QC, 1638 were excluded due to relatedness. Had we included related individuals, we would, as the reviewer notes, needed to have used a more sophisticated model, and this would have been prohibitively computationally expensive.

Furthermore, for our non-genetic analyses, inclusion of related individuals would have required individuals to be divided into distinct families for our 'omics linear modelling using limma. Hence we decided that the slight drop in power due to exclusion of related individuals was acceptable, given not only the benefits for the streamlining of analyses across the 'omics but also given how computationally expensive it would have been to include, and account for relatedness for the pixel level genetic association analyses in particular.

The restriction to unrelated individuals is described in the "Quality control based on genetic data" section of the methods (p.27 lines 16-21), and stated in the results ("Overview of analysed cohort", p.3 line 36).

The authors rightly use Bonferroni correction for multiple testing. If properly applied, this correction would have been conservative, since pixels are not independent of other adjacent pixels. However, the use of Bonferroni is inconsistent. While it is generally used to control for the total number of multiple tests, here it is applied to families of analyses, with a different correction factor for the pixel analyses and another for fPCs. And in all tables and supplements, everything above the nominal threshold of $5e-8$ is reported. It is unclear if or what multiple testing correction was applied to metabolites, their ratios and ICD code regressions. The manuscript does not seem to follow a single and consistent strategy on multiple testing correction.

We appreciate that our description of multiple testing should have been clearer. We have now addressed this in the manuscript and the methods section. To clarify to the reviewer: we decided that multiple testing should be applied only within each data axis as this is often the case in multi-omics studies. Additionally, we decided to use two approaches for multiple testing corrections in our manuscript that align with those traditionally in literature used for each of the respective data axes.

The Bonferroni correction is a stringent approach often used for GWAS. As the reviewer notes, this adjustment is likely to be overly conservative, and it is for this reason we additionally included all results at the standard "genome-wide significance" level, with $p < 5e-8$ in supplementary tables 5 and 6. The loci we report in the main manuscript, and all of our GWAS follow-up were restricted to loci meeting the strict Bonferroni corrected thresholds ($p < 1.72e-12$ for pixel-level analyses; $p < 8.33e-9$ for FPC analyses), as these were the genetic associations in which we had greatest confidence.

Metabolomics and other omics approaches generally use correction for false discovery rate, in particular by applying Benjamini-Hochberg ad-hoc correction to the obtained p-values. However, as this is a special case we also perform correction for multiple testing using Bonferroni for all data axes (within-axis only) and these are additionally reported in the supplementary tables 17-21 and 23-24. Hence we not only applied principled, widely accepted, omics specific false discovery corrections but also reported those which may be of interest to readers that work across these omics.

The authors' correlation of genetic effect sizes across different ethnic groups lacks sufficient detail. These correlations would be primarily more a function of linkage disequilibrium differences between two populations than informative replication of associations. A good comparison should take LD into account (such as the Popcorn method).

We agree that some of the genetic heterogeneity between ancestries will be due to differences in linkage disequilibrium, we thank the reviewer for this suggestion. We attempted to run Popcorn on the genome-wide results for FPCs 1-6. Unfortunately, given the small sample sizes available for the AFR and CSA subsets, the GWAS on these populations were underpowered, and the heritability estimates for most of the FPCs were not significantly different to one (in some instances, the point estimate for h^2 given by popcorn was <1). For Popcorn to give meaningful correlations, the utilised GWAS are required to have significant heritability; where the heritability is very low the resulting estimate of the genetic correlation is biased.⁸ Thus, we have not added the popcorn results to the paper.

We have added some text to the discussion noting the limitations of our approach using simple correlations (page 11, lines 20-25):

Our cross-ancestry genetic association analyses suggested significant genetic heterogeneity in RT; however the number of individuals with EAS and AFR ancestries included in these analyses was relatively modest sample size, and we were therefore underpowered to comprehensively explore differences across ancestries. Further investigation in individuals with greater ancestral diversity is required and emerging efforts to increase diversity in population cohorts will hopefully allow this.

Details on how the metabolites were analyzed are also very sparse. These analyses are anything but straightforward, and major potential confounders need to be controlled for. BMI and sex are two examples, both of which were already associated with the outcome and therefore potential sources of systematic bias.

All methods used to clean and analyse the metabolomics data are detailed in the methods section ("*Metabolomics data preparation*" – page 34 lines 13-24). All our models were corrected for genetically derived sex at birth, age, imaging device number, standing height, mean refractive error measured by spherical equivalent, eye and the first ten genetic principal components (see Methods "*-Omics and clinical data analyses*" – page 35-36). As a quality control check, we now tried to fit an additional model which includes BMI as a covariate. We obtained very similar results to our original model as presented in **Supplementary Fig 21**. As we want to keep the models homogenous across different data types, correction for BMI for metabolomics data is thereby presented only as a quality check.

The BMI sensitivity analyses is described in the methods (page 35, lines 24-28):

The effect of BMI on the association between retinal thickness and metabolic levels was tested via the inclusion of BMI in the regression model. The correlation between the effect size of each metabolite for each pixel across the corrected and non-corrected model was estimated and visualised for each metabolic class.

We have now added further text to the main manuscript (p.7 lines 2-8):

After extraction and cleaning (Methods), 325 metabolic measures, from 10,668 participants were included in the association analyses. We examined associations at the single metabolite level and performed hierarchical clustering of both metabolites and pixels to investigate potential spatial effects (Methods). After correction for False Discovery Rate (FDR) using Benjamini-Hochberg multiple testing correction (Methods), all metabolites were significantly associated with RT at least one pixel (Supplementary Table 17, Extended Data 7A). The inclusion of BMI as a covariate did not change the global results (Methods, Supplementary Figure 21).

Supplementary Figure 21. Correlation of effects across all pixels, for each metabolite (divided into metabolic groups) derived from models correcting and not correcting for BMI.

The authors have tested for associations between polygenic risk scores (PRS) of retinal features with diseases and other complex traits. It is not clear what the justification was for only including a small subset (approximately 10%) of the entire Biobank sample. This may have been the subset for which they had optical coherence tomography (OCT) data, but it is unclear why the authors preferred to run analyses on a much smaller sample when they could have calculated PRS for all half a million participants.

We apologise that the description for these analyses was not clear. To clarify, no PRS were generated for retinal thickness, based on our performed GWAS, rather these analyses examined PRS scores derived from other GWAS, and their

associations with OCT derived retinal thickness. As such, these analyses were restricted to participants for whom RT and genetic data were available and then we constructed PRS scores for multiple traits. Subsequently, we tested each PRS for association with retinal thickness.

To provide clarification in the paper, we have added further text to the methods (page 33, lines 1-2):

Polygenic Risk Scores (PRS) and Genetic Risk Scores (GRS) were used to investigate the relationship between genetic susceptibility to a number of diseases and traits and RT.

And in the main manuscript (p.9 lines 5-8):

We also examined associations at the pixel-level between RT, and PRS or genetic risk scores, for a number of diseases (for simplicity collectively referred as PRS below, see Methods). This allowed for investigation of associations between RT and genetic susceptibility to diseases, including a number of retinal diseases, and related phenotypes.

The UK Biobank has information on just over 400 cases with multiple sclerosis, how many cases were available for each of the analyses reported?

The number of MS cases in our analysis was 133; this is included in the results section of the main manuscript (p.8, line 22). Additionally, the number of available cases for each disease Phecode is available in supplementary table 20.

The authors defined diseases based on ICD10 codes, which only partially overlap with the ICD9 information. It is quite common to have a disease reported only in ICD9 and not under ICD10 in the UK Biobank and there is a possibility that some genuine cases could have been counted as controls.

We agree with the reviewer and re-performed the analysis by using both ICD9 as well as ICD10. We have now combined these into PheCodes, an approach similar to that utilised by Zekavat and colleagues ⁷. This allowed us to additionally simplify the diagnoses into easier to interpret terms. ICD10 code regression on pixel thickness was hence replaced by PheCode regression on pixel thickness. The interpretation of our results did not change as the major associations were still observed and in the same order of importance when using PheCodes instead of ICD10 only.

The full results of the new PheCodes analyses are available in Supplementary Table 20.

Ambiguous or loose terminology is frequently seen in the manuscript. It is not clear what “clustering” analyses were conducted, as they are not described in the Methods section.

Throughout the manuscript, we utilised unsupervised hierarchical clustering, with the number of clusters for each analysis determined visually based on the resulting dendrogram. All the clustering methods have been described in the supplementary methods: for the GWAS, page 31, lines 16-18, and for all other 'omics on page 36, lines 11-16.

The authors mention “Concordance was lower in EUR vs AFR” (perhaps meaning “correlation”?), and there is an interchangeable use of Spearman’s “rho” and “r” (the former, often used in the text, applies to entire populations, the latter to representative population samples, which would be appropriate in this manuscript).

To improve clarity, we have rewritten the “Cross-ancestry comparisons, and biological insight” section of the manuscript, in response to this and the reviewer’s subsequent comment (page 6, lines 2-10):

We assessed the effects of all loci identified in our EUR discovery analyses in the AFR and CSA ancestry individuals (Extended Data 5, Supplementary Table 13). In the pixel-level, and FPC analyses, the EUR and CSA effect estimates for the top pixel, or top FPC showed strong correlations (pixel-wise $r=0.618$, $p= 1.141e-24$; FPC $r = 0.630$, $p = 2.00e-14$), with weaker correlations for the EUR and AFR effects (pixel-wise $r = 0.305$, $p=4.54e-06$; FPC $r = 0.249$, $p = 6.8e-03$). For each sentinel identified in the pixel-level analysis, we also examined whether the effect estimates for all pixels across the scan were correlated between ancestries. The median correlation, for EUR versus CSA individuals was $r = 0.306$, with 28.0% of sentinel SNPs having $r \geq 0.5$. Again, correlations were lower in EUR vs AFR (median correlation: $r = 0.195$; 11.5% sentinels with $r \geq 0.5$).

The term “median scan-wise correlation” is difficult to understand, and “log-adjusted p-value values” is almost certainly incorrect.

Please see previous response. We have now adjusted the terminology and corrected the typographical errors.

As with wording, the authors could, and should, have been more careful with formatting. Tables and figures were not following the same order in which they were called for in the text. The manuscript starts with a Supplementary Figure 7, and Supplementary Figure 1 is only seen on Page 32. Often, there is very little legend to explain the content (for example, “Supplementary Figure 14: Clustering of pixels, based on SNP effects for loci identified through the pixel-level analyses”).

Supplementary Figures 1-6 (now Supplementary Figures 1-7) are referred to in the “Retinal Thickness Imaging Data” section of the results (p.3 Line 9), with Supplementary 7 (updated to Supplementary Figure 8) referenced in the same paragraph (page 3 line 12).

We agree with the reviewer regarding the legend for supplementary 14 (now supplementary figure 17), and we have now extended the legend with further detail:

Supplementary Figure 17. Clustering of pixels, based on SNP effects for loci identified through the pixel-level analyses. In this figure, every pixel is colored according to the cluster it was assigned to in the unsupervised hierarchical clustering analysis. The ETDRs grid is also presented to ease interpretation of the clusters' locations.

In many instances, unsupported statements are made, e.g. “rs183659670 is associated with NNAT expression in the retina” or “another novel signal ... in a region with a significant chromatin interaction with SOX2 in adult and fetal cortex.” (no citation, no such analyses reported in the Methods).

We have double checked all statements related to SNP to gene mapping are cross-referenced. In the two instances given by the reviewer, we have made the following updates (note - the wording of these sentences has additionally been updated following completion of the GWAS and eQTL co-localisation analyses, and chromatin interaction analyses [reviewer #4]):

Page 6 lines 14-15:

“rs183659670 is associated with NNAT expression in the retina (PP=0.95 of shared GWAS and eQTL signal, Supplementary Table 8)”

Page 5 lines 33-35:

“Another novel signal rs74614808 (pixel-wise min $P=5.60e-15$) was detected in a region with a significant chromatin interaction with SOX2 in retina, adult and fetal cortex (Supplementary Table 9).”

Reviewer #4 (Remarks to the Author):

The manuscript "Multi-omic spatial effects on high-resolution AI-derived retinal thickness" by Jackson et al. presents a comprehensive analysis of genetic, omic, blood trait and disease associations with spatially defined retinal thickness in the human macula, the region in the retina in charge of central vision. This is the first study to apply a deep learning approach, convolutional neural network to over 90k optical coherence tomography (OCT) images from ~55k participants in the UK biobank, to measure retinal thickness in >29,000 pixels in the macula, generating a high resolution spatial retina thickness map. By further applying functional principal component analysis (fPCA) to the 29k pixel-level retinal thickness phenotypes, the authors were able to identified 6 fPCs that captured the majority of variance explained across all participants, which can more easily be used for downstream functional analyses. Genome-wide association study analysis was applied to the retinal thickness across >29k pixels and to the top 6 fPCs, identifying 224 unique loci across all pixels and fPCs tested, with a high concentration of associations in the foveal region. Over half these loci were not found to be associated with retinal thickness in previous GWAS the authors inspected, though there is a more recent study they didn't compare their results to - Zekavat, et al. *Sci Transl Med* 2024 (PMID: 38266105; Zekavat et al., medRxiv May 2023, <https://doi.org/10.1101/2023.05.16.23290063>). Many of the new loci were found to be associated with ocular traits, such as intraocular pressure, vertical cup-disc ratio, and age-related macular degeneration. While the sample size for non-European participants is 40 fold smaller than that of European participants, the authors evaluated the genetic correlation with spatial retinal thickness between the EUR, African and central and south Asian sample sets, and found stronger correlation between the EUR and CSA genetic associations compared to EUR and AFR. The authors integrated various functional genomic data and annotations to prioritize candidate genes for the retinal thickness loci. In addition, the authors test for associations of circulating metabolites, blood and immune biomarkers, ICD10 codes and genetic risk scores of different ocular and systemic disease with macular thickness across the fine-scale spatial grid. They found that specific metabolite associations clustered spatially in the retina, that retinal thinning was associated with multiple systemic disorders such as multiple sclerosis in the nasal perifoveal region, and that parafoveal thickness was particularly susceptible to systemic insults. While some of these retinal thickness associations with systemic diseases were previously reported (Zekavat et al 2024) this is the first time that correlation with spatial location was examined, which may have important clinical implications. Notably, the authors created an interactive website to visualize the OCT features, GWAS, metabolite associations, and other results reported in this paper, which is incredible valuable for investigating the results and generating or testing new hypothesis for follow up studies.

This study was overall done with statistical rigor, and could have great value to the ocular disease community as well as those studying other complex traits. However, there are several important points on the methodologies used or factors considered in the analyses that I think need to be addressed for the results and conclusions to be sound.

Major comments:

1. The authors removed individuals from the analysis in with ocular diseases or taking medications for eye diseases listed in Supplementary table 1. I would recommend also removing individuals that have had laser or surgery treatments to the eye used to treat glaucoma or high eye pressure. The UK biobank codes for these are: 5327 - “Ever had laser treatment for glaucoma or high eye pressure”, 5326 “Ever had surgery for glaucoma or high eye pressure”

We thank the reviewer for highlighting this additional exclusion for identifying individuals with glaucoma for exclusion. Amongst our primary analysed cohort of 43,148 individuals, we found 52 (0.1%) individuals who had had laser treatment or surgery for glaucoma or high eye pressure, for one or both eyes. To assess the impact of the inclusion of these individuals in our analyses, we repeated the genetic associations, excluding these individuals. We compared the estimated effect sizes and p-values for the analyses with, and without these individuals, and found negligible differences.

Pixel-wise results:

FPC results:

Given the very high concordance of these results, we have not included these sensitivity analyses in the supplementary data. However, we have noted in the discussion that our filtering strategy will likely not have removed all disease individuals (page 11 lines 9-11):

“While our analyses primarily intended to capture variation in “healthy” eyes, it is likely our exclusion criteria will not have captured all individuals with retinal disease, and a small minority of diseased participants will have been included.”

2. The authors performed GWAS of pixel-level retinal thickness phenotypes and the top 6 fPC score phenotypes using the glm (generalized linear regression) command in PLINK 2.0. glm fits a linear regression model for continuous variables, such as the retinal thickness phenotypes. A linear regression model assume that the residuals are normally distributed. Have the authors checked the distributions of these phenotypes? Are they normally distributed? If there is skewness, rank inverse normal transformation of the response variable might be appropriate to prevent potential outlier associations.

The below figures show the distribution of 100 randomly selected pixels, and the Pearson's moment coefficient of skewness for each pixel across the scan.

Distributions of 100 random pixels

Pearson's moment coefficient of skewness

Pixels showed approximately normal distributions, with slight skewness, both positive and negative, across different areas of the scan. To allay concerns of any deviations from normality, we undertook sensitivity analyses for all reported pixel-level genetic associations, with the phenotype undergoing a rank inverse normal transformation. The p-values from the transformed phenotypes were highly similar to those from the untransformed phenotypes. As expected, effect estimates vary in terms of magnitude due to the different pixel variances (effect size for most significant pixel for each SNP plotted), however they are highly correlated.

The FPCs all follow approximately normal distributions:

Distributions of FPCs 1 to 6.

We have now mentioned the normality of the phenotypes in the methods (page 27, lines 28-29):

All phenotypes were confirmed to follow approximately normal distributions.

3. Smoking is a known environmental factor that affects retina degeneration and retinal layer thickness as shown in Zekavat SM, et al. Sci Transl Med. 2024 (PMID: 38266105). Tobacco use was also amongst the top ICD10 traits associated with retinal thickness phenotypes in this paper in Figure 4d. I think it would be important to check how the GWAS results change when correcting for smoking status in the GWAS model, as this may be a confounding factor. I would recommend adding smoking as a covariate also in the association models of the PRS, metabolomics, ICD10, CCI scores, blood traits and infection antigens with retinal thickness phenotypes.

We have performed additional sensitivity analyses to assess the impact of including smoking as a covariate in our genetic, and phecodes analyses. After restricting to individuals whose smoking status could be determined, there were 3666 current smokers, 15,228 previous smokers and 24,076 never smokers included in the analyses.

For the GWAS, these sensitivity analyses are described in the methods (page 32, lines 21-25):

We undertook sensitivity analyses on the subset of 42,970 individuals, for whom smoking status could be derived (UK Biobank Field 20116: 3666 current smokers; 15,228 previous smokers; 24,076 never smokers. Genetic associations for the reported sentinel SNPs were repeated for the top pixel and/or FPC, with smoking status included as an additional covariate. The effect estimates and p-values were then compared for analyses with and without adjustment for smoking.

We have added the following results to the paper:

Page 6 lines 43-47:

Given the previously demonstrated role of smoking on retina degeneration⁴⁵ and retinal layers thickness¹⁵, we undertook sensitivity analyses for our reported sentinel SNPs, to determine the effect of adjusting for smoking status (current/former/never). This additional adjustment made limited difference to the associations (Supplementary Figure 20), suggesting the identified genetic associations were not acting via smoking.

The following plots were added as Supplementary Figure 20:

A.

B.

Supplementary Figure 20. Smoking Sensitivity analyses. Comparison of effect estimates and -log10 P-values for SNPs identified as meeting the Bonferroni corrected significance threshold in: A) the pixel-level analyses and B) GWAS of FPCs 1 to 6.

Full genome-wide association analyses with additional adjustments for smoking, were only repeated for the FPCs. For all FPCs, the genetic correlation (LDSC) between the analyses, with and without smoking was not significantly different from 1 (Figure below). For brevity, we have not included this additional result in the manuscript.

We additionally included smoking in the PheCodes analysis and compared results with and without smoking as a covariate. To do so, correlation of the effect size of each PheCode was performed across all pixels between the two models.

This sensitivity analysis has been added to the methods (page 35, lines 24-28):

The effect of BMI on the association between retinal thickness and metabolic levels was tested via the inclusion of BMI in the regression model. The correlation between the effect size of each metabolite for each pixel across the corrected and non-corrected model was estimated and visualised for each metabolic class. The same approach was used to test for the effect of smoking status on the association between retinal thickness and PheCodes.

The results are stated in the methods (page 8, lines 20-21):

The inclusion of smoking as an additional covariate did not change the results (Supplementary Figure 22).

Supplementary Figure 22. Correlation of effects across all pixels, for each PheCode (divided into PheCode Classes) derived from models correcting and not correcting for smoking.

4. In the Methods section under “Identification of independent significant loci” the authors describe the LD clumping algorithm they employed across all associated variants and retinal thickness pixel-level phenotypes to identify the independent signals for the genome-wide significant loci. It might be informative to use COJO (<https://yanglab.westlake.edu.cn/software/gcta/>) or another similar tool applied to summary statistics to identify secondary, tertiary, etc independent signals in an associated region.

Thank you for this suggestion. For all identified loci, we ran cojo for the top associated pixel/FPC to identify potential secondary signals. We then used conditional analyses to confirm secondary signals were independent. Through this analyses we identified seven additional SNPs in each of the pixel-level, and FPC analyses.

The identification of secondary signals is now outlined in the methods (page 29, lines 21-28):

For all loci meeting the Bonferroni-corrected significance level outlined above, we sought to identify whether there were independent, secondary SNP association signals in the region. For the top pixel/FPC for each locus, we first ran cojo-select, in GCTA v.1.94.1⁷, using p-value thresholds of $p < 1.72e-12$ for the pixel-level analyses and $p < 8.33e-9$ for the FPC analyses. If any SNPs additional to the sentinel were identified through the cojo analyses, we ran conditional analyses using PLINK2 for that region with the sentinel SNP as a covariate, to determine true independent signals. We report any SNP meeting $p < 1.72e-12$ (pixel-level) or $p < 8.33e-9$ for (FPCs) in the conditional analyses as a secondary SNP; these are listed in the “SNP_Secondary” column in Supplementary Tables 5 and 6.

And the results:

page 4, lines 27-29:

We identify 224 RT-associated genetic loci that met the pixel-level Bonferroni corrected threshold (Supplementary Table 5). Seven loci harboured secondary signals, giving 231 independently associated signals in total.

page 4, lines 36-37:

The FPC-based RT association analyses identified 127 independently associated SNPs at 120 loci (Supplementary Table 6), ...

All GWAS follow-up analyses, (ie SNP to gene mapping, gene over-representation analyses etc.) were repeated, with the secondary signals added and all relevant result figures and tables updated.

5. The authors removed loci with less than 5 LD-independent SNPs in the region with $P < 5e-5$, as these were assumed to be false positives. Did the authors test for anti-correlation with minor allele frequency, MAF? I am concerned that true associations may be discarded in this approach, as low frequency variants may have few or no LD proxy variants.

This filter was a pragmatic decision, aimed at reducing false positives, potentially at the expense of excluding true positives. In the pixel level analyses, there were 137 loci where the top SNP had $p < 5E-8$, but fewer than 5 SNPs with $p < 5E-5$. Of these, there were just two where the top SNP met the Bonferroni corrected $p < 1.72E-12$. We agree our approach may result in true associations being discarded, and have noted this limitation in the discussion.

Discussion (page 11, lines 15-20):

This work has some shortcomings. The complexity of the high resolution phenotypes used here, required us to make a number of pragmatic decisions regarding our approach. Among these was the decision to impute missing data points, and in our GWAS to exclude loci with fewer than five SNPs showing association with $p < 5E-5$. The imputation assumed data were missing at random, however missingness could have been more likely to occur for pixels with more extreme values of RT, while the GWAS filtering may have resulted in some true associations being excluded.

6. What is the lambda of the GWAS? This is a standard measure used to evaluate the calibration of the GWAS and identify GWAS with significant inflation or deflation. If lambda values are found to be significantly higher than 1 (e.g., >1.1, >1.2), computing the intercept in the LD score regression model can help determine if the large lambda is due to uncorrected bias (e.g., population stratification) or to true polygenicity. This step is particularly important since there is no replication of the genetic associations in an independent study. If the lambda is high, I would recommend reporting the genome-wide significant loci using genomic control adjusted p-values.

As stated in our response to reviewer 3, we agree that measures of inflation should be included. For all GWAS, we used the LD score regression (LDSC) intercept to assess inflation of the test statistics. These analyses are now described in the methods (page 28, lines 1-2), and the LDSC intercepts for both the pixel-level and FPC GWASs are summarised in Supplementary Figure 12 (Figure shown in response to reviewer 3).

For the pixel-level GWASs, the mean LDSC intercept was 1.022 (range 0.980 - 1.058). For fPCs 1-6, the LDSC intercepts were: 1.034; 0.993; 1.005; 1.010; 1.007; 1.025. Overall, there was no evidence of inflation due to population stratification.

7. The authors test whether any of the 224 loci they found to be associated with retinal thickness phenotypes have been previously reported, inspecting three publications, but not the latest one which performed GWAS on the different retinal layer thicknesses measured in macula OCT images from the UK biobank - Zekavat SM et al., 2024 (PMID: 38266105). Some of the 140 novel loci reported in this manuscript might have been found in Zekavat et al.

The comparison with previous papers, now includes Zekavat et al. 94 out of the 224 of the loci identified through the pixel-level analyses were identified in Zekavat et al., with 50/120 loci from the FPC analyses. This brings the number of novel loci down to 123.

The overlap of signals with those identified previously, is summarised in two new supplementary tables (Supplementary Table 11 and 12). We have also removed the sentences regarding the rs77877421 (*FOXP1*) signal from the “Novel RT associations” section, as this no longer represents a novel signal.

In addition, the Upsettr plot in Extended Data 3 has been updated:

The comparison of p-values for SNPs identified by Zekavat et al. has been appended to Supplementary Figure 12:

D.

8. To prioritize the causal genes underlying the GWAS loci of retina thickness and fPC phenotypes, the authors integrated a variety of functional genomics data with the genetic loci. I have a few suggestions to make their analysis more specific and rigorous. They mapped eQTLs in retina, blood and brain to GWAS loci using linkage disequilibrium between the variants. It has been shown that not all GWAS variants which are also significant eQTLs at $FDR < 0.05$ or that are in LD with an eQTL, their causal mechanism is via the eQTL (GTEx consortium Science 2020; Barbeira et al Genome Biology 2021, PMID: 33499903). A more rigorous approach is to use Bayesian colocalization methods applied to co-occurring GWAS and QTL signals to assess the posterior probability that the GWAS and eQTL loci are tagging

the same causal variant, such as eCAVIAR (PMID: 27866706) or SharePro (<https://www.biorxiv.org/content/10.1101/2023.07.24.550431v1>), both of which assume allelic heterogeneity in the loci, which is often the case, and take local LD into account.

We thank the reviewer for this suggestion. We have now followed up on candidate eQTL mapped SNPs identified with FUMA by running a colocalisation analysis between the identified mapped snps and GTEx/eyeGEx eQTL data. Details on the colocalisation are found in the methods (page 30, lines 20-23):

For candidate eQTL-mapped SNPs identified by FUMA, bayesian colocalisation analysis was performed using coloc (version 5.2.3)¹⁶ with SuSiE (version 0.12.4)¹⁷, accounting for multiple causal variants. Summary statistics for each SNP's top pixel/FPC were colocalised with GTEX v7 (<https://gtexportal.org/home/downloads/adult-gtex/ctl>) and EyeGEx eQTL data using default parameters. Results were restricted to those with mapped eQTL $P < 5e-06$ and a posterior probability of a shared causal variant (H_4) > 0.8 .

The co-localisation results have been added to Supplementary Table 8.

The authors use chromatin interaction maps (Hi-C) from fetal and adult human brain to link GWAS SNPs to genes based on overlapping enhancer-promoter and promoter-promoter interaction regions. While I assume that many chromatin interactions are shared between brain and retina, there are chromatin interactions that are likely retina-specific. Thus, I would recommend also testing the overlap of retina Hi-C interactions (Marchal et al., Nat Comm 2022; PMID: 36207300) with the GWAS loci to prioritize potential implicated genes.

We have extended the chromatin interaction mapping to include retina utilising the suggested HiC data.

This is now outline in the methods (page 30, lines 27-30):

Chromatin interaction mapping was also performed with pre-processed HiC data for retina. (<https://www.ncbi.nlm.nih.gov/geo/query/acc.cgi?acc=GSE202471>). Both directions were considered and restricted to interactions with FDR < 0.01 .

The results have been added to Supplementary Table 9.

9. The authors test whether the 224 retinal thickness loci identified are associated with other ocular diseases and traits, stating that “Many of these have been previously implicated in GWAS for ocular traits”. They highlight/describe a few results on page 7 referencing Figure 5. Can the authors be more specific about how many of the 224 retinal thickness loci (known and new) were associated with other ocular traits with published GWAS, and can the authors summarize these results in Supplementary tables or add relevant columns to Supplementary Tables 5 and 6. It would also be interesting to perform a PheWAS of the 224 loci with other traits to test the pleiotropic effect of these variants on non-ocular traits, or look these results up in the UKB or other biobank PheWAS.

We thank the reviewer for the suggestion. We performed a PheWAS for all sentinel and secondary snps with OpenGWAS as now described in the methods (page 32, lines 1-7):

Phenome wide association studies (PheWAS) were performed on sentinel and secondary SNPs with curated publicly available data through the openGWAS platform (<https://gwas.mrcieu.ac.uk/>), via the R-package ieugwasr (version 1.0.0). SNPs were queried against batches: EBI database of complete GWAS summary data (ebi-a), GWAS summary datasets generated by many different consortia initially developed for MR-Base rounds 1 and 2 (ieu-a, ieu-b), pan-ancestry genetic analysis of the UK Biobank (ukb-e) and an expanded set of genome-wide association studies of brain imaging phenotypes in UK Biobank (ubm-b), filtered for significance threshold $P < 1e-05$.

The PheWAS results are now presented in Supplementary Table 14, Supplementary Figure 18, and described in the main paper (page 6, lines 12-21):

We undertook a phenome-wide association study (PheWAS) for all SNPs (sentinel, and secondary SNPs) identified in the pixel-level and FPC analyses. 1,666 traits were identified with at least one SNP association ($P < 1e-05$) (Supplementary Table 14 and Supplementary Figure 18). The top associated trait was height, which showed association with 46 pixel-wise SNPs and 25 FPC-wise SNPs. Two of the previous GWAS of retinal thickness layers, Retinal nerve fibre layer (RNFL) thickness and ganglion cell inner plexiform layer (GCIPL) thickness, were included in the PheWAS database. Both measurements were amongst the top associated phenotypes for pixel-wise (RNFL: 3rd, GCIPL: 9th) and FPC-wise (RNFL: 3rd, GCIPL: 24th) SNPs.

Supplementary Figure 18. Top 30 traits by number of SNP associations from PheWAS for SNPs identified by FPC (A) and pixel-wise (B) analyses.

Amongst the top 30 traits for FPC and pixel level analyses were ocular traits, retinal nerve fibre layer (RNFL) thickness and ganglion cell inner plexiform layer (GC IPL) thickness. All identified ocular trait associations for both analyses are displayed below, these plots were not included in the main paper.

10. The authors note in the Methods section on page 28 that they used polygenic risk scores (PRS) to investigate the relationships of traits with retina thickness. The authors should distinguish between two types of PRS here. The PRS they define as type 1 (trait PRS) and type 2 (metabolic PRS, blood abundance of metabolites) that the authors downloaded from the UK biobank, are indeed PRS as these scores considered the effect sizes of variants genome-wide irrespective of whether they passed a given significance cutoff. However, the type 3 PRS or internal PRS computed for macular disorder and retinal thickness related measures, and the PRS computed for metabolites measured with the Biocrates platform, based on the definition on page 28, lines 21-24, are not PRS but rather a genetic risk score (GRS), as the score only considers the effects of a set of variants that are genome-wide significant in the respective GWAS and not all variants genome-wide. Please clarify this point in the paper and use the correct terminology in the Results section. PRS have been shown to be more powerful than GRS, and it would be better to use a PRS if the summary statistics (beta and std) are available for the given GWAS and not only the genome-wide significant loci.

We thank the reviewer for pointing out this difference as we were not aware of it.

We have clarified that we use both polygenic risk scores and genetic risk scores in the manuscript (page 9, lines 6-7):

We also examined associations at the pixel-level between RT, and PRS or genetic risk scores, for a number of diseases (for simplicity collectively referred as PRS below, see Methods).

The distinction between PRS and GRS is now clarified in the methods, and we highlight the difference in prediction power that the two approaches present (page 33, lines 1-20):

Polygenic Risk Scores (PRS) and Genetic Risk Scores (GRS) were used to investigate the relationship between genetic susceptibility to a number of diseases and traits and RT. We used three types of scores in this study. Type 1 scores comprise trait PRS and GRS composed mainly of scores describing disease susceptibility. The trait PRS are further subdivided into three subclasses. Two of these, the Standard and Enhanced PRSs, were downloaded from the UK Biobank having been generated and deposited in the UKBiobank as described by Thompson and colleagues²⁵ (Categories 301 and 302). These PRS had already undergone extensive QC and data processing and thus were used, as is, in the association analyses.

Type 2 scores comprises metabolic GRS measuring genetic predisposition to blood abundance of a set of metabolites, measured with the Biocrates platform³². Finally, type 3 scores are defined as Internal GRS (Type 3 GRS). These were GRS of particular relevance to the RT phenotype and included two macular disorders and two measures related to the RT phenotype. This included GRS for: (i) Macular Telangiectasia type 2²⁶, (ii) Age-related macular degeneration²⁷, (iii) Retinal thickness²⁸, and (iv) retinal venular and arteriolar calibre^{29,30}. To construct the type 2 and type 3 GRS, we extracted the top SNPs at each genome-wide significant locus associated with the trait as reported by the authors. For type 2 metabolic GRS, we discarded all metabolites that had fewer than three top SNPs available to construct the GRS. We then used the R-package bigSNPR (version 1.12.2)³¹ to extract the selected SNPs from UK Biobank and create the scores.

It is worth noticing that prediction power of PRS and GRS may vary substantially as the first are constructed with prediction in mind while the second are derived from standard GWAS results.

11. For the correlation between PRS and ICD10 pixel-level effects, I would suggest correcting for age, sex, top 10 genotype PCs.

We apologise that the description for these analyses was unclear. These correlations compare the effect estimates from the models examining pheCode-RT associations, with the effect estimates from the corresponding trait PRS-RT associations. The pheCode-RT associations, and the PRS-RT associations were both corrected for all covariates. We examined the correlation of effect estimates

from these two models, across all pixels. We have reworded the methods to clarify (page 35, lines 32-34):

Correlations between the estimated effects from the PheCode-RT pixel-level associations, and the corresponding effect estimates from the PRS-RT associations, were examined by calculating the Pearson's correlation coefficient across all pixels.

And the results (page 9, lines 30-33):

There were 25 diseases captured through both the PheCodes and PRS analyses in this study. For each disease, we examined the correlation of the effect estimates from these two analyses, and we found the highest association agreement for type 2 and type 1 diabetes, glaucoma, psoriasis, hypertension, asthma and Parkinson's disease (Supplementary Table 22).

12. The authors refer to PRS, metabolomics, ICD10 scores, Charlson Comorbidity Index (CCI) computed based on disease categories, blood traits and infection antigens all as omics data in the Results and Methods (-Omics Data Analyses). I think the authors should make the distinction in the text between the omics data, such as PRS and metabolomics, and phenotype/disease or lab based traits, such as ICD10 scores, CCI and blood traits and infection antigens.

Throughout, we now refer to “-omics” analyses (for metabolomics and PRS), and “clinical data” analyses for the PheCodes, blood traits and infection antigens.

Minor comments:

1. I would change the header “Main” to “Introduction”.

This has been updated.

2. In the Introduction, the authors can add a few more references to the first sentence, such as = Liang et al., Cell Genomics 2023, Yan et al Cell Reports 2020; and add Monavarfeshi et al., PNAS 2023 as a reference for single cell data in the optic nerve head and optic nerve.

These additional references have been added.

3. The authors should add a few references to the second sentence in the Introduction on page 1, lines 4-5 and lines 8-15.

We have added a number of references to the introduction.

4. There are a few additional references that would be appropriate to add on page 1, lines 18-22, such as the association of OCT-measured retinal layer thickness with cognitive function and decline (Sekimitsu et al., British Journal of Ophthalmology 2023; PMID: 36990674) or with neuropsychiatric and cardiometabolic disorders (Zekavat et al. , 2024; PMID: 38266105).

These additional citations have been added.

5. On page 1, line 28, the authors should add the reference Zekavat et al. , 2024; PMID: 38266105 to the GWAS of retinal sublayers performed.

This reference has been added.

6. In the final paragraph of the Introduction, where the authors summarize the main results of the paper, the authors can add a few more sentences with a bit more detail, be a bit more specific on the main findings.

We have updated this section as follows
(page 2, lines 33-38):

Our analyses reveal why retinal imaging is ‘a window to the brain’, by detecting associations with neurological disorders such as multiple sclerosis, and highlight retinal thickness as potential biomarker for vascular, and endocrine disorders. Furthermore we show for several diseases and metabolites, there are specific regions of the macula driving associations. Our genetic analyses identify 294 RT associated genetic loci, highlighting the complex spatial anatomy of the macula, providing context for novel candidate genes and their biological mechanisms.

7. In the description of the quality control of the genotype data in the Methods section, the authors note that pairwise relationships, up to 3rd degree were removed. Can they add more details on the method and cutoff they used to identify related individuals.

We have added the following clarification to the methods (page 27, lines 17-19):

Pairwise relationships, up to 3rd degree were identified, based on the kinship coefficient estimates from KING¹³, as made available through the UK Biobank genetic data release (3rd degree relatives defined as kinship >0.088).

8. In the Methods section under “Genome-wide Association Analyses” the authors note that the covariates were standardized with mean 0 and covariance 1. Did the authors mean ‘variance’ of 1, not covariance?

Thank you for highlighting this error; we have updated this to “variance”.

9. In the Results section, it would be helpful to clarify that the functional PCA is applied to the pixel level retinal thickness phenotypes from the CNN analysis of OCTs. This is described in the Methods section, but would be good to mention this in the Results section, so it will be easier for the reader to follow what was done.

We have updated the manuscript to make this clearer (page 3, lines 24-25):

We applied this approach to the reprocessed scans (RT measurements across all 29,041 pixels), using tensor product splines as base functions, using the MFPCA R package.

10. Figure 4 legend, a legend for the bubble size and color is missing. Please define ORA in figure legend.

We have now corrected the legend (page 20):

Figure 4. PheCodes results. *A) Barplot showing number of significant association between PheCodes and thickness at different pixels. B) Heatmap showing significant association between PheCodes and thickness FPCs. Color represents association direction and magnitude (red=negative, blue=positive). Size represents the magnitude of association. C) 2D smoothed results from over-representation analysis (ORA) on main PheCodes results. D) volcano plot showing average effect size across retinal thickness and median $-\log_{10}(p\text{-value})$ for each PheCode.*

11. On page 5, lines 18-19 I think the reference is to Suppl Table 13, not Suppl Table 12. I might recommend adding the FDR cutoff used for the gene set enrichment results in the Results section.

We have updated this reference, and all of the Supplementary Table numbering, given the revisions.

We have updated the gene set ORA results as follows (page 6, lines 39-40):

Both analyses were enriched (Benjamini-Hochberg corrected $p < 0.05$) for genes involved in eye and visual system development and cell differentiation.

12. The authors performed unsupervised clustering of the SNPs based on their association results with ~29k pixel level phenotypes and identified 10 clusters (show in Suppl Figure 13). Can the authors list the SNPs found for each of the 10 clusters in a Supplementary table?

SNPs are now annotated by cluster in Supplementary Table 5.

13. On page 2, lines 38-45, the authors describe some of the patterns observed from performing association testing of the retinal thickness phenotypes

(pixel level phenotypes and 6 fPCs) with different characteristics including sex, age, height and spherical equivalence, however they don't show these results in supplementary tables or figures. Supplementary figure 10 shows the spatial distribution of the association significance ($-\log P$) with the pixel level phenotypes, but not the effect size and direction of effect. This would be informative to show and record. Can the authors include these results in Suppl Tables and figures.

The results for the associations with the FPCs are summarised in Supplementary table 3.

For the pixel-level results, Supplementary Figure 10 has been updated to include both the effect and minus \log_{10} p-values:

Supplementary Figure 10: Pixel-level associations with basic characteristics (effect, left; $-\log_{10}$ p-value, right). Pixel-level associations with A) sex B) standing Height C) refractive error (spherical equivalent), D) age and E) age-squared. F) shows the marginal effect of age for central foveal pixel (64_128).

14. The authors created an “evidenceScore” to prioritize candidate genes in a given GWAS locus based on the number of lines of evidence for that gene. They

point to a column in Supplementary Table 10, which I assume is the column named “sum_cols”. It would be good to clarify this in the Methods section.

Apologies, we have now corrected this error.

15. In the ‘Metabolomic association analyses’ section in the Results section, it would be informative to add some statistics on the top metabolite associations highlighted, e.g., beta, adjusted p-value, number of pixels they are associated with.

We have now included the number of significant pixels, average beta, and median -log₁₀ p-value for all metabolite associations mentioned in the manuscript.

16. It might be worth noting that most of the metabolite associations with retinal thickness phenotypes were with pixels in the foveal and nasal regions.

This observation is in line with our ORA results, which is mentioned in the main manuscript (page 7, lines 44-45: “Pixel over-representation analysis (ORA) revealed that the parafovea, particularly its lower temporal side, was susceptible to metabolic influences (Figure 3C).”).

17. On page 6, lines 10-11, the authors point to Supplementary Table 15, for the age by metabolite interaction results, but I do not see those results there. Also, on page 6, lines 14-18, the authors discuss the association results of the metabolite PRS with retinal thickness and point to Supplementary Table 16 and Extended Data 8D. There is no panel D in Extended Data 8.

We apologise for the unclear table labelling. The referenced table (now Supplementary Table 18) does indeed contain the interaction results; the description/columns have been updated for clarity.

The “D” in Extended Data 8 was a typographical error which we have now corrected.

18. When you first mention testing for association of PRS with retinal thickness in the Results section in regards to metabolic PRS on page 6, line 14-18, I think it would be helpful to add a sentence on how the PRS correlation for metabolic PRS as with the other traits, was applied to all pixel retinal thicknesses and that the average beta and median p-value across all pixels was used to evaluate correlation (as in Supplementary Table 16 and other Suppl tables), and point the reader to the Methods for more details. This was not clear to me until I read the Methods section.

We apologise that this was not clear. We now confirm the metabolite PRS associations are presented analogously to the measured metabolite associations (page 8, lines 6-7):

Associations with metabolite PRSs and all pixels were performed in the same fashion as for the measured metabolites

19. When the ICD10 disease associations with retinal thickness results are described on page 6, it would be informative to provide information on the direction of effect, i.e. is that disease associated with thinning or thickness of the retina.

In the manuscript results, we have now included the number of cases, number of significant pixels, average beta, and median $-\log_{10}$ p-value for all Phecode associations.

20. In the Discussion on page, lines 21-23, the authors note that “These complementary approaches, applied to AI-reprocessed data identified a greater number of loci, than previous GWAS conducted with RT derived directly from the TOPCON scanner, despite the heavy multiple testing burden.” This is not precise, as in a recent GWAS and PheWAS of retinal thickness measures based on the TOPCON scanner (Zekavat et al., 2024), 259 unique loci were found that is comparable to the 224 loci found in this manuscript. You can emphasize the added value of your work with the spatial information.

We have updated this section of the discussion (page 10, lines 19-23):

These complementary approaches, applied to AI-reprocessed data identified a number of novel loci, not identified in previous GWAS conducted with RT derived directly from the TOPCON scanner, despite the heavy multiple testing burden. Moreover, these results give micron-level spatial resolution, highlighting distinct spatial patterns for several loci, whereas previous analyses were averaged across regions on the ETDRS, or Macula 6 grids.

21. In the Discussion on the bottom of page 8 to page 9, the authors note “Future work will extend this approach to retinal sub-layers which will further tease apart many of the associations since sub-layers are composed of particular cell types.” This has been done in Zekavat et al., 2024, where they measured the thicknesses of different retinal layers in macula OCTs, such as the inner retinal fiber layer or ganglion cell complex layer. Please edit accordingly.

We have updated this section of the discussion (page 11, lines 2-5):

Existing GWAS have examined thickness of retinal sub-layers, averaged over the entire macula area^{9,14,15}, and have shown some associations to be layer specific; future work will extend our spatial approach to the retinal sub-layers, allowing us to further tease apart many of the associations and explore cell type specific effects.

1. Currant, H. *et al.* Genetic variation affects morphological retinal phenotypes extracted from UK Biobank optical coherence tomography images. *PLoS Genet.* **17**, e1009497 (2021).
2. Currant, H. *et al.* Sub-cellular level resolution of common genetic variation in the photoreceptor layer identifies continuum between rare disease and common variation. *PLoS Genet.* **19**, e1010587 (2023).
3. Gao, X. R., Huang, H. & Kim, H. Genome-wide association analyses identify 139 loci associated with macular thickness in the UK Biobank cohort. *Hum. Mol. Genet.* **28**, 1162–1172 (2019).
4. Chaglasian, M. *et al.* The development of a reference database with the Topcon 3D OCT-1 Maestro. *Clin. Ophthalmol.* **12**, 849–857 (2018).
5. Mahmoudinezhad, G. *et al.* Local Macular Thickness Relationships between 2 OCT Devices. *Ophthalmol Glaucoma* **4**, 209–215 (2021).
6. Bulik-Sullivan, B. K. *et al.* LD Score regression distinguishes confounding from polygenicity in genome-wide association studies. *Nat. Genet.* **47**, 291–295 (2015).
7. Zekavat, S. M. *et al.* Phenome- and genome-wide analyses of retinal optical coherence tomography images identify links between ocular and systemic health. *Sci. Transl. Med.* **16**, eadg4517 (2024).
8. Galinsky, K. J. *et al.* Estimating cross-population genetic correlations of causal effect sizes. *Genet. Epidemiol.* **43**, 180–188 (2019).
9. Zekavat, S. M. *et al.* Phenome- and genome-wide analyses of retinal optical coherence tomography images identify links between ocular and systemic health. *Sci. Transl. Med.* **16**, eadg4517 (2024).

10. Yang, J. *et al.* Conditional and joint multiple-SNP analysis of GWAS summary statistics identifies additional variants influencing complex traits. *Nat. Genet.* **44**, 369–75, S1-3 (2012).
11. Giambartolomei, C. *et al.* Bayesian test for colocalisation between pairs of genetic association studies using summary statistics. *PLoS Genet.* **10**, e1004383 (2014).
12. Wallace, C. A more accurate method for colocalisation analysis allowing for multiple causal variants. *PLoS Genet.* **17**, e1009440 (2021).
13. Manichaikul, A. *et al.* Robust relationship inference in genome-wide association studies. *Bioinformatics* **26**, 2867–2873 (2010).
14. Currant, H. *et al.* Sub-cellular level resolution of common genetic variation in the photoreceptor layer identifies continuum between rare disease and common variation. *PLoS Genet.* **19**, e1010587 (2023).
15. Currant, H. *et al.* Genetic variation affects morphological retinal phenotypes extracted from UK Biobank optical coherence tomography images. *PLoS Genet.* **17**, e1009497 (2021).

Notes

- We have used the following formatting in responses:

- **General responses**
- *Changes made to the manuscript*

Line numbers relate to the clean document (no tracked changes)

Reviewer #1 (Remarks to the Author):

The authors have provided detailed responses to address most of my comments. The replies have provided a clearer understanding of the approach and its limitations. To further strengthen the paper, I have several additional questions:

1. Regarding Question 6 and your response to Question 2, after performing A* segmentation, did you manually accept/reject the generated masks individually, ultimately retaining 6,409 out of 8,500 pairs? You mentioned that there are 1,250 hard pairs of B-scans with manually reviewed segmentation. Did you evaluate the performance of the A* algorithm on these 1,250 hard examples?

We could not evaluate the performance of the A* predictions directly, as we did not have ground truth segmentations. Instead we used heuristics to accept adequate A* segmentations and hard examples as follows:

1. For 8500 OCT scans, we computed their A* segmentation masks, and then calculated the minimum retinal thickness and the standard deviation for each B-scan from these segmentation masks. Only samples with a minimal retinal thickness between 30 and 80 pixels, which translates to 180 to 480 μm at an axial resolution of 6 $\mu\text{m}/\text{pixel}$) and $\text{SD} \leq 10$ pixels (60 μm) were accepted. This resulted in 6,409 accepted pairs of B-scans and predicted masks.

2. Another 200 hard examples were found by the entropy of the predicted segmentation mask; there was a previous miscommunication on how many hard examples were used. We first sampled and processed 1,250 FDA data items and excluded segmentation errors (as in mentioned above 1). Next, we computed the sum of entropy for each segmentation mask where each pixel had a probability value of being foreground. We then regarded examples with high entropy value as hard examples, which were sorted in descending order, and the top 200 images were added to the original dataset for training the deep learning model.

We have added the following text to the methods (page 27, line 31 to page 28, line 13):

A is widely used in pathfinding and can be used to track retina layer boundaries in this use case. However, A* is not robust to poor contrast or abnormal brightness and random noise, and quality control on A* predicted segmentation was performed. Specifically, A* was applied to 1 random B-scan from 8500 FDA data. Then only masks whose ILM-RPE thickness were between 30 and 80 pixels, which translates to 180 to 480 μm at an axial resolution of 6 $\mu\text{m}/\text{pixel}$) and $SD \leq 10$ pixels (60 μm) were accepted. This resulted in 6409 after the thickness restriction. Next, another 1250 FDA files were sampled and 1 random B-scan from them predicted using A*; these masks were filtered for minimal ILM-RPE thickness as above. Then we computed the sum of entropy for each segmentation mask where each pixel had a probability value of being foreground. We regarded examples with high entropy value as hard examples, which were sorted in descending order, and the top 200 images by entropy were added to the 6409 B-scans for training the deep learning model. Examples of A-star segmentations that were accepted and rejected are provided in Supplementary Figure 1, which shows three pairs of images with the raw B-scan on the left and the A-star segmentation on the right. The second and third segmentations in this figure were rejected as they were not fully correct. These reviewed B-scans and A-star segmentation masks were used as training data for the second part of the pipeline, the deep learning model called PSPNet.*

2. In Supplementary Figure 8, it is noted that the cleaned dataset includes 54,844 subjects with OCT data from at least one eye and corresponding phenotypes. I assume that most image quality control was performed by this step. Could you provide more details on how the dataset was further reduced to 6,409 + 1,250 pairs of B-scans and their corresponding masks, given that each eye has 128 B-scans and there are over 54,000 subjects?

We would like to clarify that the 6,409 + 200 pairs of B-scans and their corresponding masks represent a specific, small subset of the entire dataset that was used for supervised training of our deep learning (DL) segmentation model. This happened before it was applied to the full UK Biobank cohort. This subset was carefully selected as explained in the response above (with clarification now added to the methods). After the deep learning model was trained, we performed inference on the full dataset for the analysis, which initially includes OCT data from over 85,000 individuals.

To further clarify, the 54,844 individuals noted in Supplementary 9, refers to the number of individuals with scans following application of the DL model, alignment, and all quality control filtering.

3. One limitation of the paper is its lack of independent validation, which is not available at the current stage. If new data are available in the coming years,

authors should make the detailed results (e.g., summary statistics) and code available for other groups to replicate some of the findings.

We agree with the importance of sharing summary results and code to facilitate replication of these results in future.

We have created a repository on osf.io to host summary statistics for this project (<https://doi.org/10.17605/OSF.IO/KZUGV>). Here we have made available the full pixel-level results for the metabolite, disease, genetic risk score, blood cell trait, and inflammatory marker associations. Full GWAS summary statistics for the six FPCs are also available to download. Due to the large size of the pixel-level GWAS results, we will not be uploading the full 29,041 sets of genome-wide summary statistics to this online repository; however we have added a statement to the “Data availability” section that these results can be made available to other researchers on request. Look-ups for particular SNPs can be made using our online portal (<https://retinomics.org/>).

Code for the genetic analyses can be found at <https://github.com/bahlolab/retinalThicknessGWAS> , while code and data for the retinomics.org can be found at <https://github.com/uw-biomedical-ml/ukb-retinomics>

We have updated the code and data availability sections accordingly:

Data Availability

All data used in this study can be accessed from the UK Biobank upon request, via an application process (<https://www.ukbiobank.ac.uk/>).

RT pixel-level and the six 2D FPC data will be returned to the UK Biobank.

GWAS summary statistics for the six 2D FPCs, and pixel-level results for all ‘omics and disease code analyses, are available to download from

<https://doi.org/10.17605/OSF.IO/KZUGV>. Due to the size of the full pixel-level GWAS results, these will be made available to researchers on request. Look-ups for pixel-level results, including SNPs, and all ‘omics data, can be made using our online portal (<https://retinomics.org/>).

Code Availability

We used publicly available open-source software for these analyses. Scripts for the genetic analyses can be found at <https://github.com/bahlolab/retinalThicknessGWAS>.

Code and data for the retinomics.org can be found at <https://github.com/uw-biomedical-ml/ukb-retinomics>.

4. The fonts in most figures are too small to read.

We have updated figures with larger font sizes, where applicable, and possible (Figures 1, 2 and 4).

Reviewer #2 (Remarks to the Author):

Authors addressed reviews in their entirety.

Reviewer #3 (Remarks to the Author):

The second version of the manuscript is a definite improvement over the previous one. The authors have diligently gone through the reviewers' comments and responded to the critiques.

While there are many improvements for which the authors may be commended, this second review will mainly focus on what could be better, or what remains unclear in the manuscript.

Before getting to more detail, it seems that sometimes the authors in their rebuttal tend to simply engage in a discussion with the reviewers - this is great but some of the reviewers' comments were about a lack of clarity and reviewers were simply acting as a sample of the journal's readers. Some of the questions were legitimate and the reader would benefit from the inclusion in the manuscript or supplementary information of some clarifications that the authors are currently sharing with the reviewers. This is a suggestion that the authors may want to consider moving forward.

We thank the reviewer for their positive comments regarding the revised paper. We acknowledge that some of the queries raised by the reviewers indicate that some parts of the manuscript would benefit from further clarification. We have aimed to take on board this suggestion with this second round of comments.

1. In their rebuttal letter, the authors retort that 'We deliberately steered clear of defining "good" or "bad" for retinal thickness...We summarised patterns of retinal thickness, with our novel FPC approach, prior to any disease association for example... These will aid biological understanding of retinal function.' While this is

irreproachable, there was a broader question that went beyond the use of epithets for conditions and phenotypes. There are questions about the presence of the confounding or bias that the authors seem to be evading. It is unclear what certain associations and correlations are teaching us. Some of these associations are exceedingly broad, ranging from dental symptoms, esophageal conditions, to habits or lifestyles. The doubt is that these associations (that were removed from the current version) may not be aiding us much but are arising from some biasing factor (very hard to imagine any real biological relationship between GERD or toothaches and retinal images).

A previous comment was about pixels with the same ETDRS grid coordinates having potentially different characteristics, maybe due to biasing effects of refraction, eye size, etc. The authors' response is that "previous research papers... have aligned OCT volumes by registering the fovea and measuring anatomical regions relative to the fovea... these effects do not invalidate our work nor much of the previous research in retinal analysis." And then "if anatomical alignment to the fovea were invalid or fortuitous, one would expect the peaks of SNP significance to be randomly uniformly distributed and isolated and create a bias towards the null hypothesis".

It may be difficult to accept the validity of this argument. While there may be some precedent set by similar publications, there are questions about how a potential bias may have affected the results of this paper. The assumption the authors are making is that anatomical alignment to the fovea were fortuitous, or random and a source of type II error. The real concern here is the possibility that an anatomical misalignment happened due to the presence of biasing factors. For instance, a non-random factor (refraction, eye size) may affect the size of the retinal images, the concentric patterns described on pages 4, line 30 and page 5 line 17 of the revised version of the manuscript may be explained by non-genetic factors (socio-economic, nutrition, education). Bias related to socio-economic factors and nutrition has in the past been difficult to control by simply using linear adjustment and is frequently encountered in genetic association and often generates correlations similar to those observed by the authors. This point is not meant to be destructive or invalidate this work and manuscript. But when the most significant correlations for

the first factor on table S15 are still reported for "milk type used" and "bread type: brown", this will take some acknowledging and possibly explaining.

We thank the reviewer for raising many important points regarding the findings of our work. We agree, further discussion regarding the limitations of our approach, and caution regarding the interpretation of our findings, is required. We have added the following paragraph to the discussion (page 11, line 41 to page 12, line 2), where we have attempted to condense the reviewer's main points:

This paper developed a hypothesis-generating resource and describes a series of exploratory analyses. While we have attempted to account for major confounders (sex, age, height, spherical equivalent, and genetic ancestry) in all our analyses, our models will inevitably not have accounted for all confounding factors, and some of our association results will have been impacted by bias. As in traditional, observational epidemiology, genetic associations may be subject to confounding, in particular by socioeconomic factors.^{60,61} Bias may have also been introduced through our phenotyping approach, for example if the accuracy of foveal alignment was impacted by some non-random factor, like eye size. Future studies, which more comprehensively account for confounders and biases through tailored models, specific to the disease/trait being examined, are needed to fully characterize all reported associations. We also note that whilst the UKBB was intended to be representative of the ageing UK population, it is subject to a range of selection biases.⁶⁴ All described associations thus must be interpreted in this context.

2. The authors respond to the criticism about inconsistent use of multiple testing correction methodologies, and they clarify their position that despite the choice to adopt Bonferroni (an over-conservative approach), to use multiple approaches for multiple testing corrections in our manuscript that align with those traditionally in literature used for each of the respective data axes. The reviewer has sympathy for this argument, but this is also why this approach that seeks to have the best of several worlds may unfortunately be called inconsistent. Previous publications will follow have ways to correct for multiple testing, but rarely they would use all of them in one place, and much less frequently ignore them in the supplementary tables (as tables S5 and S6) where the correction is made for the genome-wide level but not number of phenotypes. This reviewer is not recommending going to extremes, but some effort may need to be made in that respect.

Previously, Supplementary tables 5 and 6 included all variants meeting the genome-wide significance level ($p < 5E-8$), with those meeting the Bonferroni corrected significance level additionally highlighted by the "bonferroniSig" column

($p < 1.72e-12$ for pixel-level analyses; $p < 8.33e-9$ for FPC analyses). The loci we report in the main manuscript, and all of our GWAS follow-up were restricted to loci meeting the strict Bonferroni corrected thresholds. We included variants with $p < 5E-8$, but not meeting the Bonferroni corrected thresholds in these tables, as we thought they may still be of interest to readers, in particular given the overly conservative nature of this multiple-testing correction.

We appreciate this was not fully clear, and the results presented in these supplementary tables were somewhat incongruous with the results reported in the main paper. We agree it would be more consistent to only include in supplementary table 5 and 6, the loci that meet the strict Bonferroni-corrected thresholds; we have updated these tables accordingly.

3. In the previous iteration, the reviewers found that descriptions of the metabolome statistical analyses were not exhaustive and clear. The authors responded by writing that "all our models were corrected for genetically derived sex at birth, age, imaging device number, standing height, mean refractive error measured by spherical equivalent, eye and the first ten genetic principal components".

This reviewer finds this description unclear; 'limma' is just a package and linear models can be very different. What goes into the design matrix, how was the relationship between the different dependent variables modeled? For some people familiar with 'limma', Bayesian statistics are applied to independent variables (originally RNA probes, but this can be easily extended to other omics, such as metabolites). The authors mention that an empirical Bayes framework to model relationships between outcome variables (here RT pixels), which is all great, but model specifications are not well explained or documented.

We appreciate that the description of these analyses were unclear. The analyses applied to the metabolomics data is the same as for the other data axes analysed (except for the genomics). For this reason, there is no specific metabolomics statistical analysis section but rather a "Metabolomics data preparation" section as well as a section called "Omics and clinical data analyses" which contains the modelling also used for the metabolomics studies. We have included additional details to the latter section, to clarify further the model matrix and the linear models that were fit (page 39, lines 6 to17):

Thirdly, association testing was performed using the R package limma (version 3.50.3)³⁶ as follows: The “assay matrix” containing the retinal thickness measurements, was constructed as a row for each pixel analysed and a column for each subject. The model matrix was composed of columns containing the factor of interest, the intercept, and a column for each of the covariates: (i) genetically derived sex at birth, (ii) age, (iii) imaging device number, (iv) standing height, (v) mean refractive error measured by spherical equivalent, (vi) eye and (vii) the first ten genetic principal components. Participant data presenting with missing information for any of these covariates was discarded. Regression analysis was performed by building separate independent least-squares regression models for each pixel thickness, using the lmFit command, with the assay and model matrices as input. The function eBayes was then used to borrow information between pixels about the variability of the data and apply the empirical Bayes statistics to the results from the linear models, hence increasing power. This process was repeated for each factor in each data axis analysed.

4. A very minor point: the authors write in their letter that "had we included related individuals, we would, as the reviewer notes, needed to have used a more sophisticated model, and this would have been prohibitively computationally expensive." The authors' approach (linear regressions and, not linear mixed models) is very reasonable. Having said that, these techniques are not such a big deal, are widely used and computationally efficient since they take only hours to complete. However, this is a minor point, and the reviewer is happy to accept the authors' approach to GWAS and the drop in power is negligible, just as the authors state in their reply.

We thank the reviewer for accepting our rationale for this decision.

Reviewer #4 (Remarks to the Author):

The authors have done a lot of work to address my comments most of which are satisfactory. There are just a few mostly minor points/comments that I think need to be addressed or to be corrected before the manuscript can be accepted for publication. The pages and line numbers I refer to below were taken from the revised manuscript with the edits tracked in red (478786_1_related_ms_9158970_sspfgs.pdf).

1. The authors performed colocalization analysis between the GTEx and Retina eQTLs and the GWAS loci of retina thickness and fPC phenotypes, and used Hi-C from retina and brain to propose target genes. Can the authors add to a table the list of loci and genes proposed to be the underlying causal gene based on colocalization and Hi-C data. It would also be biologically meaningful to test in what pathways the significant genes are enriched from the colocalization and Hi-C analyses using gene set enrichment analysis.

We believe the results and analyses suggested by the reviewer are already included in the paper, but acknowledge they are not described clearly.

The full list of mapped genes is available in Supplementary Table 10. We apologise that this was not clear from the text, and have added some additional text to clarify:

Results (page 4, lines 42-46):

All loci were mapped to candidate genes via positional mapping, colocalisation with eQTLs in retina (eyeGX)²⁷, blood and brain (GTEx)²⁸, and chromatin interaction data from retina²⁹ and brain³⁰ (Supplementary Tables 7-9). All genes implicated via these mappings, or annotations are given in Supplementary Table 10. Associations and the candidate genes with the most lines of evidence for each of the RT loci are summarised in Figure 2.

The caption for Figure 2 now additionally includes the following in the caption:

“For each locus, the gene stated represents the top candidate gene, i.e. the gene with the most lines of evidence for that locus (highest “evidenceScore”, see methods). All candidate genes are listed in Supplementary Table 10.”

Additionally, we have performed gene-set over-representation analyses on mapped genes. We have slightly expanded the description of these analyses in the results for clarity (page 6, line 40-44):

Gene set over-representation analysis was performed on mapped candidate genes, implicated via the pixel-level and FPC analyses, to identify enrichment of gene-ontology terms. Both analyses were enriched (Benjamini-Hochberg corrected $p < 0.05$) for genes involved in eye and visual system development and morphogenesis, pattern specification process, and cell differentiation (Supplementary Table 16, Supplementary Figure 19).

2. In reply to question 9, it was not clear to me why the authors chose not to include the bar plots showing the top PheWAS results for ocular traits in the manuscript. Also, why are the number of variants associated with some of the ocular traits

different between the bar plots shown in the rebuttal below the plots included Suppl fig. 18 different than the bar plots included in Suppl fig. 18.

We have now included 2 further plots in supplementary figure 18 (c and d), showing the ocular traits (also shown below). We thank the reviewer for highlighting the numbers discrepancy; we have ensured this has been corrected in the new figures.

3. I would suggest the following edit in bold to this sentence that was added on page 33, lines 1-20: “It is worth noting that prediction power of PRS and GRS may vary substantially as the variant weights of the PRS were constructed to optimize prediction, while the variant weights of the GRS were derived from standard GWAS summary statistics.”

Thank you for this suggestion regarding wording, we have now updated this.

4. The authors have clarified that they use both polygenic risk scores and genetic risk scores in the manuscript on page 9, lines 28-30, by adding the following sentence (and a clear explanation in the Methods section): “We also examined

associations at the pixel-level between RT, and PRS or genetic risk scores, for a number of diseases (for simplicity collectively referred as PRS below, see Methods).” However, I would suggest not to change the word GRS to PRS in the Results section since they refer to different scores and have different levels of power. I would suggest keeping GRS and editing the sentence on page 9, lines 28-30, as follows:

“We also examined associations at the pixel-level between RT, and PRS or genetic risk scores (GRS), for a number of diseases (see Methods).”

We have now made it clear throughout, whether PRS or GRS were used for each association described in the text.

For the metabolites (page 8, lines 9-10):

“Metabolic polygenic risk scores (PRSs) have recently been endorsed as a tool for trait associations.⁵⁰ Associations with metabolite genetic risk scores (GRSs) and all pixels were performed...”

For disease related phenotypes (page 9, lines 14-26):

We also examined associations at the pixel-level between RT, and PRS or GRS, for a number of diseases (collectively referred as genetic scores below, see Methods). This allowed for investigation of associations between RT and genetic susceptibility to diseases, including a number of retinal diseases, and related phenotypes.

We examined 58 trait genetic scores from 43,147 participants, and after FDR correction, 48 scores were found to significantly affect the RT of at least one pixel (Supplementary Table 21). The averaged macular thickness¹⁸ GRS was the most significantly associated (sig. pixels: 29,041, av. beta: 1.08, median log₁₀(p): Inf), followed by retinal vascular caliber (sig. pixels: 29,040, av. beta: 0.17, median log₁₀(p): 14.83), AMD⁵⁵ (sig. pixels: 16,601, av. beta: -0.05, median log₁₀(p): 2.80), and MacTel³⁵ (sig. pixels: 15,966, av. beta: 0.06, median log₁₀(p): 2.68) (Supplementary Table 21). The more recent Han et al⁵⁵ AMD GRS resulted in a more powerful association signal than the UKBB PRS release, which is based on Fritsche et al.⁵⁶

5. On page 4, lines 27-28, the authors note: “We identify 224 unique RT-associated genetic loci that met the pixel-level Bonferroni corrected 28 threshold (Supplementary Table 5).” But I only see 221 variants with “clusterAssignment” values in Supplementary Table 5. Also, there are 959 variants in Supplementary Table 5. What are the other variants? Are they variants that pass genome-wide significance that are in LD with the lead 224 significant variants? Please clarify this

somewhere in the text, or if you have legends for the Supplementary tables you can add the description there.

We thank the reviewer for highlighting the missing cluster assignment for a small number of loci. We have now ensured all loci in supplementary table 5 are correctly annotated with their cluster.

Regarding the number of variants reported in Supplementary Table 5, again, we apologise that this was not clear. Previously, Supplementary table 5 included all variants meeting the genome-wide significance level ($p < 5E-8$), with those meeting the Bonferroni corrected significance level ($p < 1.72E-12$) additionally highlighted by the “bonferroniSig” column. The clustering, and all other follow-up of loci were restricted to the 224 loci meeting the Bonferroni corrected threshold. Given this was not clear in the previous version, and in response to reviewer 3’s comments, we have now only included loci meeting the Bonferroni corrected significance threshold in Supplementary Table 5.

6. In regards to my question #19 “When the ICD10 disease associations with retinal thickness results are described on page 6, it would be informative to provide information on the direction of effect, i.e. is that disease associated with thinning or thickness of the retina.” the authors did not seem to address it. The authors added number of cases, number of significant pixels, average beta, and median $-\log_{10}$ p-value for all Phecode associations, but I was asking about adding information on the relative direction of effect of the ICD10 associations on retinal thickness. For example do the phenotypes increase or decrease retinal thickness? Can the authors also add this to the relevant Supplementary table (22?). I think the authors tried to address this is on page 8. There seems to be several typos on page 8 lines 28-29 in this sentence (under the subheader “Disease PheCodes”): “More generally, even among those non-significant, there was a negative relationship between disease sta We assessed the effects tus and retinal thickness.”

We have revised the Disease PheCodes results section, to make it clearer where diseases were associated with retinal thinning or thickening, by providing an explanation of the betas (ie direction of effect), and specifically adding mention of thinning/thickening to the text, as appropriate. (page 8, line 18 to page 9, line 8):

We analysed pixel thickness association with 863 diseases as defined by PheCodes (combining ICD-9 and ICD-10 codes as defined by the Phecode Map 1.2 <https://phewascatalog.org/phecodes>). For each association we describe below, the direction of effect is captured by the average beta (av. beta) parameter. A positive beta indicates that the disease/trait is associated with increased retinal thickness, while a negative beta represents an association with retinal thinning.

Data were available for 35,900 participants. After FDR correction, 386 diseases (44%) were significantly associated with RT (Supplementary Table 20, Figure 4B), with the majority (82%) presented as a thinning effect as well as effects on multiple FPCs (Figure 4C). More generally, even among those non-significant, there was a negative relationship between disease status and retinal thickness. The inclusion of smoking as an additional covariate did not change the results (Supplementary Figure 22).

Multiple sclerosis (N = 133 cases reported in UKBB, (sig. pixels: 26,350, av. beta: -2.27, median log₁₀(p): 7.64) had the largest negative global effect, with retinal thinning observed in MS patients compared to controls, and the strongest effects observed in the nasal perifoveal region closest to the optic disc (Figure 5 E). MS results in oligodendrocyte demyelination, with impacts on the optic nerve as previously reported.⁵¹

Metabolic and cardiovascular disorders were among the diseases most strongly associated with thickness. Essential primary hypertension was associated with retinal thinning, and displayed the highest amount of global significance, likely due to the high prevalence of this disorder (N=10,767, sig. pixels: 29,041, av. beta: -0.37, median log₁₀(p): 11.17). Endocrine/metabolic class disorders were also highlighted among the top associated traits with retinal thinning including type 2 Diabetes (T2D) (N= 2,210, sig. pixels: 27,357, av. beta: -0.58, median log₁₀(p): 8.32, Figure 5F), hypercholesterolemia (N= 5,227, sig. pixels: 17,572, av. beta: -0.25, median log₁₀(p): 3.03), other chronic nonalcoholic liver disease (N= 397, sig. pixels: 22,419, av. beta: -0.85, median log₁₀(p): 3.73) and gout (N= 742, sig. pixels: 17,804, av. beta: -0.44, median log₁₀(p): 3.31). T2D and hypertension were also identified through our genetic correlation analyses, and are recognised comorbidities for multiple retinal disorders including AMD⁵² and MacTel⁵³. T2D has a recognised retinal complication in diabetic retinopathy, affecting about one-third of T2D patients, representing a major public health burden⁵⁴. These results complement our metabolic RT association results, where many of the metabolites most highly associated with RT have also been found to be associated with these disorders.

Investigation of retinal disorders was limited, given the study design of depletion of individuals with overt retinal disease. Nevertheless, we found retinal thinning to be associated with Cataract (N= 1,153, sig. pixels: 25,865, av. beta: -0.59, median log₁₀(p): 5.47), and senile cataracts, (N= 1,023, sig. pixels: 19,164, av. beta: -0.47, median log₁₀(p): 3.36) as well as astigmatism (N=402, sig. pixels: 14,167, av. beta: -0.67, median log₁₀(p): 2.67).

In addition, we have added an additional column (“directionEffect”) to Supplementary tables 20 (PheCodes) and 21 (genetic scores), to indicate whether each disease/trait is associated with thinning or thickening.

Reviewer #4 (Remarks to the Author):

The authors have addressed all of my comments. I just have a two small corrections:

1. In regards to the colocalization analysis performed between the pixel-level retina thickness and fPC phenotypes and GTEx and Retina eQTLs the authors used FUMA that uses an older release of GTEx (v7), which while is contained within GTEx release v8, it is not directly described in the cited publication: GTEx Consortium. The GTEx Consortium atlas of genetic regulatory effects across human tissues. Science 369, 1318–1330 (2020). To avoid confusion in the Results section, I would suggest adding the release version of GTEx as follows: “(GTEx v7)” in the Results section on page 4, line 43.

Thank you for this suggestion, we have added this clarification to the results.

2. On page 4, line 43, there is a typo - ‘eyeGX’ should be ‘EyeGEx’.

We have updated this.

3. In response to my previous comment #2, the authors have assessed the normality and skewness of the pixel-level retinal thickness phenotypes and the effect of performing rank inverse normal transformation of the pixel-level retinal thickness phenotypes and 6 fPC score phenotypes on the GWAS results. They show that they are not perfectly normally distributed and have shown that the beta’s and p-values highly correlate between the RINT normalized and raw values. The authors added in the Methods section the sentence: “All phenotypes were confirmed to follow approximately normal distributions. “ Based on their plots there is some skewness to their phenotype distributions, but it does not seem to have a significant effect on the GWAS results. I would suggest including the phenotype distributions, Pearson’s moment coefficient of skewness heatmap and the scatter plot between the GWAS summary statistics with and without the transformation in Supplementary Figures and referring to them in the Methods section. I think this would be useful for the readers to see.

We have added a short section to our methods detailing the trait transformation sensitivity analyses, and three corresponding Supplementary Figures (Supplementary Figures 35-37):

“Given the distribution of a subset of the pixel-level RT phenotypes showed some deviation from normality (Supplementary Figures 35 and 36), we also undertook sensitivity analyses by repeating top SNP-pixel associations using rank inverse normal transformed phenotypes. Comparison of effect sizes and p-values based on associations of raw RT phenotypes versus transformed phenotypes are highly consistent (Supplementary Figure 37), thus we only report the untransformed results.”